



# Downsizing parameter ensembles for simulations of extreme floods

Anna E. Sikorska-Senoner[1], Bettina Schaefli[2,3], and Jan Seibert[1,4]

[1]University of Zurich, Department of Geography, Zürich, Switzerland
[2]University of Lausanne, Institute of Earth Surface Dynamics, Lausanne, Switzerland
[3]Now at: University of Bern, Institute of Geography, Bern, Switzerland
[4]Swedish University of Agricultural Sciences, Department of Aquatic Sciences and Assessment, Uppsala, Sweden

**Correspondence:** Anna E. Sikorska-Senoner (anna.sikorska@geo.uzh.ch)

**Abstract.** For extreme flood estimation, simulation-based approaches represent an interesting alternative to purely statistical approaches, particularly if hydrograph shapes are required. Such simulation-based methods are adapted within continuous simulation frameworks that rely on statistical analyses of continuous streamflow time series derived from a hydrologic model fed with long precipitation time series. These frameworks are, however, affected by high computational demands, particularly

if floods with return periods >1000 years are of interest or if modelling uncertainty due to different sources (meteorological input or hydrologic model) is to be quantified. Here, we propose three methods for reducing the computational requirements for the hydrological simulations for extreme flood estimation, so that long streamflow time series can be analysed at a reduced computational cost. These methods rely on simulation of annual maxima and on analyzing their simulated range to downsize the hydrological parameter ensemble to a small number suitable for continuous simulation frameworks. The methods are

tested in a Swiss catchment with 10'000 years of synthetic streamflow data simulated with a weather generator. Our results demonstrate the reliability of the proposed downsizing methods for robust simulations of extreme floods with uncertainty. The methods are readily transferable to other situations where ensemble simulations are needed.

## 1   Introduction

The quantification of extreme floods and associated return periods remains a key issue for flood hazard management (Kochanek

et al., 2014). Extreme value analysis was largely developed in this field for the estimation of flood return periods (Katz et al., 2002); corresponding methods have been recently extended to bivariate approaches that assign probabilities jointly to flood peaks and flood volumes (Favre et al., 2004; De Michele et al., 2005; Brunner et al., 2016), and to trivariate approaches to assign probabilities jointly to flood peaks, volume and duration (Zhang and Singh Vijay, 2007); for a review of this field, see the work of Graler et al. (2013).

Most modern applications, however, not only require the estimation of extreme peak flow, associated flood volumes and duration, but also of hydrograph shapes, in particular in the context of reservoir design or for safety checks of hydraulic infrastructure (Kochanek et al., 2014; Gaál et al., 2015; Zeimetz et al., 2018). The key is thus the construction of design hydrographs with different shapes, peak flows and volumes, with a corresponding probability of occurrence. Such approaches can be roughly classified into methods that identify the shape of these design hydrographs based on observed data (Mediero


et al., 2010) or based on theoretical considerations (unit hydrographs) (Brunner et al., 2017) and regionalisation (Tung et al., 1997; Brunner et al., 2018a), or methods that rely on runoff simulations (Arnaud and Lavabre, 2002; Kuchment and Gelfan, 2011; Paquet et al., 2013).

Simulation-based methods for design or extreme flood estimation have a long history in hydrology (for a review see Boughton and Droop, 2003), and started with the classical event-based simulation with selected design storms (Eagleson, 30 1972; Chow et al., 1988; American Society of Civil Engineers, 1996). Those event-based methods are based on the concept that the design storm and flood have the same return period. Moreover, as they usually do not simulate antecedent conditions prior to the event and do not account explicitly for storm patterns (duration, spatial and temporal variability), they may yield biased flood frequency distributions (Viglione and Blöschl, 2009; Grimaldi et al., 2012a). Although some modern extensions of this event-based concept account for variable initial conditions prior to the event through sensitivity tests (Filipova et al., 2019), 35 most of the work using event-based simulations assume default initial conditions. Indeed, such event-based simulation is still in use, in particular in the context of probable maximum flood (PMF) estimation based on probable maximum precipitation (PMP) (Beauchamp et al., 2013; Gangrade et al., 2019).

Modern extensions of this approach, however, use continuous hydrologic modelling for design flood estimation, either i) to generate a range of initial conditions for use in combination with design storms (Paquet et al., 2013; Zeimetz et al., 2018) or 40 ii) to generate long discharge time series from long observed precipitation records or from synthetic precipitation time series obtained with a weather generator (Calver and Lamb, 1995; Cameron et al., 2000; Blazkova and Beven, 2004; Hoes and Nelen, 2005; Winter et al., 2019). The above approach ii) is computationally intensive, especially if long time series are to be simulated using ensembles of hydrological parameter sets, but in exchange, return period analysis is straight forward for simulated peak flows or volumes. Full hydrographs for risk analysis are then obtained either by selecting a range of simulated extreme events 45 or by extrapolating a synthetic design hydrograph (Pramanik et al., 2010; Serinaldi and Grimaldi, 2011).

These fully continuous simulation schemes are particularly useful for studies where recorded discharge time series are too short for extreme flood analysis (Lamb et al., 2016; Evin et al., 2018). Although such an approach is based entirely on a continuous hydrologic simulation, it is noteworthy that such a fully continuous approach might still be considered as being "semi-continuous" from a hydraulic perspective, since corresponding studies often lack the final hydraulic routing step along 50 the floodway (Grimaldi et al., 2013). For clarity, we therefore use the term "continuous hydrologic simulation scheme" to distinguish it from the above mentioned hydraulic approach (see also Appendix A). These continuous hydrologic simulation frameworks are still rare for time series $\geq$ 100 years, particularly because of the high computational power needed for such simulations (Grimaldi et al., 2013). An example is the work of Arnaud and Lavabre (2002), who use a continuous simulation framework to generate an ensemble of possible extreme hydrographs, which are then used as individual scenarios for hazard 55 management. Another option is to summarize all simulated flood hydrographs into probability distributions for peak flow and flood volume (Gabriel-Martin et al., 2019).

For rare events with high return periods typically in use for hydrologic hazard management (e.g. up to 10'000 years), the large number of simulations in fully continuous frameworks can easily become prohibitive, in particular if the framework should also account for different sources of modelling uncertainty, such as input uncertainty (different weather generators) or





the uncertainty of the hydrological model itself, which is often incorporated into the model parameter sets (using distribution of model parameters rather than a single best set) (Cameron et al., 1999). Other important uncertainty sources in hydrological modeling are linked to the used calibration (discharge) data, input (rainfall) data, and model structure (Sikorska and Renard, 2017; Westerberg et al., 2020).

Studies dealing with modelling or data uncertainties in such continuous simulation frameworks are rare, as most previous
studies have focused on the uncertainty related to the hydrological model parameters only (e.g. Blazkova and Beven, 2002, 2004; Cameron et al., 1999). In addition to the uncertainties from seven hydrological model parameters, Arnaud et al. (2017) investigated how the uncertainty related to six rainfall generator parameters propagates through the simulation framework, using more than 1000 French basins with hydrologic observation series of 40 years (median over all basins) and several hundreds of replicats. In their study they found that the uncertainty of the rainfall generator dominates the uncertainty in the simulated
extreme flood quantiles. With the exception of the work of Arnaud et al. (2017) using a simplified hydrologic model, studies that deal with meteorological and hydrological modeling uncertainty in fully continuous simulation frameworks are currently missing. This is despite the fact that recent improvements in computational power with cluster and cloud computing theoretically open up the unlimited possibility of analyzing different combinations of meteorological scenarios and parameter sets of a hydrological model within such ensemble-based simulation frameworks. Yet, computational constraints of hydrological mod-
els, especially at a high temporal resolution (sub-daily or hourly), and data storage, still remain bounding factors for simulation of long time series or for simulation of extreme floods with high return periods (up to 10'000 years).

Accordingly, for settings where full hydrologic-hydraulic models are used for continuous simulation, some pre-selection of hydro-meteorological scenarios is often needed, particularly for computationally demanding complex hydrological or hydraulic models. How this selection should be completed, i.e. based on which quantitative criteria, remains unclear. The meteorological
scenarios have the particularity that all scenarios generated with the same weather generator present different, but equally likely realisations of the assumed climate condition; in other words, they represent the natural variability of the climate. Reducing the number of meteorological input scenarios is not possible without simulating them with a hydrological model first, as long as the continuous simulation scheme is of interest, i.e., if full time series are to be analysed without the possibility of extracting single events. This is due to the non-linear response of any hydrological model to meteorological input (scenario), which translates
into hydrological scenarios with different statistical properties, albeit resulting from an ensemble of input scenarios having the same statistical properties.

We are therefore essentially left with finding ways to reduce at least the computational requirements associated with hydrological model parameter uncertainty, apart from reducing the length of time series, which for analysis of extremes, is an unattractive option. Accordingly, in this work, we propose an assessment of different data-based methods to select a
reduced-size parameter ensemble for the use with a hydrological model within a continuous simulation, ensemble-based hydro-meteorological framework. Our specific research questions are as follows: (1) how can we downsize (reduce) parameter ensembles for simulation of extreme floods so that the variability and the range of the full ensemble is preserved as closely as possible? (2) can such a reduced parameter ensemble be assumed to be reliable for the simulation of extreme floods, during the reference period (used for parameter ensemble downsizing), as well as during an independent validation period? and (3)





which metrics would be suitable to assess the performance of such a reduced parameter ensemble against the reference (full) ensemble? Specifically, three different methods of reducing a full hydrological modelling parameter ensemble to a handful of parameter sets are proposed and tested for deriving the uncertainty ranges of simulated rare flood events (up to 10'000 years return period). All three methods rely on simulation of annual maxima and are tested on continuous synthetic data (simulated with a hydrological model); the aim is thus i) to provide long enough simulation periods for extreme flood analysis, ii) to

avoid the propagation of errors due to data/model calibration etc. and iii) to be able to focus entirely on the uncertainty of the hydrological response. The principal idea underlying these methods is that the downsizing of the parameter ensemble may be performed with a reduced length of input time series that is much shorter than the full simulation time frame and that then can be applied to the full time window for analysis of extremes.

## 2 Methods

### 105 2.1 Study framework and objectives

The focus of this study is a fully continuous hydro-meteorological ensemble-based simulation framework for extreme flood estimation. The underlying streamflow time series ensemble is built based on meteorological scenarios and multiple hydrological model runs using a number of calibrated model parameter sets. A meteorological scenario represents a single realisation from a stochastic weather generator with constant model parameters. These meteorological scenarios are equally likely model

realisations that differ in the precipitation and temperature patterns and together they represent the natural variability of the climate (and not the model uncertainty of a weather generator). These realisations are then used as inputs into a hydrological model to simulate the hydrological response. To account for hydrological modelling uncertainty, a range of different parameter sets is used for each meteorological scenario. These two sources of hydrologic variability then accumulate along the modeling chain and can be represented as an ensemble of possible hydrological responses (Fig. 1).

Within such a defined framework we first want to understand how variable the hydrologic response simulation is, and second, develop methods to downsize the model parameter ensemble to a smaller subset that could be dealt with within such a modeling chain for extreme flood simulations. This subset should represent the entire range of variability of the hydrological response but with little computational effort and should also be transferable to independent time periods. Hereafter, we call this subset the *representative parameter ensemble*.

Downsizing of the parameter ensemble is particularly needed if (i) the probability distribution of the parameter sets is unknown because parameter sets result from independent calibrations or regionalization approaches and only a limited number of sets can be run with the hydrological model, or (ii) the distribution is known but due to time-consuming simulations it is not possible to run the hydrological model for a full ensemble of multiple meteorological scenarios.

  The question of how many parameter sets are needed to cover most of the simulation range is important. However, here

we set this value to a constant number and rather test different selection approaches. Hence, for the purpose of our work, we furthermore would like this *representative parameter ensemble* to be composed of only three sets, which should be representative of a lower (infimum), a middle (median) and a upper (supremum) interval of the full hydrological ensemble (Fig. 1),

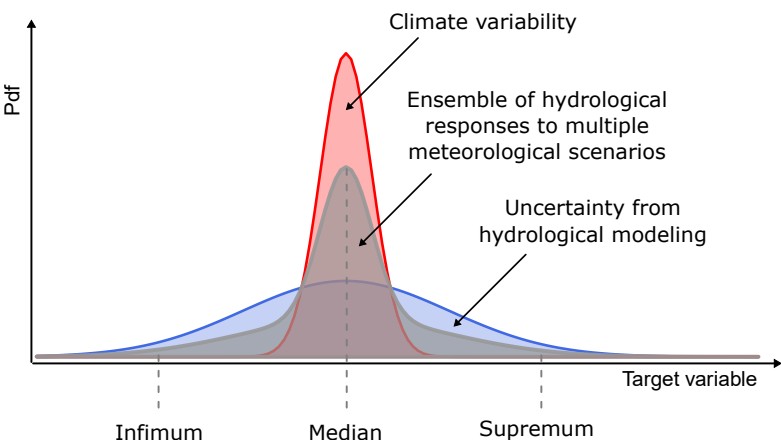

**Figure 1.** Framework overview. The infimum and supremum refer to the the largest interval bounding the ensemble simulation from below and the smallest interval bounding it from above.

and which, together, should enable the construction of predictive intervals for extreme flood estimates that represent the full variability range of all ensemble members. The infimum (from the Latin – smallest) and supremum (from the Latin – largest)

refer to the greatest lower bound and the least upper bound (Hazewinkel, 1994), i.e., the largest interval bounding the ensemble from below and the smallest interval bounding it from above. The choice of infimum and supremum is favourable over the maximum and minimum as the latter would imply a complete parameter ensemble range whereas here we use the terms to describe the range of a certain ensemble.

The key challenge for such a downsizing is the fact that we would like to select parameter sets (i.e. select in the param-

eter space) but based on how representative the corresponding simulations are in the model response space. Moreover, the downsized ensemble should be representative not only for simulated time periods but also be transferable to independent time periods. The first question to answer is which model response space the selection should focus on. In the context of extreme flood estimation, focusing on the frequency distribution of annual maxima (AM) is a natural choice; we thus propose to use the representation of AMs in the Gumbel space as the reference model response space for parameter selection.

The next step is the development of selection methods to select parameter sets that plot into certain locations in the model response space (i.e. in the Gumbel space). Given the nonlinear relationship between model parameters and hydrological responses, this selection has to be obtained via an inverse modelling approach, i.e. we have to first simulate all parameter sets and then decide which parameter sets fulfil certain selection criteria in the model response space.

For that purpose, we developed three methods, which are based on: a) ranking, b) quantiling, and c) clustering, described

in detail in Sect. 2.2. The main idea behind all three methods is that the parameter set selection is made based on the full hydrological simulation ensemble but using only a limited simulation period that is much shorter than the time window of full meteorological scenarios used within the simulation framework.

Next, for the purpose of this study, let us define the following variables:





- $I$ is a number of parameter sets available with $i = 1, 2, ....,$ being a parameter set index;

- $\boldsymbol{\theta}_i$ is the $i$-th parameter set of a hydrological model;

- $J$ is a number of annual maxima (years) per one hydrological simulation, $y = 1, 2, ...,$ is a year of simulation (index of unsorted annual maxima), and $j = 1, 2, ....,$ is an index of sorted annual maxima;

- $X_j$ is the $j$-th sorted annual maximum and $X_y$ is the unsorted annual maximum from the year $y$.

- $M$ is a number of meteorological scenarios considered with $m = 1, 2, ....,$ being a meteorological scenario index;

- $\boldsymbol{S}_m$ is the $m$-th meteorological scenario;

- $H(\boldsymbol{\theta}_i | \boldsymbol{S}_m)$ is the hydrological simulation computed using the $i$-th parameter set of a hydrological model and the $m$-th meteorological scenario;

- $X_{y,i,m}$ is the annual maximum for the year $y$ extracted from $H(\boldsymbol{\theta}_i | \boldsymbol{S}_m)$;

- $\boldsymbol{\theta}_{\mathrm{inf}}$, $\boldsymbol{\theta}_{\mathrm{med}}$ and $\boldsymbol{\theta}_{\mathrm{sup}}$ are the representative parameter sets of the hydrological model, i.e., infimum, median and supremum
that correspond to the intervals named in the same way.

## 2.2 Developed methods for selecting the representative parameter sets

For the sake of simplicity, let us choose a single meteorological scenario $\boldsymbol{S}_m$ for now. Using $\boldsymbol{S}_m$ as an input into a hydrological model combined with $I$ parameter sets results in an ensemble of hydrological simulations, $H(\boldsymbol{\theta}_{1,2,...} | \boldsymbol{S}_m)$. Now, the goal is to select a limited number (here 3) of hydrological model parameter sets, i.e., $\boldsymbol{\theta}_{\mathrm{inf}}$, $\boldsymbol{\theta}_{\mathrm{med}}$ and $\boldsymbol{\theta}_{\mathrm{sup}}$, from the available pool of
$I$ sets ($I \gg 3$) based on the simulation of annual maxima (AM). These AMs are extracted from time series with continuous hydrological simulations, i.e., $H(\boldsymbol{\theta}_{1,2,...} | \boldsymbol{S}_m)$ using a maximum approach that guarantees that the highest peak flow within each calendar year for each hydrological simulation is selected (Fig. 2). This assumption is made to cover the situation when different model realizations (i.e, for $i = 1, 2, ...$) lead to different flood events being classified as the largest event within the year. In this case, we ensure that the largest flood event simulated within each $y$-th year and each $i$-th parameter set is selected.
This means however that AMs selected for the same year $y$ but with a different parameter set may originate from different flood events and even from a different dominant flood process, e.g. heavy rainfall or intensive snowmelt (Merz and Blöschl, 2003). This could be the case when one parameter set better represents processes driven by the rainfall excess while others by the snowmelt dynamics. For simplicity, we do not distinguish events by their different flood genesis and pool all AMs together.

Using the above notations, the selection of representative parameter sets can be summarized as:

1. Simulation of continuous streamflow times series: the hydrological model is run with all available $I$ parameter sets over the simulation period. This gives $I$ different hydrological realizations (simulation ensemble members) covering the same time span.




**Figure 2.** Overview of the modeling chain and the selection methods of the representative parameter sets; top panel: delivery of hydrological simulation ensembles and ensemble ranges; bottom panel: three methods (A-C) proposed for selecting the representative parameter sets based on annual maxima (AM) marked with red circles.

2. Selection of annual maxima (AMs): for each $i$-th hydrological realisation annual maxima are selected as the highest peak flow within each $y$-th simulation year. This results in a $J$ set of AMs per each $i$ hydrological simulation. The selection of AMs is repeated for all $I$ hydrological simulations.

3. Selection of three representative parameter sets based on the simulation of AMs and following on from the three methods detailed below.





### 2.2.1 Ranking

(a) AMs computed from $I$ hydrological simulations (i.e., using $I$ model parameter sets) are sorted by their magnitude from
the highest to the lowest within each $y$-th simulation year independently (Fig. 2A).

(b) For each $y$-th simulation year, AMs which correspond to the 5th, 50th and 95th rank for that year are selected.

(c) Parameter sets that correspond to the selected AM ranks are then attributed as 5th, 50th and 95th parameter sets per each
    $y$-th year independently.

(d) The parameter sets selected in step (d) are compared over all $J$ simulation years and the sets which are chosen most
often as the 5th, 50th and 95th ranks are retained as the parameter sets $\boldsymbol{\theta}_{R5}$, $\boldsymbol{\theta}_{R50}$ and $\boldsymbol{\theta}_{R95}$ representative for the entire
    simulation period and for the entire hydrological simulation ensemble.

### 2.2.2 Quantiling

(a) For each $i$-th parameter set, AMs computed with this parameter set are sorted by their magnitude over the entire simu-
    lation period ($J$ years) and plotted in the Gumbel space (Generalized Extreme Value distribution Type-I), thus creating
the ensemble of sorted AMs simulated with different parameter sets.

(b) The 5%, 50% and 95% quantiles of these ensembles are computed as $\boldsymbol{Q}_5$, $\boldsymbol{Q}_{50}$ and $\boldsymbol{Q}_{95}$ over the entire simulation period
    (Fig. 2B).

(c) Next, for each $i$-th ensemble member, a metric $R_{\mathrm{MSE}}$ is computed such that for each $j$-th point of the $i$-th ensemble
    member measures distances in Gumbel space from $\boldsymbol{Q}_5$, $\boldsymbol{Q}_{50}$ and $\boldsymbol{Q}_{95}$. This metric is somehow similar to the mean
square error and is computed as:

$$R_{\mathrm{MSE},Q5,i} = \frac{1}{J}\sum_{j=1}^{J}\left(\boldsymbol{Q}_{5,j} - H_j(\boldsymbol{\theta}_i|\boldsymbol{S}_m)\right)^2 \tag{1}$$

$$R_{\mathrm{MSE},Q50,i} = \frac{1}{J}\sum_{j=1}^{J}\left(\boldsymbol{Q}_{50,j} - H_j(\boldsymbol{\theta}_i|\boldsymbol{S}_m)\right)^2 \tag{2}$$

$$R_{\mathrm{MSE},Q95,i} = \frac{1}{J}\sum_{j=1}^{J}\left(\boldsymbol{Q}_{95,j} - H_j(\boldsymbol{\theta}_i)|\boldsymbol{S}_m\right)^2 \tag{3}$$

(d) Finally, the ensemble members which in the Gumbel space lie closest to $\boldsymbol{Q}_5$, $\boldsymbol{Q}_{50}$ and $\boldsymbol{Q}_{95}$, i.e., received the smallest
    values for $R_{\mathrm{MSE},Q5}$, $R_{\mathrm{MSE},Q50}$ and $R_{\mathrm{MSE},Q95}$, respectively, are chosen as the ensemble members representative for the
    entire hydrological ensemble, and the parameter sets corresponding to these members, i.e., $\boldsymbol{\theta}_{Q5}$, $\boldsymbol{\theta}_{Q50}$ and $\boldsymbol{\theta}_{Q95}$, are
    retained as representative.




### 2.2.3 Clustering

(a) Similar to the quantiling method, for each $i$-th parameter set AMs computed with this parameter set are sorted by their magnitude over the entire simulation period and plotted in the Gumbel space, creating $I$ ensemble members of sorted AMs simulated with different parameter sets.

(b) These members are next clustered in the Gumbel space into three representative groups (clusters) based on all $J$ simulation years using the k-means clustering with Hartigan–Wong algorithm (Hartigan and Wong, 1979), as implemented in the function kmeans from the package "stats" (R Core Team, 2019), see Fig. 2C.

(c) Next, these clusters are sorted by their magnitude and for the lower cluster a 5th percentile, for the upper – 95th percentile, and for the middle – 50th percentile are computed, i.e., $\boldsymbol{P}_5$, $\boldsymbol{P}_{50}$ and $\boldsymbol{P}_{95}$. Note that we use here percentiles instead of cluster means to make this method comparable with the other two methods and to better cover the variability of the parameter sample.

(d) For each $i$-th ensemble member, the metric $R_{\mathrm{MSE}}$ is computed in relation to three estimated cluster percentiles $\boldsymbol{P}_5$, $\boldsymbol{P}_{50}$ and $\boldsymbol{P}_{95}$ as:

$$R_{\mathrm{MSE},P5,i} = \frac{1}{J}\sum_{j=1}^{J}\left(\boldsymbol{P}_{5,j} - H_j(\boldsymbol{\theta}_i|\boldsymbol{S}_m)\right)^2 \tag{4}$$

$$R_{\mathrm{MSE},P50,i} = \frac{1}{J}\sum_{j=1}^{J}\left(\boldsymbol{P}_{50,j} - H_j(\boldsymbol{\theta}_i|\boldsymbol{S}_m)\right)^2 \tag{5}$$

$$R_{\mathrm{MSE},P95,i} = \frac{1}{J}\sum_{j=1}^{J}\left(\boldsymbol{P}_{95,j} - H_j(\boldsymbol{\theta}_i|\boldsymbol{S}_m)\right)^2 \tag{6}$$

(e) For each of these three clusters, the ensemble member that lies closest to the cluster percentile, i.e., received the smallest value of $R_{\mathrm{MSE}}$, is selected as the representative member for that cluster and the parameter sets which correspond to these members, $\boldsymbol{\theta}_{P5}$, $\boldsymbol{\theta}_{P50}$ and $\boldsymbol{\theta}_{P95}$, are retained as representative.

For plotting, we used the Gringorten's method (Gringorten, 1963) to compute the plotting positions of AMs in the Gumbel plots:

$$k_j = \frac{j - 0.44}{J + 0.12} \tag{7}$$

where $k_j$ is a plotting position for the $j$-th (sorted) AM in the Gumbel space.





**Table 1.** Comparison of three methods for selecting representative parameter sets based on annual maxima (AM).

| Criteria | Ranking | Quantiling | Clustering |
|---|---|---|---|
| Selection window | year | all simulation years | all simulation years |
| Annual maxima (AM) | unsorted over years | sorted over years | sorted over years |
| Sorting space | simulated annual maxima | Gumbel space, quantiling | Gumbel space, clustering |
| Sorting extent | AMs over simulations | AMs over years | AMs over years |
| Selection criteria | ranks | $R_{MSE}$ | $R_{MSE}$ |
| Interpretation of pred. intervals | no | yes | yes |
| Parameter grouping | no | no | yes |

## 2.3 Estimation of the predictive intervals for extreme flood simulations

While the three methods described in Sect. 2.2 vary in the way the representative parameter sets are selected (see Sect. 2.4 for a summary), each of these selection methods results in three (different) representative hydrological simulation ensemble members and can be thought of as representing the lower (infimum), upper (median) and middle (supremum) interval of the full simulation range. The parameter sets corresponding to these are then noted as $\theta_{inf}$, $\theta_{med}$, $\theta_{sup}$. The simulations corresponding to these three parameter sets together create the so-called predictive interval, which can be used for extreme flood simulation studies.

## 2.4 Comparison of three selection methods

The major difference between these three methods is that the ranking method is evaluated based on individual simulation years using simple ranking of flow maxima independently of their frequency, i.e., it works on unsorted annual maxima. Note that in this way, for each $y$ simulation year, a different rank order of the $I$ parameter sets may be achieved. In an extreme case, where for each year $y$ different parameter sets are chosen, a choice of the representative sets over all simulation years may become problematic due to difficulties in identifying the parameter sets most frequently selected over all simulation years.

In contrast to the ranking method, both other methods, i.e., quantiling and clustering, are performed on sorted AMs over all simulation years, i.e., in the flow frequency space. This enables statistical statements to be made about the selected parameter sets and about the predictive intervals constructed with the help of these parameter sets. Furthermore, selected parameter sets can be assumed to be representative over the entire simulation period (see Table 1 for a detailed overview of three methods).

## 2.5 Assessment of the behavior of the approach

Testing the methods for a time period different than the one that was used for the parameter ensemble downsizing is crucial for assessing how well the reduced ensembles cover the reference simulation ensemble. Thus, we propose to assess the behavior of the developed approach by repeating the selection of the three representative parameter sets with the three proposed methods





with multiple ($M$) meteorological scenarios. Using multiple meteorological scenarios enables, first, accounting for the natural variability of the hydrological response due to climate variability, and second, gives us the possibility to evaluate the bias of the approach. Particularly, with the help of multiple meteorological scenarios we explore how the choice of the representative parameter sets $\boldsymbol{\theta}_{\text{inf}}$, $\boldsymbol{\theta}_{\text{med}}$, $\boldsymbol{\theta}_{\text{sup}}$ depends on the meteorological scenario.

### 2.5.1 Leave-one-out cross-validation

To evaluate the three selection methods, we perform a leave-one-out cross-validation simulation study, in which a meteorological scenario $S_r$ is removed from the analysis and the selection of the representative parameter sets is executed based on all other remaining meteorological scenarios, i.e., using all $m = 1, 2, ..., M$ and $m \neq r$. The evaluation of selection methods is then executed against the one meteorological scenario initially removed from the set. In detail, the following steps are executed for each of the three methods independently:

(a) Pick-up and remove one meteorological scenario $\boldsymbol{S}_r$ from $\boldsymbol{S}_{1,2,..,M}$ scenarios available;

(b) Analyze all other meteorological scenarios $\{\boldsymbol{S}_{M-r}\} = \{\boldsymbol{S}_{1,2,..,M}\} \setminus \{\boldsymbol{S}_x\}$ each containing $I$ ensemble members resulting from $I$ parameter sets, $\{H(\boldsymbol{\theta}_i|\boldsymbol{S}_{m-r})\}$, for $i = 1, 2, ..., I$, $m = 1, 2, ..., M$ and $m \neq r$, and based on the selected three representative parameter sets $\boldsymbol{\theta}_{\text{inf},m-r}$, $\boldsymbol{\theta}_{\text{med},m-r}$, $\boldsymbol{\theta}_{\text{sup},m-r}$ as described in sect. 2.2;

(c) Estimate the predictive intervals of these $\boldsymbol{S}_{M-r}$ meteorological scenarios as the band spread between $H(\boldsymbol{\theta}_{\text{inf},m-r}|\boldsymbol{S}_{m-r})$ and $H(\boldsymbol{\theta}_{\text{sup},m-r}|\boldsymbol{S}_{m-r})$, the interval defined in step (b);

(d) Evaluate the meteorological scenario $\boldsymbol{S}_r$ removed at step (a) against the predictive intervals of $\boldsymbol{S}_{M-r}$ meteorological scenarios to assess how well the defined identified intervals represent the ensemble members of this $\boldsymbol{S}_r$ meteorological scenario.

The simulation is repeated $M$ times to use each meteorological scenario once.

### 2.5.2 Multi-scenario evaluation

To further evaluate the three methods, we perform a simulation study using multiple ($M$) meteorological scenarios. In this study, the three selection methods are executed on one $x$ meteorological scenario randomly (without replacement) selected from the $M$ available scenarios and evaluated against all remaining scenarios. In detail, the following steps are executed for each of the three methods independently:

(a) Pick-up one meteorological scenario $\boldsymbol{S}_p$ out of the $\boldsymbol{S}_{1,2,..,M}$ scenarios available;

(b) Analyze the $I$ simulated hydrological ensemble members of this scenario $H(\boldsymbol{\theta}_i|\boldsymbol{S}_p)$, $i = 1, 2, ..., I$, resulting from $I$ parameter sets $\boldsymbol{\theta}_i$ for $\boldsymbol{S}_p$ and select three representative parameter sets corresponding to $\boldsymbol{\theta}_{\text{inf},p}$, $\boldsymbol{\theta}_{\text{med},p}$, $\boldsymbol{\theta}_{\text{sup},p}$, as described in sect. 2.2;



(c) For all other remaining meteorological scenarios $\{\boldsymbol{S}_{M-p}\} = \{\boldsymbol{S}_{1,2,..,M}\} \setminus \{\boldsymbol{S}_p\}$, take all hydrological ensemble members $\{H(\boldsymbol{\theta}_i|\boldsymbol{S}_m)\}$, for $m = 1, 2, ..., M$ and $m \neq p$, that correspond to $\boldsymbol{\theta}_{\text{inf},p}, \boldsymbol{\theta}_{\text{med},p}, \boldsymbol{\theta}_{\text{sup},p}$. This results in $M-1$ model simulations for each of $\boldsymbol{\theta}_{\text{inf},p}, \boldsymbol{\theta}_{\text{med},p}, \boldsymbol{\theta}_{\text{sup},p}$ one per meteorological scenario;

    (d) Compute the 5th percentile for $\{H(\boldsymbol{\theta}_{\text{inf},p}|\boldsymbol{S}_m)\}$, the 50th for $\{H(\boldsymbol{\theta}_{\text{med},p}|\boldsymbol{S}_m)\}$, and the 95th for $\{H(\boldsymbol{\theta}_{\text{sup},p}|\boldsymbol{S}_m)\}$. The computed 5th and 95th percentiles are assumed to together describe the predictive intervals;

(e) Evaluate the predictive intervals against all $\boldsymbol{S}_{M-p}$ meteorological scenarios for assessing how well the identified prediction intervals represent the ensemble members of these $\boldsymbol{S}_{M-p}$ scenarios.

The steps (a-e) are repeated $M$ times to use each meteorological scenario once. We call this evaluation a multi-scenario evaluation because the evaluation is performed using multiple meteorological scenarios at once ($\boldsymbol{S}_{M-p}$) in contrast to the leave-one-out cross-validation (Sect. 2.5.1), where the evaluation is performed against only one meteorological scenario ($\boldsymbol{S}_r$).

## 295  2.6   Evaluation criteria

### 2.6.1   Visual assessment

The simplest way of assessing the behavior of these three methods is a visual inspection of curves plotted in the Gumbel space, which can tell us how well the selected members reproduce the simulation ensemble and particularly whether the assignment of the representative parameter sets is correct or not. For this purpose, we propose to plot all simulated hydrological ensemble 300 members together with the selected representative members in the Gumbel space for each considered meteorological scenario $m$ individually and to visually assess the assignment of the three selected parameter sets, $\boldsymbol{\theta}_{\text{inf},m}, \boldsymbol{\theta}_{\text{med},m}, \boldsymbol{\theta}_{\text{sup},m}$, and the corresponding intervals, i.e., $H(\boldsymbol{\theta}_{\text{inf},m}|\boldsymbol{S}_m)$, $H(\boldsymbol{\theta}_{\text{med},m}|\boldsymbol{S}_m)$ and $H(\boldsymbol{\theta}_{\text{sup},m}|\boldsymbol{S}_m)$. The order of the intervals' assignment is assumed to be correct if it holds in the Gumbel space that:

$$H(\boldsymbol{\theta}_{\text{inf},m}|\boldsymbol{S}_m) \leq H(\boldsymbol{\theta}_{\text{med},m}|\boldsymbol{S}_m) \leq H(\boldsymbol{\theta}_{\text{sup},m}|\boldsymbol{S}_m). \tag{8}$$

We further define a ratio of incorrectly attributed scenarios, with mixed-up intervals, i.e. for which Eq. 8 does not hold, as a measure of the bias as:

$$R_{bias} = \sum_{m=1}^{M} \frac{R_m}{M} \tag{9}$$

where $R_m$ is computed for the $m$-th scenario as:

$$R_m = \begin{cases} 0 & \text{if } H(\boldsymbol{\theta}_{\text{inf},m}|\boldsymbol{S}_m) < H(\boldsymbol{\theta}_{\text{med},m}|\boldsymbol{S}_m) < H(\boldsymbol{\theta}_{\text{sup},m}|\boldsymbol{S}_m) \\ 1 & \text{else.} \end{cases} \tag{10}$$

### 310  2.6.2   Quantitative assessment

To quantitatively compare the three selection methods, we propose to compute the three following metrics:





(I) The ratio of simulation points in the Gumbel space, i.e. annual maxima, lying outside the predictive intervals computed for each $m$-th scenario as:

$$R_{\text{spo},m} = \sum_{i=1}^{I} \sum_{j=1}^{J} \frac{R_{\text{spo},m,i,j}}{I \cdot J} \tag{11}$$

where $R_{\text{spo},m,i}$ is the ratio for each $i$-th parameter set of the meteorological scenario $m$ and is computed for each simulation point $j$ (in the Gumbel space) as:

$$R_{\text{spo},m,i,j} = \begin{cases} 0 & \text{if } H_j(\boldsymbol{\theta}_{\text{inf},m}|\boldsymbol{S}_m) \leq H_j(\boldsymbol{\theta}_i|\boldsymbol{S}_m) \leq H_j(\boldsymbol{\theta}_{\text{sup},m}|\boldsymbol{S}_m) \\ 1 & \text{else.} \end{cases} \tag{12}$$

(II) In the leave-one-out cross-validation, the ratio of hydrological simulation ensemble members lying outside the predictive intervals is computed for each $m$-th scenario as:

$$R_{\text{hso},m} = \sum_{i=1}^{I} \frac{R_{\text{hso},m,i}}{I} \tag{13}$$

where $R_{\text{hso},m,i}$ is the ratio computed for each $i$-th ensemble member as:

$$R_{\text{hso},m,i} = \begin{cases} 0 & \text{if } H(\boldsymbol{\theta}_{\text{inf},m}|\boldsymbol{S}_m) \leq H(\boldsymbol{\theta}_i|\boldsymbol{S}_m) \leq H(\boldsymbol{\theta}_{\text{sup},m}|\boldsymbol{S}_m) \\ 1 & \text{else.} \end{cases} \tag{14}$$

(III) In the multi-scenario evaluation, the ratio of meteorological scenarios lying outside the predictive intervals is computed for each scenario $p$ as:

$$R_{\text{mso},p} = \sum_{m}^{M} \frac{R_{\text{mso},m}}{M-1} \quad \text{m=1,2,...,M} \ \& \ m \neq p \tag{15}$$

where $R_{\text{mso},m}$ is computed as:

$$R_{\text{mso},m} = \begin{cases} 0 & \text{if } H(\boldsymbol{\theta}_{\text{inf},m}|\boldsymbol{S}_m) \leq H(\boldsymbol{\theta}_i|\boldsymbol{S}_m) \leq H(\boldsymbol{\theta}_{\text{sup},m}|\boldsymbol{S}_m) \forall i = 1,2,..,I \\ 1 & \text{else.} \end{cases} \tag{16}$$

With respect to $R_{\text{spo}}$, the question arises of how to define the ratio of simulation points being outside the predictive intervals if multiple hydrological simulations (leave-one-out cross-validation) or multiple meteorological scenarios (multi-scenario evaluation) are considered. Here we propose to use different percentiles, i.e., the 5th, 50th, and 95th percentiles, to characterize the ratio of the simulation points lying outside the computed predictive intervals for each of the methods.

In a similar way, for $R_{\text{hso}}$ and $R_{\text{mso}}$ an additional condition must be defined, i.e., how many out of $J$ hydrological simulation points for $R_{\text{hso}}$, or how many out of $I$ hydrological simulation ensemble members for $R_{\text{mso}}$ must lie outside the defined





predictive intervals, so that the hydrological simulation $H(\boldsymbol{\theta}_i|\boldsymbol{S}_m)$, or the meteorological scenario $\boldsymbol{S}_m$, is considered as lying

outside these intervals.

For this purpose we define the rejection threshold $r_{\mathrm{thr}}$ (dimensionless) that has to be reached, so that the meteorological scenario or hydrological simulation is assumed as lying outside the predictive intervals. In this work, we consider the following values for $r_{\mathrm{thr}} = \{0.50, 0.25, 0.10, 0.05\}$.

These three metrics are computed for all three methods and for all $M$ meteorological scenarios, and the median values over

these $M$ scenarios are taken as a measure for comparing the three methods.

## 3 Experimental set-up

### 3.1 Study catchment

For testing the methods developed here, a small natural catchment is preferable. For this purpose, the Dünnern at Olten catchment with an area of 196 km$^2$ is selected, located in the Jura region in Switzerland (Fig. A2 in Appendix B). The Dünnern

stream is a tributary of the Aare River and belongs to the basin of the Rhine River. The mean elevation of the Dünnern at Olten catchment is 711 m. a.s.l. with an elevation span from 400 to 760 m. a.s.l.. The flow regime is defined as nival pluvial jurassien (Weingartner and Aschwanden, 1992; Schürch et al., 2010) with high flows in winter and spring and low flows in autumn. With no direct human influence within the entire catchment known, it can be assumed close to natural (BAFU, 2017). This catchment is part of a large-scale extreme flood modelling effort in Switzerland for the entire Aare catchment (Viviroli et al.,

350   2020).

### 3.2 HBV model for hydrological simulations and calibration data

To simulate the hydrological catchment responses to meteorological scenarios, the HBV model is used. The HBV model is a semi-distributed bucket-type model and it consists of four main routines: (1) precipitation excess, snow accumulation and snowmelt, (2) soil moisture, (3) groundwater and streamflow responses and (4) runoff routing using a triangular weighting

function. Due to the presence of the snow component, the HBV model is applicable to mountainous catchments (e.g., Jost et al., 2012; Addor et al., 2014; Breinl, 2016; Griessinger et al., 2016; Sikorska and Seibert, 2018; Brunner and Sikorska-Senoner, 2019).

In this study, the version HBV light (Seibert, 1997; Seibert and Vis, 2012) with 15 calibrated parameters is used; see Table A1 in Appendix C for details on model parameters and their calibration ranges. Model inputs are time series of precipitation and

air temperature, and long-term averages of seasonally varying estimates of potential evaporation, all being area-average values for the entire catchment. These inputs are next redistributed along pre-defined elevation bands using two different constant altitude-dependent correction factors for precipitation and temperature. The model output is streamflow at the catchment outlet at time steps identical to input data (hourly in this study).



For the study catchment, meteorological inputs (hourly precipitation totals, hourly air temperature means, average hourly
evaporation sums) for the HBV model are derived from observed records from meterological stations and are averaged to the
mean catchment values using the Thiessen polygon method. The recorded continuous hourly streamflow data at the catchment
outlet (Olten station) covers the period 1990-2014.

### 3.3 Identification of multiple HBV parameter sets

To calibrate the HBV model described in Sect. 3.2, we used a multi-objective function $F_{\mathrm{obj}}$ with three scores: the Kling-Gupta
efficiency ($R_{\mathrm{KGE}}$) and the efficiency for peak flows ($R_{\mathrm{PEAK}}$), which are both sensitive to peak flows, and a measure based on
the Mean Absolute Relative Error ($R_{\mathrm{MARE}}$) that is sensitive to low flows. $F_{\mathrm{obj}}$ is obtained through weighing these metrics as
follows:

$$F_{\mathrm{obj}} = 0.3 R_{\mathrm{KGE}} + 0.5 R_{\mathrm{PEAK}} + 0.2 R_{\mathrm{MARE}} \tag{17}$$

For details on $R_{\mathrm{KGE}}$, see the work of Gupta et al. (2009); for details on $R_{\mathrm{PEAK}}$ and $R_{\mathrm{MARE}}$, see the work of Vis et al. (2015).
The weights in $F_{\mathrm{obj}}$ are chosen following our previous experience in modelling Swiss catchments (Sikorska et al., 2018;
Westerberg et al., 2020). This objective function $F_{\mathrm{obj}}$ is next used together with the Genetic Algorithm approach (Appendix C)
to calibrate the HBV model with hourly data. The available observational datasets are split into a calibration (1990-2005 years)
and a validation (2006-2014 years) period. To set up the initial conditions, one year of model simulations are discarded from
the calibration simulation and the remaining used for model performance computation. For the validation period, the initial
conditions are taken from the calibration simulation.

The calibration is repeated 100 times resulting in 100 independent optimal parameter sets (see Appendix C). The median
model efficiency measured with $F_{\mathrm{obj}}$ over all 100 runs is 0.7 in the calibration and in the validation periods, which can be
assumed to be a good model performance on an hourly scale.

### 3.4 Generation of synthetic meteorological scenarios using a weather generator

Meteorological scenarios of synthetic precipitation and temperature data for the Dünnern at Olten catchment are generated
with the weather generator model GWEX developed by Evin et al. (2018) and referred to in their paper as GWEX_Disag. This
stochastic model is a multi-site precipitation and temperature model that reproduces the statistical behavior of weather events
on different temporal and spatial scales. The major property of GWEX is that it uses marginal heavy-tailed distributions for
generating extreme precipitation and temperature conditions. Moreover, it has been developed to generate long term ($\approx 1000$
years) meteorological scenarios. GWEX_Disag generates precipitation amounts at a 3-day scale and then disaggregates them
to a daily scale using a method of fragments (for details on the precipitation model, see the work of Evin et al. (2018), and for
details on the temperature model, see the work of Evin et al. (2019)).

The meteorological scenarios used in this study are a subset from the long-term meteorological scenarios developed for the
entire Aare River basin using recorded data from 105 precipitation stations and from 26 temperature stations in Switzerland
(Evin et al., 2018, 2019). For this larger scale research project, GWEX_Disag was set up using daily precipitation and tem-




perature data from the period 1930-2015 and hourly records of precipitation and temperature from 1990-2015 for the Aare River basin. The daily values generated with GWEX_Disag were then disaggregated to hourly values using the meteorological analogues method, which for each day in the simulated dataset finds an analogue day in observed data, i.e., with a known hourly time structure. Next, catchment means were computed using the Thiessen polygon method.

For the present study, 100 different meteorological scenarios (precipitation and temperature) covering the same time frame of 100 years at an hourly time step are available for the Dünnern at Olten catchment. The simulated data is assumed to be representative for current climate conditions, i.e., no variation due to climate or land use change, or river modification is considered. Thus, differences between scenarios are exclusively due to the natural variability of the meterological time series.

### 3.5    Generation of synthetic hydrological simulation ensembles

Finally, for our analysis, 100 meteorological scenarios with continuous data of 100 years of precipitation and temperature, and 100 calibrated parameter sets of the HBV model are available. These 100 meteorological scenarios are used as input into the HBV model to generate streamflow time series with 100 different HBV parameter sets. To set up the initial conditions of the model, a one year warming up period is always used prior to the simulation period.

From each of these continuous hydrological simulations, 100 Annual Maxima (AM, one per each calendar year) are selected
(see Fig. 3). This results in the following analysis set up:

- $I = 100$ and $i = 1, 2, ...., 100$;

- $J = 100$, $y = 1, 2, ...., 100$ and $j = 1, 2, ...., 100$;

- $M = 100$ and $m = 1, 2, ...., 100$.

with $100 \times 100 \times 100$ combinations of the annual maximum $\times$ parameter set $\times$ meteorological scenario.
These series of AMs are next used to test the developed methods of selecting the representative parameter sets from the ensemble of 100 available sets.

## 4    Results

### 4.1    Representative parameter sets

The representative parameter sets selected with each of the three methods are summarized over all 100 meteorological scenarios
in the form of violin plots (Fig. 4). We present parameter sets by their unique indices that are kept the same for all three methods in the entire analysis. Note that this indexing of the model parameter sets is used for illustration purposes only and does not contain any quantitative information on the model performance. The focus is here on parameter set indices rather than on their actual values since the proposed method selects entire parameter vectors and not individual parameter values. Although the choice of the representative parameter sets depends on the meteorological scenario and on the selection method, certain
patterns can be detected in the selected parameter sets for all three methods. Namely, some sets are more often chosen than





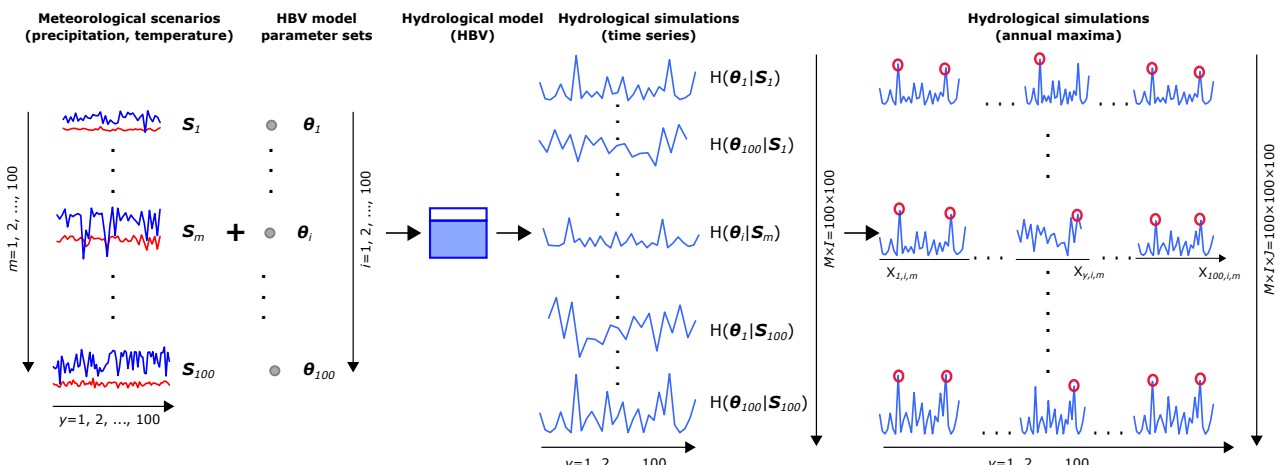

**Figure 3.** Setup of the experimental study.

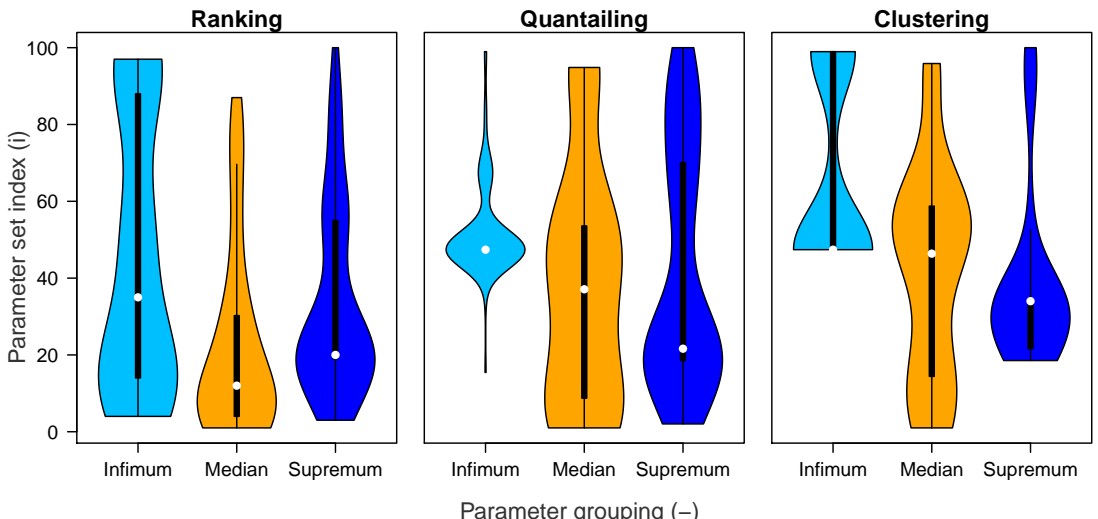

**Figure 4.** Violin plots showing indices of the parameter sets most often selected as the representative parameter sets over 100 meteorological scenarios chosen with three methods. Indexing of the model parameter sets is created as $i =$ 1, 2, ..., 100 and is kept the same for all three methods and is presented here for illustration purposes only.

others as a representative set for different meteorological scenarios. This grouping is particularly visible for the supremum set in all three methods but is strongest for the clustering method; for the infimum set, the grouping effect is the most pronounced for the clustering and the quantiling method, and for the median set in the ranking method.

The five most frequently chosen parameter sets for each method are summarized in Table 2. Although different parameter sets are usually selected by different methods, in a few cases the same set is chosen with more than one selection method





**Table 2.** The three representative parameter sets $\theta_{\text{inf}}$, $\theta_{\text{med}}$ and $\theta_{\text{sup}}$ most frequently selected with three methods. $i$ stands for the set index and ct. for the number of counts. The expressions $\sum$ ct. 1st-3rd and $\sum$ ct. 1st-5th stand for the sum of counts for the first three and for the first five most frequently selected sets. Bold font indicates parameter set indices which are selected as representative with at least two methods among the five sets most frequently chosen.

| Method | Ranking | | | | | | Quantiling | | | | | | Clustering | | | | | |
|---|---|---|---|---|---|---|---|---|---|---|---|---|---|---|---|---|---|---|
| Repr. set | $\theta_{\text{inf}}$ | | $\theta_{\text{med}}$ | | $\theta_{\text{sup}}$ | | $\theta_{\text{inf}}$ | | $\theta_{\text{med}}$ | | $\theta_{\text{sup}}$ | | $\theta_{\text{inf}}$ | | $\theta_{\text{med}}$ | | $\theta_{\text{sup}}$ | |
| Par. set | $i$ | ct. | $i$ | ct. | $i$ | ct. | $i$ | ct. | $i$ | ct. | $i$ | ct. | $i$ | ct. | $i$ | ct. | $i$ | ct. |
| 1st | **97** | 21 | 1 | 11 | 20 | 25 | **47** | 78 | **2** | 22 | **19** | 32 | **47** | 62 | **2** | 13 | 34 | 48 |
| 2nd | 16 | 15 | **2** | 7 | **19** | 13 | **66** | 10 | 93 | 11 | **86** | 15 | **97** | 35 | **46** | 11 | **22** | 33 |
| 3rd | 6 | 12 | 14 | 7 | **57** | 9 | 67 | 4 | **46** | 9 | 69 | 11 | **66** | 2 | 62 | 11 | 98 | 7 |
| 4rd | 54 | 10 | 3 | 6 | 7 | 7 | 82 | 2 | 18 | 7 | **22** | 10 | 73 | 1 | **53** | 10 | **86** | 6 |
| 5th | 14 | 8 | 8 | 5 | 55 | 6 | **97** | 2 | **53** | 7 | **57** | 8 | – | – | 51 | 9 | 50 | 3 |
| $\sum$ ct. 1st-3rd | 48 | | 25 | | 47 | | 92 | | 42 | | 58 | | 99 | | 35 | | 88 | |
| $\sum$ ct. 1st-5th | 66 | | 36 | | 60 | | 96 | | 56 | | 76 | | 100 | | 54 | | 97 | |

(highlighted in Table 2). Among the first five most frequently chosen sets, the same parameter set is selected as the median set once for all three methods and twice for at least two methods. For the supremum set, among the first five most frequently chosen sets, the same set is selected four times at least for two methods but never for all three methods. For the infimum set, only one set is chosen for two methods among the first five most frequently chosen sets. Interestingly, for the supremum set in the clustering method, only four parameter sets among all 100 available are chosen over all 100 scenarios.

### 4.2 Infimum, median and supremum intervals

Using the selected representative sets, representative intervals for extreme flood predictions are constructed for each of the 100 meteorological scenarios and each of the three selection methods. Examples of these intervals for two meteorological scenarios are presented in Figs. 5–6. Note that apart from selecting representative intervals, the clustering method leads to grouping all ensemble members into three selected clusters.

According to a first visual assessment, these three methods lead to slightly different constructed frequency intervals particularly in the upper tail of the distribution, i.e., for the most rare (highest) flows, which are of highest interest. Moreover, the ranking method leads to less symmetrically spread intervals, with the median and infimum intervals lying close to each other. The other two methods lead to more symmetrically spread intervals.

For the quantitative assessment, the ratio of scenarios incorrectly attributed, i.e. with intervals being mixed-up, $(R_{bias})$ varies between the three methods and is the lowest for the ranking method ($R_{bias}$ =0.54). For the clustering method, the three intervals are always correctly attributed for all 100 meteorological scenarios tested ($R_{bias}$ =0.0). For the quantiling method, this ratio is equal to $R_{bias}$ =0.02 and thus also very low. Hence, we can conclude that both clustering and quantiling methods


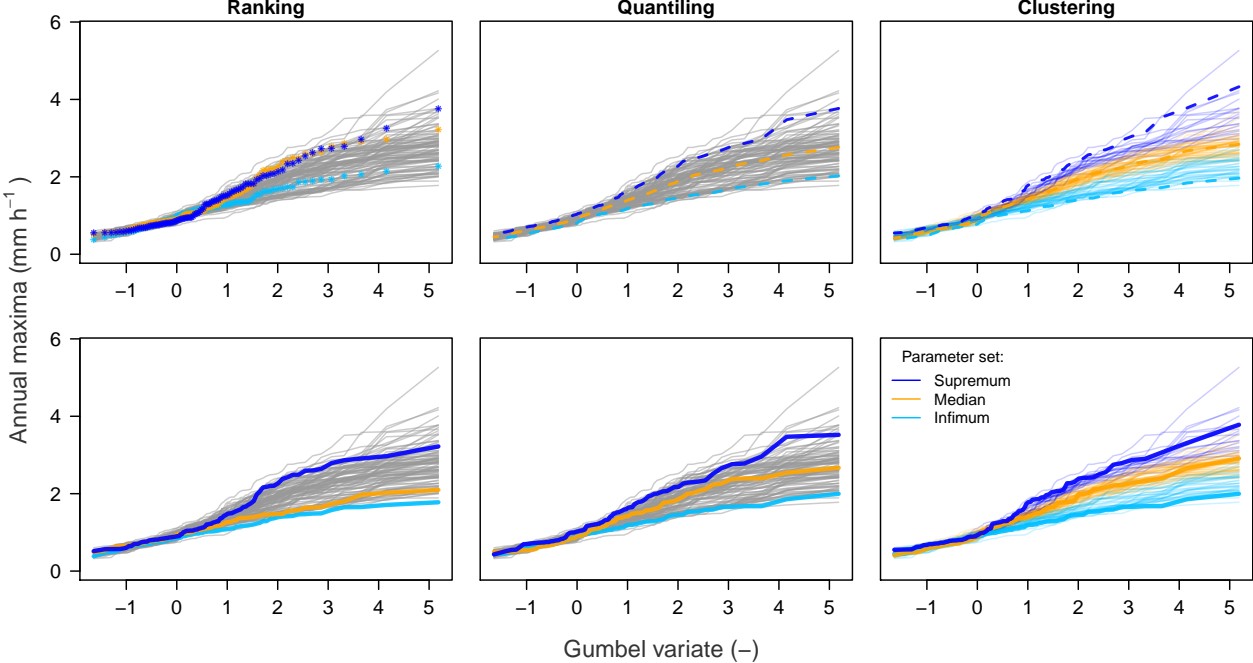

**Figure 5.** Example of the representative parameter sets' selection with three methods in the Dünnern at Olten catchment (meteorological scenario $m =14$). The top panel presents intermediate steps of selecting the representative sets and the bottom panel the finally constructed intervals, i.e., infimum, median and supremum. The dashed lines (top panel) indicate the computed representative intervals (i.e., steps (a-c) in ranking and clustering, and (a-b) in quantiling) and the solid lines (bottom panel) indicate the hydrological simulation members corresponding to the parameter sets selected as representative (step (d) in ranking and quantiling, and (e) in clustering). The Gumbel variates of -1, 0 and 3 correspond to events with 1-year, 2-year and 21-year return periods, whereas the Gumbel variate of 5 corresponds to the event with the 149-year return period (the event with the 100-year return period would correspond to the Gumbel variate of 4.6).

provide correctly attributed intervals with a bias $\leq 2\%$. For the ranking method, the correctness of the interval attribution is
poor, and in more than 50% of the meteorological scenarios, the simulations corresponding to the selected parameter sets lead to mixed-up frequency intervals.

### 4.3 Evaluation of the three selection methods

The behavior of the three selection methods is further evaluated with the 100 meteorological scenarios using the leave-one-out cross-validation test (Sect. 2.5.1) and the multi-scenario evaluation method (Sect. 2.5.2) and corresponding metrics
(Sect. 2.6.2). Examples for two meteorological scenarios are presented in Figs. 7–8 for the leave-one-out cross-validation test and in Figs. 9–10 for the multi-scenario evaluation. From the visual assessment, it is difficult to judge the methods, as they seem to perform similarly well. However, the range of the predictive intervals obtained with 99 meteorological scenarios (one

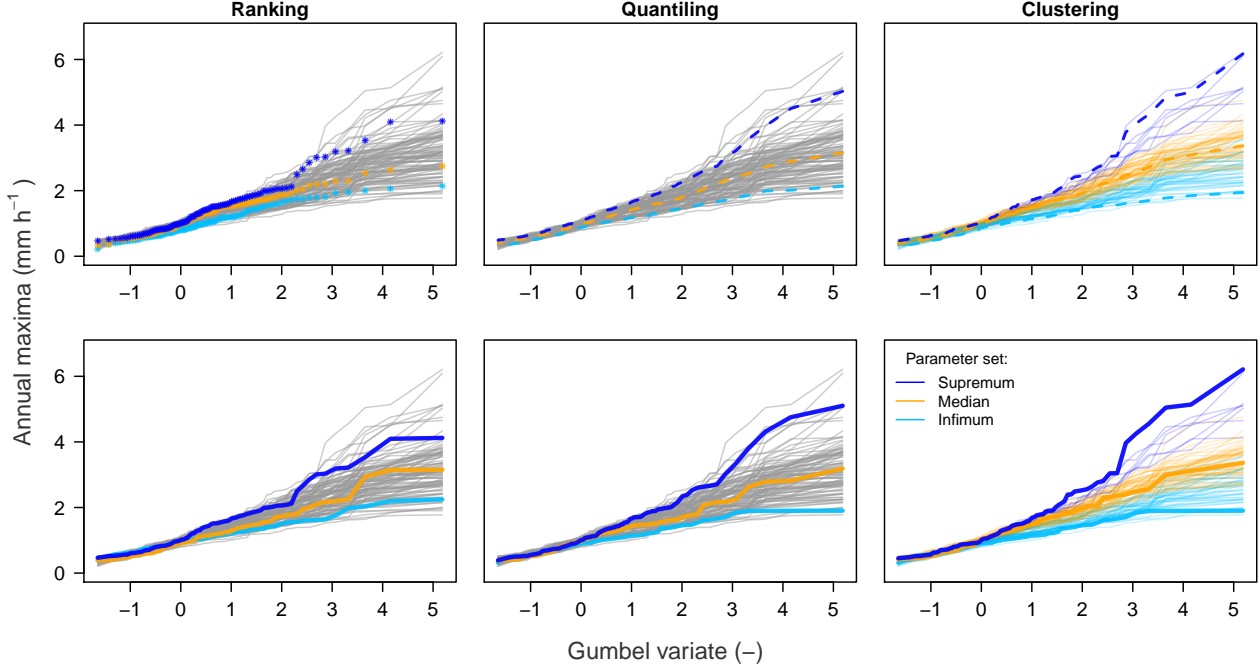

**Figure 6.** Example of the representative parameter sets' selection with three methods in the Dünnern at Olten catchment (meteorological scenario $m = 93$); description as in Fig. 5.

left out) is considerably narrower for ranking and quantiling on one hand and much wider for clustering on the other hand (top row in Figs. 7–8). Accordingly, the correspondence between the prediction interval and the full simulation range of the left-out scenario differs between the methods (bottom row in Figs. 7–8).

This is reflected in the quantitative assessment of the methods' behavior, summarized in Table 3. Namely, the leave-one-out cross-validation reveals that the quantiling method receives the highest values for both evaluation criteria, i.e., the median ratio of simulation points lying outside the predictive intervals ($R_{spo}$) and the median ratio of hydrological simulation ensemble members lying outside the predictive intervals ($R_{hso}$). Thus, this method performed the poorest among all three methods tested here. Yet, with $R_{spo} \leq 0.14$ for the 50th percentile and $R_{hso} \leq 0.05$ for the threshold $r_{thr} \geq 0.50$, even this method can be qualified as behaving well based on the leave-one-out cross-validation. For the ranking and the clustering methods, similar values for these two metrics are achieved, with slightly lower values for the ranking method.

In summary, it can be said that all criteria values are relatively low for all three methods, and thus the computed criteria values can only be used to order the methods by their behavior, while none of the methods is rejected.

In contrast to the above findings, the multi-scenario evaluation reveals different results with $R_{spo}$ being the lowest for clustering, and the largest for the ranking method. Similarly, the median ratio of meteorological scenarios lying outside the predictive


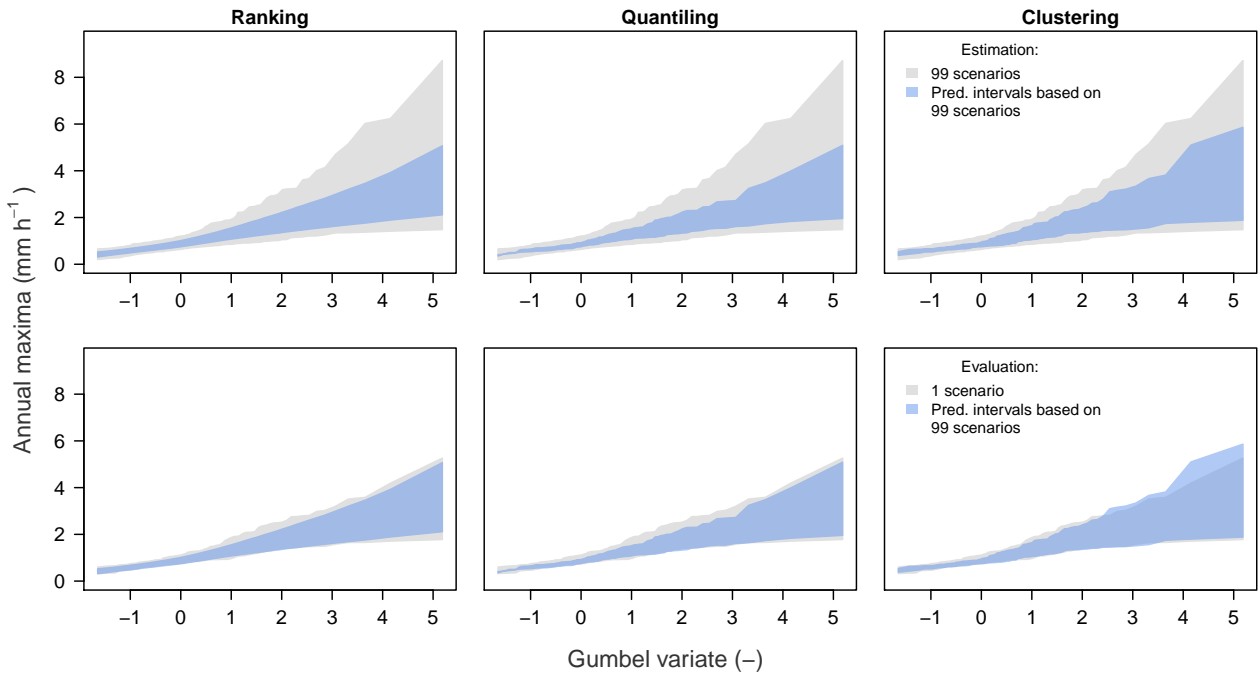

**Figure 7.** Example of leave-one-out cross-validation for the three selection methods (meteorological scenario $m = 14$). Top panel – illustration of the prediction interval resulting from selecting representative parameter sets for 99 meteorological scenarios (in blue) and compared to the full simulated range with all 100 parameter sets for all 99 scenarios (for each of the three methods). Bottom panel – comparison of the 99-scenarios prediction interval from the top row against the simulated range with all 100 parameter sets for the scenario left out during the prediction interval construction ($m = 14$).

intervals ($R_{\mathrm{mso}}$) is the lowest for clustering and the highest for the ranking method for all considered threshold values ($r_{\mathrm{thr}}$ in Table 3).

Also, here all computed criteria values are relatively low with $R_{\mathrm{spo}} \leq 0.05$ for the 50th percentile and $R_{\mathrm{mso}} = 0$ for the thresh-
old $r_{\mathrm{thr}} \geq 0.50$ for the poorest behaving method (ranking). Hence, again here all three method can be qualified as behaving well based on the multi-scenario evaluation, and only the order of their behavior can be established.

## 5 Discussion

### 5.1 Behavior of three selection methods

The results from our experimental study demonstrate that generally all three methods are capable of selecting representative
parameter sets that yield reliable predictive intervals in the frequency domain, i.e. all three methods are fit-for-purpose for extreme flood simulation, with the ranking method performing, however, clearly less well than the others (larger bias, as


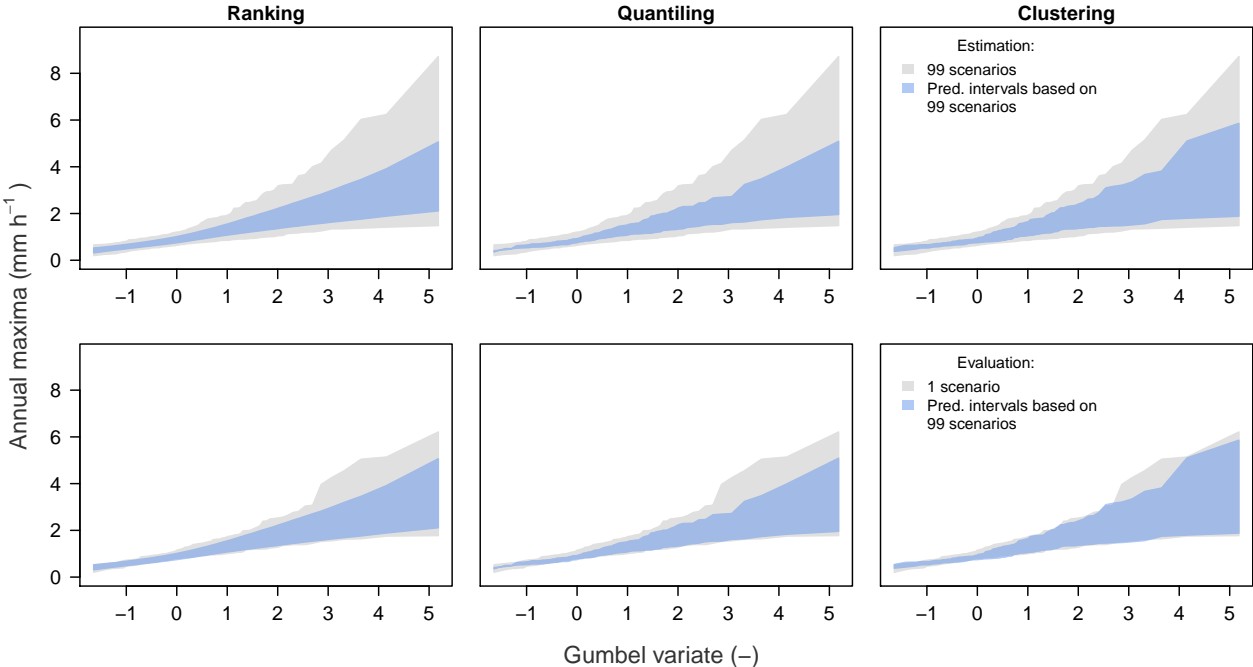

**Figure 8.** Example of leave-one-out cross-validation for the three selection methods (meteorological scenario $m = 93$); description as in Fig. 7.

**Table 3.** Metrics of the behavior of the approach for three methods of selecting representative parameter sets and the predictive intervals in the leave-one-out cross-validation and in the multiple-scenario evaluation. The values represent the median values over all 100 scenario runs.

| Metric / selection method | Leave-one-out cross-validation | | | Multi-scenario evaluation | | |
| --- | --- | --- | --- | --- | --- | --- |
| | Ranking | Quantiling | Clustering | Ranking | Quantiling | Clustering |
| $R_{spo}$ [-] | | | | | | |
| 5th percentile | 0 | 0.03 | 0 | 0.01 | 0 | 0 |
| 50th percentile | 0.02 | 0.13 | 0.065 | 0.048 | 0.02 | 0 |
| 95th percentile | 0.36 | 0.49 | 0.38 | 0.17 | 0.13 | 0.065 |
| $R_{hso}$ ($R_{mso}$ [*]) [-] | | | | | | |
| $r_{thr} \geq 0.50$ | 0.02 | 0.05 | 0.02 | 0 | 0 | 0 |
| $r_{thr} \geq 0.25$ | 0.13 | 0.28 | 0.17 | 0.025 | 0.01 | 0 |
| $r_{thr} \geq 0.10$ | 0.26 | 0.57 | 0.41 | 0.20 | 0.091 | 0.03 |
| $r_{thr} \geq 0.05$ | 0.38 | 0.76 | 0.55 | 0.51 | 0.26 | 0.071 |

[*] in the multi-scenario evaluation

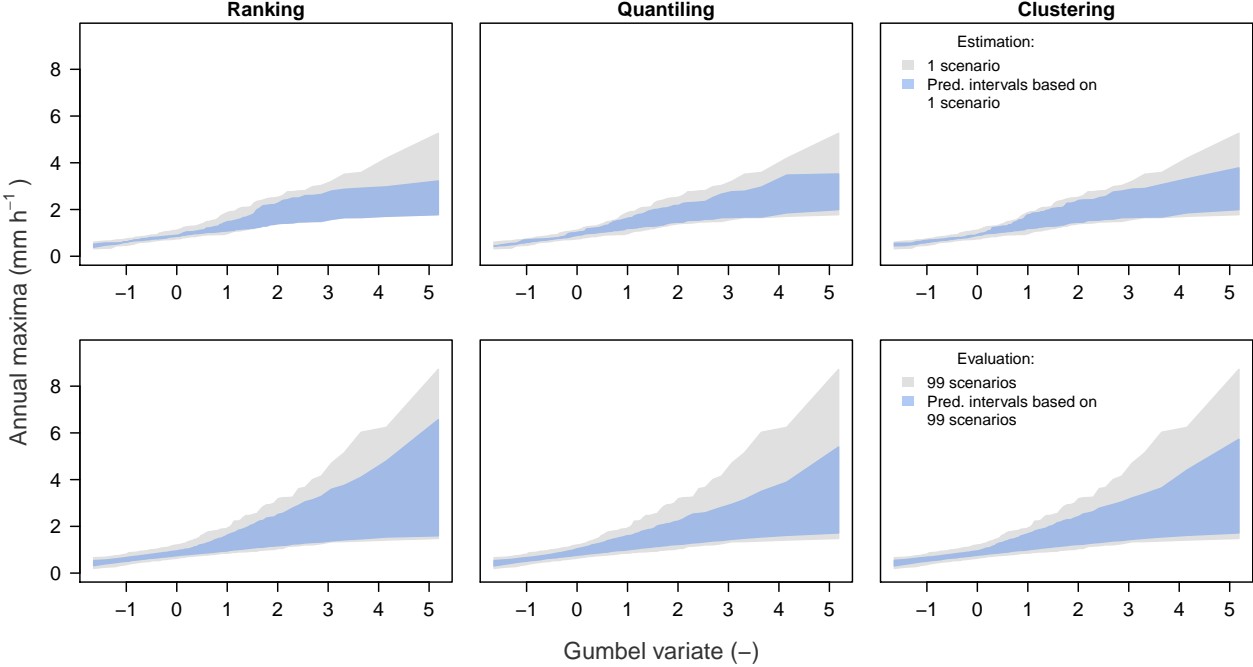

**Figure 9.** Example of multi-scenario evaluation for the three selection methods (meteorological scenario $m = 14$). Top panel – illustration of the prediction interval resulting from selecting representative parameter sets for a single meteorological scenario ($m = 14$, in blue) and compared to the full simulated range for this scenario (for each of the three methods). Bottom panel – comparison of the predictive interval resulting from the selected parameter set from the top row but applied to all remaining 99-scenarios against the full simulated range for these 99 scenarios.

visible in Sect. 4.2). As the developed methods rely on selecting three representative sets as infimum, median and supremum, they respect the maximum variability between individual ensemble members for a given meteorological scenario.

In the validation tests, the behaviour scores of the three methods, however, were attributed differently depending on the evaluation criteria. To further compare the methods, we provide below a detailed discussion of the major differences and present a synthesis of how the methods rank on average (averaged across all scenarios) for the quantitative evaluation criteria as well as for visual inspection (bias) and for additional ease of use criteria (Table 4).

From the visual assessment, it clearly appears that the ranking method is the most biased method (with more than half of all meteorological scenarios having mixed-up intervals), while the other two methods can be considered as being unbiased
with correctly attributed intervals for 98% (quantiling) or more (clustering) of all meteorological scenarios considered here (Sect. 4.2). As expected, these findings are further confirmed by the results from the multi-scenario evaluation that yield the best behaviour for the clustering method and the worst for the ranking method (Sect. 4.3).





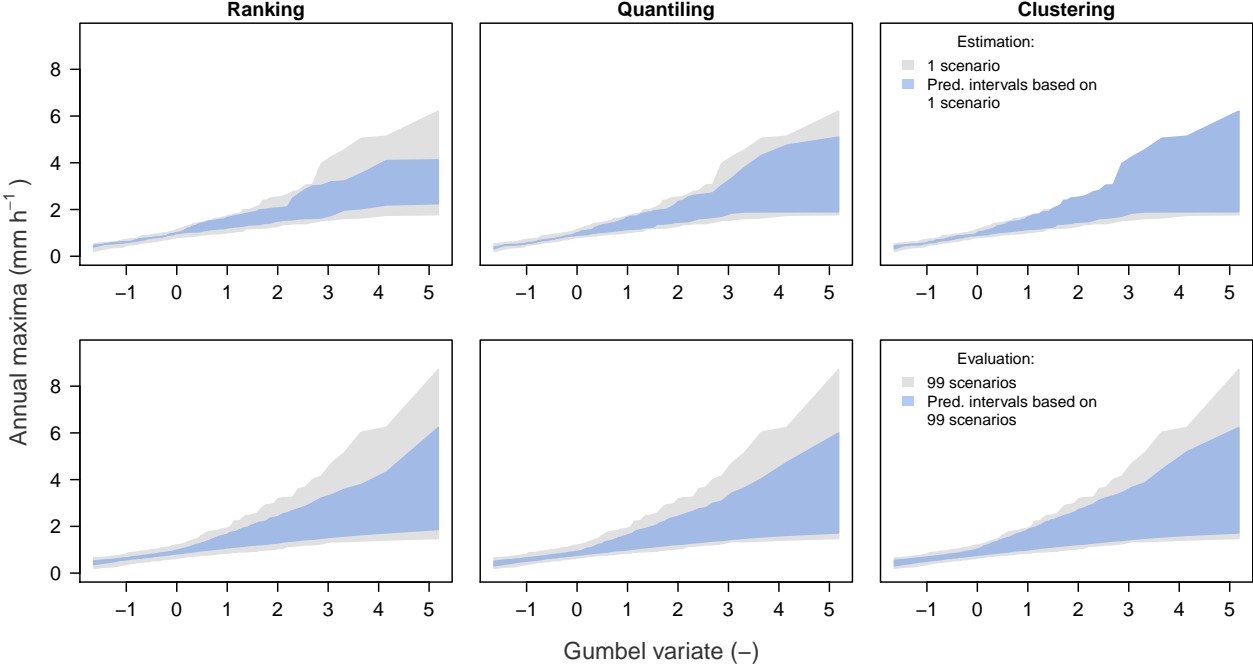

**Figure 10.** Example of multi-scenario evaluation for the three selection methods (meteorological scenario $m =$93); description as in Fig. 9.

**Table 4.** Synthesis of scoring ranks attributed to the three methods for selecting representative parameter sets (based on quantitative metrics). The ranks are attributed descending from the best (1) to the worst (3) behaviour. The median scoring rank (last line) corresponds to the median over all criteria.

| Score criteria | Ranking | Quantiling | Clustering |
|---|---|---|---|
| Unbiasedness (not mixed-up intervals) | 3 | 2 | 1 |
| Leave-one-out cross-validation | 1 | 3 | 2 |
| Multi-scenario evaluation | 3 | 2 | 1 |
| Independence from meteorological scenario | 3 | 1 | 1 |
| Independence from simulation years | 3 | 1 | 1 |
| Ease in application | 1 | 3 | 3 |
| Interpretation of prediction intervals | 3 | 1 | 1 |
| Median scoring rank | 3 | 2 | 1 |

Interestingly, the leave-one-out cross-validation study, in contrast to the the multi-scenario evaluation, attributes the lowest criteria value to the ranking method, i.e. ranks it as the best method (Table 4). This requires a careful interpretation and
understanding of how the predictive intervals are constructed in both evaluation studies. In the leave-one-out cross-validation



study, the representative parameter sets are selected and the predictive intervals are constructed based on 99 meteorological scenarios and then evaluated against the full simulation range corresponding to the left out scenario. In other words, this test evaluates how well the selection methods applied to all but one scenario can predict the full simulation range of the left out scenario. In the multi-scenario evaluation, the representative parameter sets are selected based on a single scenario, and the predictive intervals are then assessed by applying these three selected sets (selected based upon a single scenario) to the other 99 meteorological scenarios. This test quantifies how well the methods applied to a single scenario are transferable to all other scenarios.

Hence, by comparing findings from these two evaluation studies, it appears that the ranking method has the lowest transferability, i.e. performs poorly if using a single scenario for selecting the representative sets (multi-scenario evaluation). In exchange, the ranking method outperforms the two other methods when a high number of meteorological scenarios is used for selecting the representative parameter sets (leave-one-out cross-validation). This means that the ranking method strongly depends on the meteorological scenario choice, while the other two methods result in representative parameter sets that are transferable to other meteorological scenarios.

This outcome can be understood if we consider how the selection methods are constructed: The ranking method, in fact, depends strongly on the selected simulation period (and hence on the meteorological scenario) because the selection of the representative parameter sets is performed on unsorted annual maxima for each simulated year independently. The other two methods are performed over the entire simulation period, which makes them less strongly dependent on individual simulation years. However, the ranking method can be considered as the (computationally) easiest in application due to its selection criteria relying purely on ranking within individual simulation years. The other two methods need to be performed in the Gumbel space over the entire simulation period and, in the case of the clustering method, require some additional computational effort (which remains low, however, compared to the hydrologic simulation). The use of the Gumbel space in selecting the representative parameter sets helps, however, to interpret the constructed prediction intervals and to directly assign return periods to them.

Overall, it appears that the clustering method behaves the best (with a median scoring rank of 1) due to its unbiasedness and due to a good performance achieved for all evaluation criteria, for both the leave-one-out cross-validation and the multi-scenario evaluation (Table 4).

## 5.2 Limitations and perspectives

This study proposes a framework for representative hydrological parameter selection to be used within fully continuous ensemble-based simulation frameworks that are based on meteorological inputs generated with a weather generator. Based on our experimental case study, we demonstrate that the proposed three methods are reliable for downsizing the available full parameter sample down to a number of three representative sets. Possible applications of our methods include all fully continuous simulation schemes for extreme flood analysis where computational or methodological limitations prevent the simulation of full hydrological parameter distributions.

The analysis is based on a specific modeling setup with 100 calibrated parameter sets of the selected hydrological model (HBV), 100 meteorological scenarios derived from a weather generator, each of 100 continuous simulation years. This number


of 100 was chosen as a compromise between minimizing the intensive model calibrations and the simulations at an hourly time step, and maximizing the information content of the parameter sample and the climate variability. We have chosen the same number of 100 for meteorological scenarios, parameter sets and simulation years to not favourable any of these components in the methods' comparison.

We should emphasize that the presented methods are independent of the selected parameter calibration approach or from the
selected hydrological response model and are thus readily transferable to any similar simulation setting, in particular also to settings where the full parameter samples to be downsized come from model regionalization (i.e. in applications to ungauged or poorly gauged catchments). Moreover, although the methods are tested with a bucket type hydrologic model, the most valuable application of the proposed methods would be to computationally more demanding hydrological models that can profit even more from a reduced computational demand.

Furthermore, the proposed approach is tested here using synthetic hydrologic data, i.e., using streamflow simulations of the hydrological model in response to meteorological scenarios. This use of synthetic data makes the approach results independent from the catchment properties and limits the effect of the hydrological model error and errors in calibration data on the methods' comparison results.

We can, however, not directly assess here how much variability in the full hydrological ensemble is due to the climate
variability and how much is due to the uncertainty resulting from the hydrological model parameters, because these two components are not linearly additive. This can easily be seen by comparing the ranges of predictive intervals constructed using one scenario and 99 scenarios in the multi-scenario evaluation for two example scenarios in Figs 9-10. In addition, any ensemble simulation also encompasses other uncertainty sources of the modeling chain, such as resulting from the weather generator or from the structure of the selected hydrological model, from the prediction of very rare flood events, etc. (Lamb
and Kay, 2004; Schumann et al., 2010; Kundzewicz et al., 2017).

Hence, downsizing the hydrological model parameter sample can only aim at understanding and characterizing the hydrological part of the full hydrological ensemble resulting from a combination of multiple parameter sets and multiple meteorological scenarios. These methods are however not applicable for characterizing the climate variability (nor for downsizing the number of meteorological scenarios needed).

Moreover, in developing the selection methods, we did not distinguish between different flood-types such as heavy rainfall-excess or intensive snowmelt events (Merz and Blöschl, 2003; Sikorska et al., 2015). Also, as we focused only on large annual floods (annual maxima), we did not represent the flood seasonality in our analysis. Yet, some recent works emphasise the need to include such information on the flood type (Brunner et al., 2017) or on flood seasonality (Brunner et al., 2018b) into bivariate analysis of floods, or to represent a mixture of both flood type and flood seasonality in flood frequency analysis (Fischer et al.,
2016; Fischer, 2018). Thus, the proposed selection methods could potentially be extended to account for different flood types during representative parameter selection, using e.g. automatic methods of flood type attribution from long discharge series (Sikorska-Senoner and Seibert, 2020).

Finally, we downsize the parameter sample to three sets which represents the predictive intervals of the full ensemble of hydrological responses fairly well, given different meteorological scenarios. This number of three sets is motivated by the fact





that it can be readily processed within a fully continuous ensemble-based framework using numerous climate settings. This is common practice in flood frequency analysis, and the three sets emulate the common practice of communicating median values along with prediction limits (Cameron et al., 2000; Blazkova and Beven, 2002; Lamb and Kay, 2004; Grimaldi et al., 2012b). Optionally, one could further downsize the parameter sample to two sets (i.e., infimum and supremum) which would represent the intervals only. Downsizing to more than three parameter sets (e.g. five) could have an advantage of containing

more information on uncertainty intervals, e.g. in the case they are asymmetric, and should be explored in further study.

## 6 Conclusions

In this study, we propose and test three methods for selecting the representative parameter sets of a hydrological model to be used within fully continuous ensemble-based simulation frameworks. The three selection methods are based on ranking, quantiling and clustering of simulation of annual maxima within a limited time window (100 years) that is much shorter

than the full simulation period of thousands of years underlying the simulation framework. Based on a synthetic case study, we demonstrate that these methods are reliable for downsizing a parameter sample composed of 100 parameter sets to three representative sets that represent most of the full simulation range in the Gumbel space. Among the tested methods, the clustering method that selects parameter sets based on cluster analysis in the Gumbel space, appears to outperform the others due to its unbiasedness, and due to its transferability between meteorological scenarios. The ranking method, which is the

only tested method that completes the parameter selection on non-sorted annual maxima, can clearly not be recommended for typical settings since it i) tends to result in mixed-up prediction intervals in the Gumbel space and ii) depends too strongly on the simulation period used for parameter selection and thus lacks transferability to other periods or other meteorological scenarios. Possible applications of these methods include all fully continuous simulations schemes for extreme flood analysis, and particularly those for which computational constraints arise.

*Data availability.* The observed discharge data for calibrating the hydrological model can be ordered from the FOEN (https://www.bafu.admin.ch, last access: 12 February 2020), the observed meteorological data from MeteoSwiss (http://www.meteoswiss.ch, last access: 12 February 2020), and the topographic data from Swisstopo (http://www.swisstopo.ch, last access: 12 February 2020).

## Appendix A: Semi-continuous versus fully continuous simulation approach

This appendix briefly discusses the conceptual difference between the semi-continuous and the fully continuous approach from

two perspectives, i.e., from the hydrologic and the hydraulic perspective (Fig. A1). From a hydrologic perspective, an approach is continuous if hydrographs are selected from the continuous time series of discharge simulated with a hydrologic model, and is semi-continuous if the hydrologic model is used only to simulate initial conditions prior to the flood event. From the hydraulic perspective, two types of semi-continuous approaches can be distinguished that are marked here as semi-continuous I, in which a continuous hydrologic model is used to simulate initial conditions prior to the storm event (e.g. Paquet et al.,





2013; Zeimetz et al., 2018), and semi-continuous II, in which a continuous hydrologic model is used to simulate continuous discharge time series (e.g. Calver and Lamb, 1995; Cameron et al., 2000; Blazkova and Beven, 2004; Hoes and Nelen, 2005; Winter et al., 2019). In both of these semi-continuous approaches, simulated or design hydrographs are further routed through the hydraulic model. In contrast to these semi-continuous approaches, in the fully continuous simulation approach, as seen from the hydraulic perspective, full discharge time series are routed through the hydraulic model (Grimaldi et al., 2013).

**Appendix B:  Study catchment: Dünnern at Olten**

The locality map of the study catchment is presented in Fig. A2.

**Appendix C:  Details on the HBV model parameters and model calibration**

For searching the best parameter sets within the defined parameter ranges (Table A1), a Genetic Algorithm and Powell optimization (GAP) approach (Seibert, 2000) is used. This approach is executed in two major steps. Firstly, the GA optimization is

performed that relies on an evolutionary mechanism of selection and recombination of a user defined number of parameter sets (i.e., parameter population) randomly selected within the defined parameter ranges. The principle idea of this searching relies on regenerating the parameter sets from the subgroup of parameter sets selected using the defined objective function $F_{obj}$ as a criteria to choose the parameters that give the highest value of $F_{obj}$ at the previous step of the model calibration. The search for the best parameter set is terminated at a user defined maximum number of model interactions and results in a selected

optimal parameter set. Secondly, the optimal parameter set obtained at the previous step is used as a starting point for a local optimization search using the Powell's quadratically convergent method (Press et al., 2002). The parameter set finally achieved from the local optimization is retained as the best set. In this study, the total number of model interactions is set to 2500 for the GA and 500 for the local Powell's optimization. The GAP optimization is repeated 100 times resulting in 100 optimized parameter sets (Figure A3).




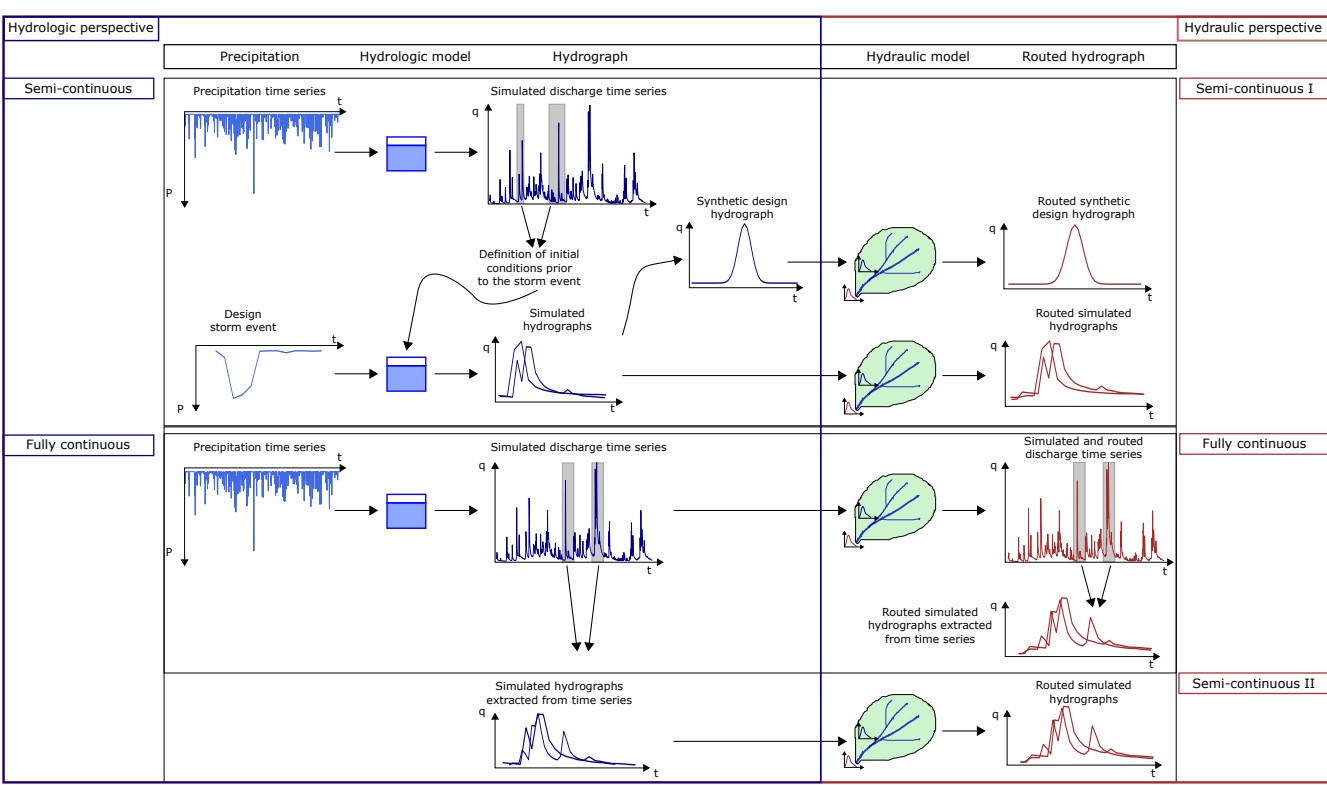

**Figure A1.** Overview of two simulation approaches from the hydrologic (marked in blue) and hydraulic (brown) perspective.


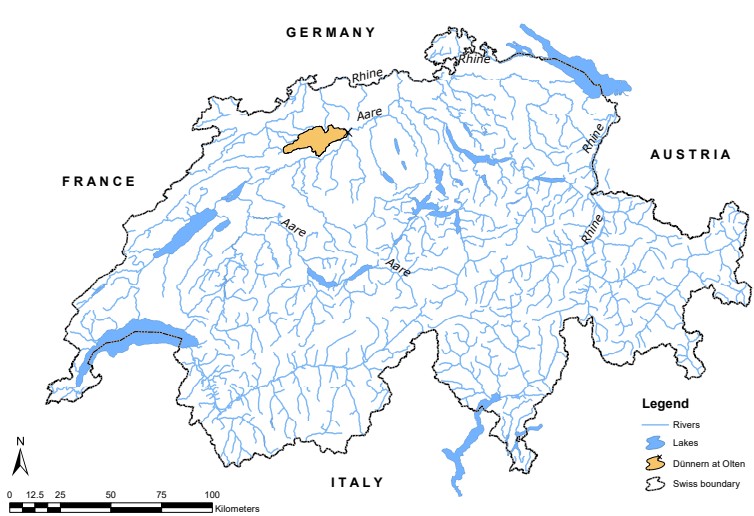

**Figure A2.** Location of the Dünnern at Olten catchment with river network extracted from Swiss Map Vector 25 (SwissTopo, 2008).


**Figure A3.** Violin plots summarizing 100 optimized parameter sets of the HBV model for the Dünnern at Olten catchment. Units as in Table A1.




**Table A1.** Parameter ranges for the calibration of the HBV model

| Parameter | Unit | Min | Max | Description |
|:---:|:---:|:---:|:---:|:---:|
| PERC | mm h$^{-1}$ | 0 | 1 | Percolation parameter |
| UZL | mm | 0 | 100 | Groundwater runoff threshold parameter |
| K0 | h$^{-1}$ | 1e-04 | 0.2 | Recession coefficient |
| K1 | h$^{-1}$ | 1e-05 | 0.1 | Recession coefficient |
| K2 | h$^{-1}$ | 1e-08 | 0.05 | Recession coefficient[*] |
| MAXBAS | h | 1 | 100 | Length of triangular weighting function |
| CET | °C$^{-1}$ | 0 | 0.5 | Correction factor for potential evaporation |
| TT | °C | -2.5 | 2.5 | Threshold temperature |
| CFMAX | mm h$^{-1}$ °C$^{-1}$ | 1e-03 | 5 | Degree-hour factor |
| SFCF | - | 0.4 | 1.6 | Snowfall correction factor |
| CFR | - | 0 | 0.1 | Refreezing correction factor |
| CWH | - | 0 | 0.2 | Water holding capacity |
| FC | mm | 50 | 550 | Maximum moisture storage in soil box |
| LP | - | 0.1 | 1 | Threshold for reduction of evaporation |
| BETA | - | 1 | 10 | Shape coefficient |

[*] For recession coefficients the following condition must be fulfilled: $K0 > K1 > K2$



*Author contributions.* AS and JS developed the idea. AS and BS jointly designed the details of this study. AS performed the analyses and produced the figures. AS with a contribution of BS wrote the first draft of the manuscript. The manuscript was edited by BS and revised by JS.

*Competing interests.* The authors declare that they have no conflict of interest.

*Acknowledgements.* The research was funded through the Forschungskredit of the University of Zurich, grant no. FK-18-118. The Sci-
enceCloud provided by S3IT at the University of Zurich enabled the computation-intensive simulations to be run on virtual machines. Meteorological scenarios used in this study were made available through the project EXAR (Hazard information for extreme flood events on the rivers Aare and Rhine), Project No. 15.0054.PJ/O503-1381, funded by the Swiss Federal Office for Environment (FOEN). The authors wish to give a special thank to Guillaume Evin, Anne-Catherine Favre and Benoit Hingray for preparing the meteorological scenarios, to Marc Vis for setting-up virtual machines, and to Tracy Ewen for proofreading the manuscript.



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
