# Peer review of "Downsizing parameter ensembles for simulations of extreme floods"

_Natural Hazards and Earth System Sciences, 2020_

## Referee Comment (RC1) · Anonymous Referee #1 · 1 May 2020

PLEASE REFER TO SUPPLEMENT PDF FOR BETTER TYPESETTING OF THE REPORT

General comments:

The paper is well written and properly structured. The scope of the study is clear, and the purpose of the selection of models among a given ensemble is relevant in the perspective of hydrological and hydraulic simulation for extreme flood estimation. However, the presented methods, their scoring, and the interpretation of these scores deserve a better statistical assessment. For example, the presented scores, used in the comparison of the selection methods, rely only on the counts of simulated annual maxima being in or out of the predictive interval, without consideration of scale or frequency within their distribution. No assessment of the width of the predictive intervals

is given relative to a global metric (standard deviation of annual maxima of the calibration data for example). Even if a relative ranking of the three methods can be provided here, it is difficult to have a proper statistical characterization of them in terms of robustness and reliability, which can be an issue for using them for extreme flood evaluation. Furthermore, no interpretation of these selections in term of flood process or modelling is given. It is not supposed to be the core of this article, but it would help to connect the results to some hydro-climatological features and their impact in terms of variability. The problem of parameter equifinality is not evoked here, although it is the main factor of the parameters set variability, given that here, the 100 models have been calibrated by the same algorithm using the same data. I would recommend then a major revision of this paper, in order to tackle with those main issues. 

Detailed comments/questions:

Quoted sentences are written in italic.

Line 10: 10'000 years of synthetic streamflow data simulated with a weather generator. Simulated "thanks to" a weather would be more appropriate, the weather generator doesn't generates streamflow directly, it feds the hydrological models. Line 12: The methods are readily transferable to other situations where ensemble simulations are needed. This is only evoked as a perspective at the end of the paper, without providing an example of such "other" application. I am not sure it deserves to be in the abstract.

Line 39: initial conditions for use in combination with design storms. Regarding the SCHADEX method detailed in (Paquet el al., 2013), I would add "initial conditions for use in combination with design or randomly drawn storms".

Line 43: especially if long time series are to be simulated using ensembles of hydrological parameter sets. And also if very high return times (above 1000 years) have to be robustly estimated, thus implying several thousand years of simulation.

Line 45: extrapolating a synthetic design hydrograph. How? By scaling up a synthetic

hydrograph thanks to estimated extreme quantiles of peak and volume values?

Line 51: These continuous hydrologic simulation frameworks are still rare for time series >100 years. 100 years of simulation merely allow to compute a robust 20 year return period estimation, which is pretty useless for dam safety for example. I would rather say that "high computational power are needed in order to provide estimations for high to extreme return times (up to 1000 years) required for safety-related studies" (although this is almost written in the same terms in line 57).

Lines 77-86: The problem is particularly well stated here.

Line 89: to select a reduced-size parameter ensemble for the use with a hydrological model within a continuous simulation. Here and later on I would always keep "parameter" linked to "hydrological model". Most simulation frameworks are heavily parametrized, and the uncertainty linked to the hydrological model is only one (important) of the numerous sources of uncertainty. I would then write "to select a reduced-size ensemble of hydrological model parameters for the use within a continuous simulation".

Line 92: for simulation of extreme floods. A recurrent formal remark about the word "extreme". Usually "extreme floods" refers to return times largely exceeding the observational range, currently more than 1000 years, thus being extrapolated (by FFA or simulation, or both). This is especially true in dam-safety related literature. In the presented case, the meteorological scenarios are 100 years long, meaning that only very few "extreme" floods are simulated in the whole experiment. At best a robust 1000 years estimation can be empirically inferred here given the fact than 100x100=10 000 meteorological years have been simulated. The whole set of AM being extracted can surely not be considered as a set of "extreme floods". The authors could consider using "intense floods", "rare floods" or more simply "floods" when they refer to the simulated floods.

Line 97: simulated rare flood events. Following the remarks above, the term "rare flood"

is also appropriate alternative.

Line 99: the aim is thus i) to provide long enough simulation periods for extreme flood analysis, ii) to avoid the propagation of errors due to data/model calibration etc. and iii) to be able to focus entirely on the uncertainty of the hydrological response. I my opinion this goes farer than the actual results of the study. The uncertainty linked to model parameters is assessed, and properly summarized thanks to a reduced number of meteorological simulations. But I don't understand why this "provides long enough simulation", and why it "avoids the propagation of errors". If you have 100 "bad" models due to date, calibration, etc., three of them are selected in order to keep a good representation of their variability, but you still work with "bad" models (sorry for the term "bad", it only means "affected by uncertainty" !).

Line 111: and not the model uncertainty of a weather generator. This is perhaps one of the main limit of the study. At line 69, it is written that Arnaud et al. (2017) found that the uncertainty of the rainfall generator dominates the uncertainty in the simulated extreme flood quantiles. This uncertainty will not be considered here, and I wonder how far the results exposed here would still be useful to deal with the weather generator uncertainty (which of course is not to be confused with the variability of the scenarios generated thanks a given set of parameters). A comparison of both uncertainties (model and weather generator), even basic, would have been welcomed here.

Line 120: (ii) the distribution is known. I am not sure that knowing the probability of parameters is a reasonable perspective, in my opinion the problem of equifinality of parameters in models like HBV prevents an a priori expression of parameter probability, as different sets of parameters can "produce" the same model, i.e. models having the same behaviour for a given meteorological scenario. And this is one of the interesting outcome of this study, which focuses on the hydrological response of the models, and not on the actual values of the parameters. I think that this equifinality problem deserves more writing in this paper.

Line 129: The infimum (from the Latin – smallest) and supremum (from the Latin – largest) refer to the greatest lower bound and the least upper bound (Hazewinkel, 1994), i.e., the largest interval bounding the ensemble from below and the smallest interval bounding it from above. This definition deserves to be connected to frequencies of the target variable, even if it's not straightforward. Does it (roughly) provides a 90, 95 or 99% confidence interval of the simulated variable? Given that follows in lines 184 to 234, with quantiles 5 and 95%, it "looks like" a 90% CI.

Line 138: we thus propose to use the representation of AMs in the Gumbel space as the reference model response space for parameter selection. The plotting in Gumbel is useful here to illustrate the rare to extreme quantiles, but doesn't explicitly play a role in the "parameter selection" (which doesn't imply any explicit reference to an implicit Gumbel distribution of AM in the statistical criterions/indicators used).

Line 142: inverse modelling approach. The term "inverse modelling" appears to me somehow excessive. An inverse hydrologic modelling would be for example to infer rainfall from discharges. Here it's more a "post-modelling" approach.

Line 145: the parameter set selection is made based on the full hydrological simulation ensemble but using only a limited simulation period. To be more specific I suggest to write "based on the simulation with all the hydrological models but using [. . .]".

Line 189: The parameter sets selected in step (d). Should be step (c).

Line 189: the sets which are chosen most often as the 5th, 50th and 95th ranks are retained as the parameter sets [. . .] representative for the entire simulation period. The ranking methods yet shows its weakness: the 5th and 95th of a given year have very low chance to match to the overall corresponding quantiles, given the "climate variability" illustrated in Figure 1, thus preventing the parameters selected on a given year to have a global representativeness. I am not sure it's worth keeping this method "in the game" for the rest of the paper. . .

Lines 194-206: I don't understand why the "Gumbel space" is evoked here (three times!), and constantly throughout the paper. Apart from the plots of Figure 2 and others, what is "Gumbel specific" in the metrics and statistics presented? For example, the RMSE scores are computed using each simulated annual maximum, regardless its empirical frequency.

Line 201-205, equations 1-3: This is the same equation for the three considered quantiles. Only one is necessary.

Line 211-234: same remark as above about the "Gumbel space".

Lines 214: These members are next clustered in the Gumbel space into three representative groups (clusters) based on all J simulation years using the k-means clustering. If I understood properly it means that the clustering has been performed in the J-dimensional space of the full set of members values?

Line 217: Next, these clusters are sorted by their magnitude. What variable/quantile is used for this sorting?

Line 218: Note that we use here percentiles instead of cluster means to make this method comparable with the other two methods. I am not sure of that : say that each cluster regroups one third of the ensembles, and for a given quantile in the AM distribution (say the 50%), it is evenly distributed through all the members, the percentile 5% of the lower cluster would more or less correspond to a 0.05x0.33 = 0.17 global percentile. The 5% and 95% are more "rare" than their corresponding quantile in the quantiling method. . .

Line 223-227, equations 4-6: Same remark as for equations 1 to 3.

Line 233, equation 7: This mention of the plotting position mention could be moved at line 195.

Table 1: Sorting space = Gumbel space. Once again, I don't undertand how "Gumbel-specific" the sorting process is for Quantiling and Clustering.

Table 1: Interpretation of pred. intervals / Parameter grouping. I don't see to what these lines refer in the text before.

Line 254: assessing how well the reduced ensembles cover the reference simulation ensemble. More specifically I would rather say "how well the reduced ensembles substitute the whole simulation ensemble for the selection of representative parameter sets".

Line 273: assess how well the defined identified intervals represent the ensemble members of this Sr meteorological scenario. What metric is used to do this assessment?

Line 288: Compute the 5th percentile [. . .] and the 95th for {H(sup,p/Sm)}. The mention "for m=1,2. . .,M and m≠p" could be added for more clarity.

Line 290: same question as for line 273.

Line 295 and below: as evoked above, the recurrent mention to the Gumbel space is, in my opinion, useless, and over time tedious to read. In the paper, it is quickly implicit that the plots and the metrics used are defined in the frequency space (or frequency domain), being Gumbel or not, without need to repeat it.

Line 330: Here we propose to use different percentiles, i.e., the 5th, 50th, and 95th percentiles, to characterize the ratio of the simulation points lying outside the computed predictive intervals for each of the methods. I don't understand this? Why not using only the 50th percentiles of this ratio? Refer to comments on Table 3 for a more detailed version of this question.

Line 333: how many out of J hydrological simulation points [. . .] must lie outside the defined predictive intervals. I think that the problem of such a simple "count" of points (simulated annual maximum) outside the predictive interval doesn't take into account their position in the simulated distribution. As written in the title, the methods exposed here are supposed to be used in the estimation of extreme floods, which in any posttreatment of the hydrological simulation will strongly rely on the high simulated quantiles. The scores should somehow reflect this focus on high quantiles, which is not the case here. Instead of this count of "outside points", the area outside the predictive interval could be computed, using the Gumbel variable (as x) and the discharge value (as y), thus giving a contrasted score in which lying outside the predictive interval for high quantiles is more important than for low values.

Line 337: In this work, we consider the following values for rthr = {0.50,0.25,0.10,0.05}. Following the preceding comment, a metric accounting for the scale or the frequency of the points being outside the predictive interval would avoid to distinguish such thresholds, which apart from the rtht=0.50 or 0.10 have little statistical meaning in this context.

Line 343: For testing the methods developed here, a small natural catchment is preferable. Why small?

Line 358: In this study, the version HBV light [. . .] with 15 calibrated parameters is used. The considered model can be then considered as heavily parametrized, and thus fully affected by the equifinilaty problem of its parameter evoked in the remark made for line 120.

Line 374: for details on RPEAK and RMARE, see the work of Vis et al. (2015). A brief description of RPEAK and RMARE would be welcomed here, especially as within the calibration process, the parameters conditioning the modelling of floods are surely strongly conditioned by the RPEAK score.

Line 377: The available observational datasets are split into a calibration (1990-2005 years) and a validation (2006-2014 years) period. What is the point of having a validation period here? This validation period is never used in that follows.

Line 381: The calibration is repeated 100 times resulting in 100 independent optimal parameter sets. I am surprised by the variability of the parameters obtained by these 100 calibration runs, performed on the same calibration data with the same objective

function. I would like to read a comment from the authors on that. Mine is that the optimization is not complete, seeming to depend on the aleatory exploration performed by the genetic algorithm, somehow "trapped" in local optimums, and/or affected by a strong equifinality problem (yes, once again, sorry). An alternative strategy for the generation of model parameter sets, in my opinion providing more "independent" models, could be to bootstrap 12 years among the 24 years available in order to generate 100 truly different calibration & validation samples.

Line 381: The median model efficiency measured with Fobj over all 100 runs is 0.7. To better assess the quality and the variability of the models generated at this step, it would be useful to show the distribution of NSE (Nash & Sutcliffe Efficiency) for both calibration & validation, and the ensemble plots of daily regime and classified discharge distribution for all the generated models. The ensemble simulation of the biggest observed would also be very pedagogic.

Line 383: which can be assumed to be a good model performance on an hourly scale. As mentioned above, this really need to be illustrated more richly.

Line 397: The daily values generated with GWEX_Disag were then disaggregated to hourly values using the meteorological analogues method. More details would be welcomed on that disaggregation: what fields/variable are used for analogy, what analogy criterion, what about seasonality (i.e. are the analogues identified within period of the year similar to the one of the simulation to be disaggregated, etc.).

Line 399: Next, catchment means were computed using the Thiessen polygon method. On how many simulated precipitation stations do the Thiessen average rely on for the considered catchment? How many simulated stations lie within the catchment?

Line 403: Thus, differences between scenarios are exclusively due to the natural variability of the meteorological time series. "[. . .] and modelled by the GWEX weather generator" could be added. Similarly to the models, the variability of these scenarios deserve to be illustrated, and compared to the observations, e.g. thanks to their

average and standard deviation of the annual maximum daily precipitations.

Line 406: These 100 meteorological scenarios are used as input into the HBV model to generate streamflow time series with 100 different HBV parameter sets. I am not sure that this sentence is useful. The simulation scheme is clear from the beginning.

Figure 4: The title of the second plot should be "Quantiling" instead of Quantailing.

Lines 419-434: I find the results of this paragraph difficult to interpret. Some violin plots show odd parameter selection patterns (like in Clustering/Infimum), other show weak parameter discrimination (Quantiling/Median). The Table 2 is quite difficult to read/interpret with so many counts exposed. In this paragraph and in the following ones, some "illustrations" of the most selected parameter sets should be provided, e.g. by presenting the range of hydrological responses to observed meteorological data of the selected models compared to the full ensemble. In other words, some interpretation in term of modelling and hydrological processes would be welcomed.

Line 434: Interestingly, for the supremum set in the clustering method, only four parameter sets among all 100 available are chosen over all 100 scenarios. Given that, I don't understand why in Table 2, column Clustering/sup, 5 parameters sets (# 34, 22, 98, 86, 50) are identified.

Table 2: a graphical alternative or a complement to that table deserves to be presented, to better assess the "density" of parameter sets selected by the different methods.

Line 437: intervals for extreme flood predictions. The term "extreme flood estimations" could be more appropriate.

Line 441: According to a first visual assessment, these three methods lead to slightly different constructed frequency intervals particularly in the upper tail of the distribution. To ease this visual assessment, horizontal lines marking these intervals for the upper values could be added to the plots of Figure 5.

Line 446: the three intervals are always correctly attributed. I would temper this in

writing that "the three intervals are always correctly ordered" as this exactly what it is measured in Rbias.

Line 456: From the visual assessment, it is difficult to judge the methods. See remark on Figures 7-8. Figure 5 to 8: Instead of having an x-axis graduated with the Gumbel variable U, some ticks at remarkable return times (2, 5, 10, 20, 50 & 100 years) could be added in order to ease the reading of these plots, and avoid the long caption The Gumbel variates etc. in Figure 4.

Line 463: the highest values for both evaluation criteria, i.e., the median ratio of simulation points lying outside the predictive intervals (Rspo) and the median ratio of hydrological simulation ensemble [...]. Given the definitions of §2.6.2, this is more a mean ratio than a median ratio.

Line 475: Hence, again here all three method can be qualified as behaving well based on the multi-scenario evaluation, and only the order of their behavior can be established. Honestly, at the end of this paragraph, I have no clear idea of the absolute performance of each method. One important point is that the methods provide rather different intervals (like illustrated in the Figure 6), thus a method providing wide intervals will have good "in/out" scores (like the ones in Table 3), better than for narrow intervals, but the question of the statistical relevance of such intervals is not solved.

Figure 7-8: I found the plots of the top panels of both figures rather counterintuitive: the prediction interval resulting from selecting representative parameter sets for 99 meteorological scenarios and compared to the full simulated range with all 100 parameter sets seems narrower than "statistically expected" (more and less a 5-95% confidence interval given the quantiles or percentiles involved in the process). For the highest simulated quantiles, the prediction interval seems only to cover about 50 to 66% of their variability. In the bottom plot of the Figure 7 for clustering, a second blue interval is plotted without being identified in the legend nor in the caption.

Line 480: [...] selecting representative parameter sets that yield reliable predictive intervals in the frequency domain. Following the comment on line 475, I see no statistical demonstration of the reliability of the predictive interval (like the one that could be done by controlled random generation of a given variable to which a statistical test is applied, then a proper statistical scoring). I agree with the authors on that a ranking between the three methods (according to the presented scores) is however established.

Table 3: Three quantiles of the Rspo score are given, although the caption mentions that the values represent the median values over all 100 scenario runs. What for providing the 5th and 95th quantile of a score measuring the ratio of simulation points [. . .] lying outside the predictive intervals (line 312), which should be, on average, close to 10% (once again given the quantiles involved in the selection process)? In the low part of the Table 3, the Metric method is written as Rhso (Rmso). Which scores are the ones provided?

Line 481: all three methods are fit-for-purpose for extreme flood simulation. Following the preceding comment, if the presented method cannot be statistically demonstrated, it can be considered as an ad-hoc heuristic, build for a given purpose, here extreme flood simulation/estimation. This last step is not evoked in the paper, then depriving the reader from assessing the relevancy/robustness of this heuristic.

Line 487: for additional ease of use criteria. I don't understand this sentence.

Line 488: From the visual assessment. Based on which figure?

Table 4: The different ranking features should be linked to the scores presented in Table 3. Some of them deserve to be better explained in the text: Independence from meteorological scenario, Independence from simulation years, Ease in application, Interpretation of prediction intervals. They don't refer explicitly to scores, statistics or plots presented before.

Line 497-502: This lines could be put in section 2.5 in order to better describe the assessment of the approach.

Line 514: The other two methods need to be performed in the Gumbel space over the entire simulation period and, in the case of the clustering method, require some additional computational effort. Once again this reference to "Gumbel space" is unappropriated given the scores computed, and the additional computational effort doesn't seem significant, completely justified by the added robustness.

Line 516: The use of the Gumbel space in selecting the representative parameter sets helps, however, to interpret the constructed prediction intervals and to directly assign return periods to them. Same remark as for line 295.

Lines 522-534: These lines are, in my opinion, a short summary of the study, and do not fit in this section (Limitations and perspectives).

Line 541: This use of synthetic data makes the approach results independent from the catchment properties and limits the effect of the hydrological model error and errors in calibration data on the methods comparison results. I may be more cautious on that, given that the scores and the ranking of the methods are somehow linked to 1) the variability of the ensemble models, which depends on the equifinality of the model's parameters, the calibration data and FOs, etc. and to 2) the meteorological variability of intense events generated by the weather generator which depends on the climatology, the scale etc. Only some tests on different catchments (in scale and climatology) could ground this assertion.

Line 544: We can, however, not directly assess here how much variability in the full hydrological ensemble is due to the climate variability and how much is due to the uncertainty resulting from the hydrological model parameters, because these two components are not linearly additive. A simple exercise could help by 1) choosing a "median" model (in term of median response on the meteorological ensemble) and plotting/scoring the variability of simulations for the whole meteorological ensemble, and 2) choosing similarly a median meteorological scenario and simulating with the whole set of models and then 3) comparing the spread/variance of definite quantiles in the simulations. In my opinion, this is an indispensable complement to the presented results.   Line 560: Thus, the proposed selection methods could potentially be extended to account for different flood types. Another option could be to consider Peak-Over-Threshold selection instead of a block selection (annual maximum) in building the simulated distributions. If different flood processes are present above a certain intensity threshold, flood type and seasonality sampling will be relevant.

Line 566: the three sets emulate the common practice of communicating median values along with prediction limits. But in that case, these predictive intervals have to be statistically calibrated (or checked) in order to be used in safety studies, especially if these studies lead to engineering or compliance check.

Line 572-584: Here again, the term Gumbel space could be replaced by "frequency domain" or equivalent.

Appendix A & Figure A1: Interesting but rather off-topic here, as only continuous hydrological simulation has been used in this study.

Figure A3: For a better assessment of the distribution of calibrated parameters, I suggest that the scale of the horizontal axis of the violin plots (parameter values) exactly matches the corresponding calibration range written in the Table A1.

Please also note the supplement to this comment:
https://www.nat-hazards-earth-syst-sci-discuss.net/nhess-2020-79/nhess-2020-79-RC1-supplement.pdf

---

## Referee Comment (RC2) · Anonymous Referee #2 · 8 Jun 2020

General comments:

The contribution provides an interesting approach to the selection of representative parameter sets for continuous hydrological modelling in the framework of derived flood frequency analyses considering uncertainty. The methodology is quite clear and plausible. The manuscript is well written and concise. I have only some minor comments for improvement (see detailed comments).

Detailed comments:

1. Line 129: . . . "selected in step (d)" should read "selected in step (b)"

2. Line 196: It is not clear to me how $Q5$, $Q50$ and $Q95$ are obtained? For each parameter set there is one of such quantiles. Are they averaged over all parameter

sets or are they estimated as double quantiles (quantiles from the set of quantiles)?

3. Line 344: I would suggest to put the figure A2 with the study region also in the main text.

4. Line 446: I think the bias is "highest" for the ranking method and not "lowest".

5. Figures 7-10: I assume the "blue" range is bounded by the infimum and supremum, here coming from the 0.05 and 0.95 quantiles, meaning only 90% of the possible range are cov-ered. What are the boundaries for the "grey" range? Is it covering 100%. May be this need to be indicated in the figure caption.

6. Limitations: This study uses sufficient long hourly discharge time series of 25 years for calibration on extremes. Often the hourly records are much shorter (e.g. 5 to 10 years) and a calibration on extremes is not feasible this way. Then, the calibration is done alternatively on observed flood statistics, for which often longer records are available, using synthetic rainfall as input. In this case the proposed procedure is hardly possible. Please discuss.

7. Appendix A. This appendix is not really necessary from my point of view.

---

## Author Comment (AC1) · 1 Sep 2020

We thank you for your positive feedback and your suggestions for improving our manuscript. Below we provide our replies (in *italic*) to the detailed comments of this reviewer and list the changes that will be made in the revised version of the manuscript (in blue).

**Referee's general comments:** The contribution provides an interesting approach to the selection of representative parameter sets for continuous hydrological modelling in the framework of derived flood frequency analyses considering uncertainty. The methodology is quite clear and plausible. The manuscript is well written and concise. I have only some minor comments for improvement (see detailed comments).

[Figure]

*Authors' Reply (AR): We thank reviewer 2 for the positive feedback on our manuscript.*

**Referee's detailed comments:**

1. Line 129: : : : "selected in step (d)" should read "selected in step (b)".
   *AR: this will be corrected in the text in the revised manuscript.*

2. Line 196: It is not clear to me how Q5, Q50 and Q95 are obtained? For each parameter set there is one of such quantiles. Are they averaged over all parameter sets or are they estimated as double quantiles (quantiles from the set of quantiles)?
   *AR: Q5, Q50 and Q95 are estimated for each simulation point of sorted annual maxima in the frequency space, i.e. over all parameter sets. Hence, quantiling is done on each point of the sorted annual maxima and not on the entire simulation resulting from a single parameter set. To clarify this issue, an additional text will be included in the revised manuscript:*
   "The 5%, 50% and 95% quantiles of these ensembles are computed at each j-th point in the frequency space, resulting in quantiles Q5, Q50 and Q95 over the entire simulation period. . ."

3. Line 344: I would suggest to put the figure A2 with the study region also in the main text.
   *AR: thank you for this suggestion. This figure will be moved into the main text in the revised manuscript.*

4. Line 446: I think the bias is "highest" for the ranking method and not "lowest".
   *AR: yes, this is correct. Thank you for spotting this typo!*

5. Figures 7-10: I assume the "blue" range is bounded by the infimum and supremum, here coming from the 0.05 and 0.95 quantiles, meaning only 90% of the possible range are covered. What are the boundaries for the "grey" range? Is it

covering 100%. May be this need to be indicated in the figure caption.

*AR: It is correct that the blue band is the coverage of the range bounded by the infimum and supremum parameter sets, i.e. 90% predictive intervals. The grey range corresponds to the band estimated using all 100 parameter sets. To clarify this issue, we will include additional explanation in the figure captions.*

6. Limitations: This study uses sufficient long hourly discharge time series of 25 years for calibration on extremes. Often the hourly records are much shorter (e.g. 5 to 10 years) and a calibration on extremes is not feasible this way. Then, the calibration is done alternatively on observed flood statistics, for which often longer records are available, using synthetic rainfall as input. In this case the proposed procedure is hardly possible. Please discuss.

*AR: we agree that we are in a lucky situation when the hydrologic model can be calibrated with a continuous time series of more than 20 years. Yet, if such long time series are not available, other calibration procedure could be used (e.g. based on signatures) or model parameters could be required from regionalization approaches. The way, the model is calibrated is not relevant for the selection methods as long as at least 100 parameter sets can be derived (and that can well represent rare floods). We will add the following text in the limitation section in the revised manuscript:*

"Despite the calibration of a hydrological model lies beyond the scope of this paper, it is assumed that (at least) 100 parameter sets of a hydrologic model can be made available for selecting the representative parameter sets. For that purpose, a hydrological model should be calibrated with observed data of a long enough record that covers rare floods so that rare floods could be realistically simulated. Note that for that purpose, a continuous hydrologic model does not necessarily require continuous calibration data and it could also be calibrated to discrete data (e.g. using hydrologic signatures (Kavetski et al. 2018)). If no observed data or only very short records are available, model parameters can be acquired

through regionalization approaches (see the work of Brunner et al. (2018a) for an overview of regionalization methods). The developed methods are of use for applications when a hydrologic model should be employed for simulations of rare floods. If the use of a hydrologic model is not possible, i.e. neither information for calibration nor sufficient information for parameter regionalization is available, these methods cannot be applied."

7. Appendix A. This appendix is not really necessary from my point of view.
   *AR: thank you for this suggestion. As both reviewers suggested to remove it, this appendix will be removed from the revised manuscript.*

**References**

Brunner, M. I., Furrer, R., Sikorska, A. E., Viviroli, D., Seibert, J., and Favre, A. C.: Synthetic design hydrographs for ungauged catchments: a comparison of regionalization methods, Stochastic Environmental Research and Risk Assessment, 32, 1993–2023, https://doi.org/10.1007/s00477-018-1523-3, 2018a.

Kavetski, D., Fenicia, F., Reichert, P., Albert, C.: Signature-domain calibration of hydrological models using approximate Bayesian computation: Theory and comparison to existing applications. Water Resources Research, 54, 4059–4083. https://doi.org/10.1002/2017WR020528, 2018.

---

## Author Comment (AC2) · 2 Sep 2020

We thank you for your generally very positive feedback and your detailed suggestions for improving our manuscript. Following your suggestions, we changed our manuscript at several places as described in detail in our responses to specific reviewer's comments. Most importantly, two new metrics will be included for comparing results of three selection methods; additional figures will be included in the Appendices that demonstrate the variability of the meteorological and hydrological scenarios, and demonstrate the performance of the hydrologic model in the calibration and validation periods. All figures will be redrawn to improve their readability and including missing information.

Below we provide our replies (*italic*) to all individual comments of this reviewer and list

the changes that will be made in the revised version of the manuscript (in blue).

**Referee's general comments:**
The paper is well written and properly structured. The scope of the study is clear, and the purpose of the selection of models among a given ensemble is relevant in the perspective of hydrological and hydraulic simulation for extreme flood estimation.
*Authors' Reply (AR): Thanks for the positive overall assessment.*

However, the presented methods, their scoring, and the interpretation of these scores deserve a better statistical assessment. For example, the presented scores, used in the comparison of the selection methods, rely only on the counts of simulated annual maxima being in or out of the predictive interval, without consideration of scale or frequency within their distribution.
*AR: Following this suggestion, we propose to include two additional metrics that explicitly consider the resulting prediction intervals. These are the relative band spread of predictive intervals ($R_{\Delta PIs}$) and the overlapping pools of the predictive intervals ($R_{OPPIs}$). The latter metric is an adaption of the evaluation metric suggested by the reviewer.*
*See also our detailed response to the reviewer's comment to line 333.*

No assessment of the width of the predictive intervals is given relative to a global metric (standard deviation of annual maxima of the calibration data for example). Even if a relative ranking of the three methods can be provided here, it is difficult to have a proper statistical characterization of them in terms of robustness and reliability, which can be an issue for using them for extreme flood evaluation.
*AR: The newly proposed metrics should provide a sound basis for a more objective comparison of the three selection methods developed here (for details see our reply to reviewer's comment to line 333). In addition, we propose to compute the latter metric i.e. the overlapping pools of the predictive intervals ($R_{OPPIs}$) in two ways.*

Interactive
comment
*First, looking at all annual maximas (AMs), and then by focusing only on most extreme floods, i.e. only for the upper tail of AMs in the frequency space. This should shed light on the reliability and the robustness of the methods for rare floods.*
*See also further our detailed reply to reviewer's comment to line 333.*

Furthermore, no interpretation of these selections in term of flood process or modelling is given. It is not supposed to be the core of this article, but it would help to connect the results to some hydro-climatological features and their impact in terms of variability.
*AR: As suggested by the reviewer, in the revised manuscript we will provide more details on the model calibration and validation as well as on the variability of the hydro-meterological scenarios.*
*See also our detailed replies to reviewer's comments to lines 374, 381 403.*

The problem of parameter equifinality is not evoked here, although it is the main factor of the parameters set variability, given that here, the 100 models have been calibrated by the same algorithm using the same data.
*AR: We thank the reviewer for this comment. Indeed, the equifinality issue is often a problem for any environmental model being difficult to overcome. Here, we propose a heuristic approach in which 100 parameter sets are derived from 100 independent calibration trails using a Genetic Algorithm, i.e., each parameter set comes from an independent model calibration but using the same dataset and the same calibration algorithm. By using independent model runs, the possibility of being trapped in the same local optimum should be reduced (at least to some extent). Albeit being not directly related, this heuristic method of sampling the parameter space is in line with the philosophy underlying ensemble forecasts (where model initial conditions are changed). It represents an interesting solution to systematic sampling of the posterior parameter distributions (e.g. via Markov Chain Monte Carlo Sampling) or to any Monte Carlo method relying on a very high number of model runs. This will be more explicitly*

*stated in the paper.*
*Despite the choice of 3 representative parameter sets for each case study will depend on the 100 sets available to be chosen from, the developed methodology will not depend on these available sets. Obviously different calibration methods could be here applied to derive these 100 parameter sets. We agree that it would be interesting to explore how such 100 representative optimal parameter sets should be chosen to cover different flow conditions, which would be however a completely different study than the one presented here.*
*We discuss this issue further in our detailed reply to reviewer's comments to lines 120 381.*

I would recommend then a major revision of this paper, in order to tackle with those main issues.
*AR: we thank the reviewer for this positive feedback and valuable suggestions. Our detailed responses to the issues raised above are given below in responses to detailed reviewer's comments.*

**Referee's detailed comments/questions:**
Quoted sentences are written in italic.

1. Line 10: *10'000 years of synthetic streamflow data simulated with a weather generator.* Simulated "thanks to" a weather would be more appropriate, the weather generator doesn't generates streamflow directly, it feds the hydrological models.
   *AR: this will be corrected in the revised manuscript.*

2. Line 12: *The methods are readily transferable to other situations where ensemble simulations are needed.* This is only evoked as a perspective at the end of the paper, without providing an example of such "other" application. I am not sure it deserves to be in the abstract.
   *AR: To better demonstrate the possible applications of our method, we will in-*

*clude an additional paragraph in the discussion:*
"Possible applications of these selection methods include all studies, where computational requirements are an issue, e.g., rare flood analysis in safety-studies concerning dams or bridge breaks, climate scenarios of these, evaluation of rare floods as due to changes in climatic variables using several emission scenarios and different uncertainty sources propagation. Finally, these methods could be used for quantifying different uncertainty source contributions in rare flood estimates but with little efforts from the hydrologic model as due to parametric uncertainty propagation."

3. Line 39: *initial conditions for use in combination with design storms.* Regarding the SCHADEX method detailed in (Paquet el al., 2013), I would add "initial conditions for use in combination with design or randomly drawn storms".
   *AR: Will be added.*

4. Line 43: *especially if long time series are to be simulated using ensembles of hydrological parameter sets.* And also if very high return times (above 1000 years) have to be robustly estimated, thus implying several thousand years of simulation.
   *AR: we will add this information in the text.*

5. Line 45: *extrapolating a synthetic design hydrograph.* How? By scaling up a synthetic hydrograph thanks to estimated extreme quantiles of peak and volume values?
   *AR: we will add the missing information as:*
   "by scaling up an estimated synthetic design hydrograph by quantiles of extreme peak and volume estimated using frequency analysis."

6. Line 51: *These continuous hydrologic simulation frameworks are still rare for time series >100 years.* 100 years of simulation merely allow to compute a robust 20 year return period estimation, which is pretty useless for dam safety for example.

I would rather say that "high computational power are needed in order to provide estimations for high to extreme return times (up to 1000 years) required for safety-related studies" (although this is almost written in the same terms in line 57).
*AR: We will shorten this sentence to:*
"These continuous hydrologic simulation frameworks are still rare for time series >100 years due to heavy computational requirements (Grimaldi et al. 2013)."
*And we will also include additional information as:*
"High computational power is particularly needed to provide estimations for high to extreme return periods (up to 1000 years and higher) required for safety-related studies or hydrologic hazard management. For such rare events,..."

7. Lines 77-86: The problem is particularly well stated here.
   *AR: Thanks for this comment.*

8. Line 89: *to select a reduced-size parameter ensemble for the use with a hydrological model within a continuous simulation.* Here and later on I would always keep "parameter" linked to "hydrological model". Most simulation frameworks are heavily parametrized, and the uncertainty linked to the hydrological model is only one (important) of the numerous sources of uncertainty. I would then write "to select a reduced size ensemble of hydrological model parameters for the use within a continuous simulation".
   *AR: We will correct the sentence as suggested.*

9. Line 92: *for simulation of extreme floods.* A recurrent formal remark about the word "extreme". Usually "extreme floods" refers to return times largely exceeding the observational range, currently more than 1000 years, thus being extrapolated (by FFA or simulation, or both). This is especially true in dam-safety related literature. In the presented case, the meteorological scenarios are 100 years long, meaning that only very few "extreme" floods are simulated in the whole experiment. At best a robust 1000 years estimation can be empirically inferred here

given the fact than 100x100=10 000 meteorological years have been simulated. The whole set of AM being extracted can surely not be considered as a set of "extreme floods". The authors could consider using "intense floods", "rare floods" or more simply "floods" when they refer to the simulated floods.
*AR: We propose to change the term 'extreme floods' into 'rare floods' throughout the manuscript and also in the title.*

10. Line 97: *simulated rare flood events.* Following the remarks above, the term "rare flood" is also appropriate alternative.
*AR: see our reply to the comment above.*

11. Line 99: *the aim is thus i) to provide long enough simulation periods for extreme flood analysis, ii) to avoid the propagation of errors due to data/model calibration etc. and iii) to be able to focus entirely on the uncertainty of the hydrological response.* I my opinion this goes farer than the actual results of the study. The uncertainty linked to model parameters is assessed, and properly summarized thanks to a reduced number of meteorological simulations. But I don't understand why this "provides long enough simulation", and why it "avoids the propagation of errors". If you have 100 "bad" models due to date, calibration, etc., three of them are selected in order to keep a good representation of their variability, but you still work with "bad" models (sorry for the term "bad", it only means "affected by uncertainty"!).
*AR: We use here synthetic data, instead of real observations, for deriving model simulations for testing the methods of selecting representative parameter sets of a hydrological model. This approach enabled us to generate long pseudo-observations for analysis of rare floods (100 meteorological scenarios x 100 years). Using such setting (instead of real observations that are usually of ~30 years at best at an hourly temporal scale) enabled us, first, to test the method-ology on rare events of return periods >100 years (instead of return periods of up to 20 years if real observations were used). Second, taking such a setting as*

*a start for our analysis, we focus here entirely on the parametric uncertainty of a hydrological model and do not infiltrate into other uncertainty sources, i.e. input or output uncertainties of calibration data. How these parameter sets are derived, and the way of model calibration may contribute to the model ability to simulate real events. Calibration of the hydrological model is, however, not the focus of this paper. Hence, our ensemble of 100 parameters sets is assumed to be the best representation of model parameters one can get. We agree, however, with the reviewer that the uncertainty related to model calibration and calibration data are partly included in the calibrated model parameter sets and will be included in the simulations we get with these sets.*

*To clarify these issues, we will modify our text into:*

"Using a simulation setting with synthetic data as a start for our analysis enables us i) to provide long enough simulation periods for rare flood analysis with return periods <100 years, and ii) to be able to focus entirely on the uncertainty of the hydrological response, while other uncertainty sources of a hydrologic model (due to model calibration) are not explicitly considered. Note that way the hydrologic model is calibrated lies outside of this paper scope."

12. Line 111: *and not the model uncertainty of a weather generator.* This is perhaps one of the main limit of the study. At line 69, it is written that *Arnaud et al. (2017) found that the uncertainty of the rainfall generator dominates the uncertainty in the simulated extreme flood quantiles.* This uncertainty will not be considered here, and I wonder how far the results exposed here would still be useful to deal with the weather generator uncertainty (which of course is not to be confused with the variability of the scenarios generated thanks a given set of parameters). A comparison of both uncertainties (model and weather generator), even basic, would have been welcomed here.

*AR: As mentioned in the response to the above comment and in the manuscript itself, the effect of weather generator uncertainty is not here considered. In this*

*work, we use a tested and verified set-up of the weather generator (Ervin et al. 2018, 2019) and ready computed meteorological scenarios for our test study. As already written in the original manuscript (lines 98-111):*

*"A meteorological scenario represents a single realisation from a stochastic weather generator with constant model parameters. These meteorological scenarios are equally likely model realisations that differ in the precipitation and temperature patterns, and together they represent the natural variability of the climate (and not the model uncertainty of a weather generator)".*

*Therefore, the assessment of the uncertainty due to the weather generator is not possible here. To assess the effect of the uncertainty of the weather generator on the simulated time series, uncertainty in parameters of the weather generator should be considered that would lead to very different meteorological scenarios from these that are here applied. We agree that such a study would be very interesting, but it exceeds the scope of this paper that is the selection of representative sets of hydrologic model parameters.*

13. Line 120: *(ii) the distribution is known.* I am not sure that knowing the probability of parameters is a reasonable perspective, in my opinion the problem of equifinality of parameters in models like HBV prevents an a priori expression of parameter probability, as different sets of parameters can "produce" the same model, i.e. models having the same behaviour for a given meteorological scenario. And this is one of the interesting outcome of this study, which focuses on the hydrological response of the models, and not on the actual values of the parameters. I think that this equifinality problem deserves more writing in this paper. *AR: Thank you for this comment. We refer here to the situation when the parameter distribution is estimated from the observation data, e.g. through Monte Carlo or Bayesian approaches. Hence, we will clarify that in the text of the revised manuscript as:*

"the distribution is known (i.e. estimated from data)..."

*We agree however that the issue of the parameter equifinality deserves more attention in our paper and thus we included an additional paragraph on that in the revised manuscript:*

"Using multiple parameter sets for a hydrological model is justified by the parameter equifinality..."

*And in discussion:*

"Downsizing the hydrological model parameter sample can only aim at understanding and characterizing the hydrological part of the full hydrological ensemble resulting from a combination of multiple parameter sets and multiple meteorological scenarios. The variability of hydrologic model parameters arises from the parameter equifinality (Beven and Freer 2001), and it can be overcome by using several hydrologic model parameter sets that should encompass the parametric and (implicitly) also other uncertainty sources. Our selection methods thus enable one to choose representative parameter sets from the hydrologic responses point of view and in this way to cover the variability of hydrologic responses with reduced hydrological model runs needed. These methods are however not applicable for characterizing the climate variability (nor for downsizing the number of meteorological scenarios needed)."

14. Line 129: *The infimum (from the Latin – smallest) and supremum (from the Latin – largest) refer to the greatest lower bound and the least upper bound (Hazewinkel, 1994), i.e., the largest interval bounding the ensemble from below and the smallest interval bounding it from above.* This definition deserves to be connected to frequencies of the target variable, even if it's not straightforward. Does it (roughly) provides a 90, 95 or 99% confidence interval of the simulated variable? Given that follows in lines 184 to 234, with quantiles 5 and 95%, it "looks like" a 90% CI. *AR: yes, the bounds bounded by the infimum and supremum set represents the 90% predictive intervals. This information will be added in the text.*

"Thus, the representative band should correspond to 90% predictive bands of a

target variable."

15. Line 138: *we thus propose to use the representation of AMs in the Gumbel space as the reference model response space for parameter selection.* The plotting in Gumbel is useful here to illustrate the rare to extreme quantiles. Still, it doesn't explicitly play a role in the "parameter selection" (which doesn't imply any explicit reference to an implicit Gumbel distribution of AM in the statistical criteria/indicators used).

*AR: this is a good remark. Indeed, the selection is based on the absolute values of the annual maxima without referring to their distribution but with accounting for their location in the list of maxima sorted by their magnitudes. The Gumbel space is used for plotting purposes and to visualize the selection method and results. We will correct this issue throughout the manuscript and in the sentence referred above as:*

"we thus propose to use the representation of AMs sorted by their magnitudes (i.e. frequency space) as the reference model response space for parameter selection."

16. Line 142: *inverse modelling approach.* The term "inverse modelling" appears to me somehow excessive. An inverse hydrologic modelling would be for example to infer rainfall from discharges. Here it's more a "post-modelling" approach.
*AR: we agree and will correct that in the revised manuscript.*

17. Line 145: *the parameter set selection is made based on the full hydrological simulation ensemble but using only a limited simulation period.* To be more specific I suggest to write "based on the simulation with all the hydrological models but using [: : :]".
*AR: we will correct this sentence to:*
"The main idea behind all three methods is that the hydrologic parameter set selection is made based on the full ensemble with all hydrological model simulations. . ."

18. Line 189: The parameter sets selected in step (d). Should be step (c).
    *AR: this will be corrected.*

19. Line 189: *the sets which are chosen most often as the 5th, 50th and 95th ranks
    are retained as the parameter sets [: : :] representative for the entire simulation
    period.* The ranking methods yet shows its weakness: the 5th and 95th of a given
    year have very low chance to match to the overall corresponding quantiles, given
    the "climate variability" illustrated in Figure 1, thus preventing the parameters se-
    lected on a given year to have a global representativeness. I am not sure it's
    worth keeping this method "in the game" for the rest of the paper.
    *AR: Thanks for this comment. We agree that the ranking method has its weak-
    nesses, as it also appears from our analysis. However, its great advantage is
    that it is straightforward to be performed. Thus, we prefer to keep this method for
    detailed analysis in the paper and reject it based on the results and not prior to
    that.*

20. Lines 194-206: I don't understand why the "Gumbel space" is evoked here (three
    times!), and constantly throughout the paper. Apart from the plots of Figure 2 and
    others, what is "Gumbel specific" in the metrics and statistics presented? For ex-
    ample, the RMSE scores are computed using each simulated annual maximum,
    regardless its empirical frequency.
    *AR: yes, we agree with the reviewer and will remove referring to the Gumbel
    space in selecting the representative parameter sets. We keep referring to the
    Gumbel space when it comes to visually presenting and comparing results.*

21. Line 201-205, equations 1-3: This is the same equation for the three considered
    quantiles. Only one is necessary.
    *AR: This is another very good point! These three equations will be combined into
    one equation.*

22. Line 211-234: same remark as above about the "Gumbel space".
    *AR: We will remove the reference to Gumbel space where it is not valid.*

23. Lines 214: *These members are next clustered in the Gumbel space into three representative groups (clusters) based on all J simulation years using the k-means clustering.* If I understood properly it means that the clustering has been performed in the J-dimensional space of the full set of members values?
    *AR: yes, this is correct. Clustering was computed on J-annual maxima.*

24. Line 217: *Next, these clusters are sorted by their magnitude.* What variable/quantile is used for this sorting?
    *AR: Sorting of clusters is done based on their means and their magnitudes for the quantile 90%. We have tested several quantiles in the upper range (i.e. 80%-99%), and the choice of quantile did not impact the ordering of clusters. To clarify that, we propose to add the following additional information in the revised manuscript:*
    "Next, these clusters are sorted based on cluster means by their magnitude by comparing percentiles in the upper tail of the distribution (here we used the 90th percentile). Use of a percentile from the upper tail is important as methods are focusing on rare floods. However, we found that the method was insensitive to the percentile choice as long as it lies in the upper tail (i.e. $\geq$80th percentile). Based on the percentiles computed for each cluster mean, the lower, middle and upper clusters are defined. Next, for the lower cluster a 5th percentile, for the upper – 95th percentile, and for the middle – 50th percentile are computed, i.e., $P_5$, $P_{50}$ and $P_{95}$..."

25. Line 218: *Note that we use here percentiles instead of cluster means to make this method comparable with the other two methods.* I am not sure of that : say that each cluster regroups one third of the ensembles, and for a given quantile in the AM distribution (say the 50%), it is evenly distributed through all the members, the percentile 5% of the lower cluster would more or less correspond to a 0.05x0.33 = 0.17 global percentile. The 5% and 95% are more "rare" than their corresponding quantile in the quantiling method.

*AR: we understand the reviewer's point. Yet, the clusters are rarely symmetric, i.e., there is no assumption that the clusters have to be symmetric. Hence, it is hard to define a more fair selection of intervals than the one we describe here, i.e. based on percentiles. Choosing 5% and 95% makes the clustering theoretically comparable to the other two methods. Having symmetric clusters would imply an idealistic situation when the responses of a hydrological model provide simulations of the annual peaks that are symmetrically spread around the median values, which is hardly realistic for real cases. This, however, does not have to be the case, as the choice of different parameter sets may lead to emphasize different hydrological processes in the catchment, i.e. floods resulting from snowmelts or intensive rainfalls.*

*We will add an additional sentence on that in the text of the revised manuscript:*
"Use of the 5th and 95th percentiles appears as a fair choice for asymmetrically spread clusters, which is most often the case as different parameter sets of a hydrologic model may emphasize different hydrologic processes in the catchment."

26. Line 223-227, equations 4-6: Same remark as for equations 1 to 3.
    *AR: We will combine equations 4-6 to a single equation in the revised manuscript.*

27. Line 233, equation 7: This mention of the plotting position mention could be moved at line 195.
    *AR: as we will remove the reference to the Gumbel space in line 195, we keep this equation at its original place. However, we will add additional text to link the method to plotting results in the Gumbel space:*
    "For visualizing the selection methods, we use the Gumbel space (Generalized Extreme Value distribution Type-I) with the Gringorten's method (Gringorten 1963) to compute the plotting positions of AMs in the Gumbel plots:..."

28. Table 1: *Sorting space = Gumbel space.* Once again, I don't undertand how "Gumbel specific" the sorting process is for Quantiling and Clustering.
*AR: we will correct 'Gumbel space' into annual maxima frequency when referring to the sorting space in the table.*

29. Table 1: *Interpretation of pred. intervals / Parameter grouping.* I don't see to what these lines refer in the text before.
*AR: We will include additional information in the text that refers to the interpretation of predictive intervals contained in the table. Following sentence regarding the ranking method will be added:*
"The derived predictive intervals thus are sensitive to individual years of simulations, and their interpretation may be difficult (as they do not result from any flow frequency analysis)."
*And regarding the other two methods:*
"This enables statistical statements to be made about the selected parameter sets and about the predictive intervals constructed with the help of these parameter sets (as they are constructed on the entire simulation ensemble)."
*And regarding the parameter grouping:*
"Finally, the clustering method splits all ensemble members (hydrologic simulations) into three clusters, and so each parameter set can be attributed into corresponding clusters. This could be useful if one would like to extract more information on each cluster behaviour."

30. Line 254: *assessing how well the reduced ensembles cover the reference simulation ensemble.* More specifically I would rather say "how well the reduced ensembles substitute the whole simulation ensemble for the selection of representative parameter sets".
*AR: This will be corrected accordingly.*

31. Line 273: *assess how well the defined identified intervals represent the ensem-*

*ble members of this Sr meteorological scenario.* What metric is used to do this assessment?
*AR: All assessment criteria are described in sect. 2.6. For clarity, we will include in this sentence link to this section.*

32. Line 288: *Compute the 5th percentile [: : :] and the 95th for H($\theta_{sup,p}$/Sm) .* The mention "for m=1,2: : :,M and m$\neq$p" could be added for more clarity.
    *AR: will be added.*

33. Line 290: same question as for line 273.
    *AR: see the reply to the question for line 273.*

34. Line 295 and below: as evoked above, the recurrent mention to the Gumbel space is, in my opinion, useless, and over time tedious to read. In the paper, it is quickly implicit that the plots and the metrics used are defined in the frequency space (or frequency domain), being Gumbel or not, without need to repeat it.
    *AR: we agree with the reviewer and will modify this text which should read now:*
    "The simplest way of assessing the behavior of these three methods is a visual inspection of curves plotted in the frequency space (e.g. using Gumbel distribution for plotting). . ."

35. Line 330: *Here we propose to use different percentiles, i.e., the 5th, 50th, and 95th percentiles, to characterize the ratio of the simulation points lying outside the computed predictive intervals for each of the methods.* I don't understand this? Why not using only the 50th percentiles of this ratio? Refer to comments on Table 3 for a more detailed version of this question.
    *AR: yes, this is a good suggestion. We will remove the 5th and 95th percentiles and keep only the 50th percentile for comparing methods. This should correspond to the situation with the majority of simulation points or scenarios lying inside/outside the bands.*

36. Line 333: *how many out of J hydrological simulation points [: :] must lie outside the defined predictive intervals.* I think that the problem of such a simple "count" of points (simulated annual maximum) outside the predictive interval doesn't take into account their position in the simulated distribution. As written in the title, the methods exposed here are supposed to be used in the estimation of extreme floods, which in any post-treatment of the hydrological simulation will strongly rely on the high simulated quantiles. The scores should somehow reflect this focus on high quantiles, which is not the case here. Instead of this count of "outside points", the area outside the predictive interval could be computed, using the Gumbel variable (as $x$) and the discharge value (as $y$), thus giving a contrasted score in which lying outside the predictive interval for high quantiles is more important than for low values.

*AR: We thank the reviewer for this very useful comment! In the revised manuscript, we propose to include two additional metrics that should put more focus on extreme floods. The newly proposed metrics are the relative band spread of predictive intervals ($R_{\Delta PIs}$) and the overlapping pools of the predictive intervals ($R_{OPPIs}$). The latter metric is an adaption of the suggested by the reviewer computation of pools lying outside PIs. These new metrics will be described in text in sect. 2.6.2, i.e. Quantitative assessment:*

"[(IV)] Relative band spread of PIs ($R_{\Delta PIs}$) that is computed for both tests and which compares the spread of PIs constructed with the representative parameter sets versus 90%-PIs of the full hydrologic ensemble. In details, $R_{\Delta PIs}$ is computed for each m-th scenario as:

New Eq. 13:

$$R_{\Delta PIs,m} = \sum_{j}^{J} \frac{S_{PIs,repr.,m}}{S_{PIs,full,m}}$$

m$= 1, 2, ...,$M   (1)

where SPIs,repr.,m and SPIs,full,m are band spreads of the 90%-PIs constructed with the representative parameter sets and with the full hydrologic ensemble. The band spread is computed as a difference between the upper (or supremum) and the lower (or infimum) interval at each j-th simulation point in the frequency space. [(V)] Overlapping pools of PIs $R_{OPPIs}$ that are computed for both tests in the

frequency space by taking the Gumbel variate and discharge values of sorted AMs as coordinates of the PI pools. In details, $R_{\text{OPPIs}}$ of PIs constructed with the representative parameter sets is computed for each m-th scenario as: New Eq. 14:

$$R_{\text{OPPIs},m} = \sum_{j}^{J} \frac{(k_j - k_{j-1})}{2}[H(\theta_{\text{sup},m,j}) + H(\theta_{\text{sup},m,j-1}) - H(\theta_{\text{inf},m,j}) - H(\theta_{\text{inf},m,j-1})]$$

m$= 1, 2, ...,$M&j$= 2, 3, ...,$J   (2)

Similarly, $R_{\text{OPPIs}}$ is computed for the full hydrologic ensemble using the pool restricted by the 90%-PIs, i.e. taking the 5% and 95% intervals as pool borders.

With regards to $R_{\Delta\text{PIs}}$, we propose to compute the relative band spread as a mean overall sorted AMs at first. Also, to focus more on rare floods, we propose to compute the means of rare floods limited by different Gumbel variates. Here we computed $R_{\Delta\text{PIs}}$ for the upper half of AMs ($R_{\Delta\text{PIs,j}} \geq 51$), for the most upper ten AMs ($R_{\Delta\text{PIs,j}} \geq 91$) and the most upper five AMs ($R_{\Delta\text{PIs,j}} \geq 96$).".

*These metrics are computed for both the cross-validation and multi-scenario evaluation tests and the medians over all scenarios will be summarized in the table 3.*

"Analysis of overlapping PIs pools ($R_{\text{OPPIs}}$) and relative band spreads ($R_{\Delta\text{PIs}}$) show that in the cross-validation test, all methods provide bands that are wider than the 90%-PIs computed using the full simulation ensemble. This should not surprise as the selection of relative parameter sets is based on a large sample of hydrologic model realisations (i.e. 99 scenarios) than the full ensemble for model assessment (i.e. single scenario). However, these metrics show large differences in the multi-scenario test, in which the clustering method outperforms other two selection methods, particularly when the focus lies on rare floods (compare $R_{\Delta\text{PIs,j}} \geq 51$, $R_{\Delta\text{PIs,j}} \geq 91$ and $R_{\Delta\text{PIs,j}} \geq 96$ in Table 3). The quantiling was the second good method, while the ranking was performing the worst. These observations are also confirmed when looking at the variability of these two met-

rics for different return periods (Fig.10). A better performance of the clustering method can be again noticed in the range of rare floods. While quantiling was performing worse than clustering, it was still better than the ranking method."

*In addition, we will introduce an additional figure which presents cumulative results of these two metrics for all scenarios (as a new Figure 10, Fig. 1 in this reposne).*

New Figure 10. Evaluation of the leave-one-out cross-validation and the multi-scenario test for the three selection methods using the relative band spread ($R_{\Delta\text{PIs}}$) and the relative overlapping pools ($R_{\text{OPPIs}}$), both computed with reference to the 90%-PIs of the full hydrologic simulation ensemble.

37. Line 337: *In this work, we consider the following values for $r_{thr} = 0.50, 0.25, 0.10, 0.05$.* Following the preceding comment, a metric accounting for the scale or the frequency of the points being outside the predictive interval would avoid to distinguish such thresholds, which apart from the $r_{\text{thr}} = 0.50$ or $0.10$ have little statistical meaning in this context.

    *AR: We appreciate this comment and will reduce the thresholds to only two values, i.e. 0.50 and 0.10.*

38. Line 343: *For testing the methods developed here, a small natural catchment is preferable.* Why small?

    *AR: The choice of a small catchment is driven by the fact that hydrological responses in a small catchment may be better understood and are rather unaffected by anthropogenic factors (such as dams, bridges, etc.) which may perturb the flood peaks in terms of very rare floods. We will include additional information in the text in the revised manuscript:*

    "For testing the methods developed here, a small close to natural catchment is preferable, i.e. with only little anthropogenic influences, in which hydrological responses are transparent, and the generation of rare flood (peaks) is not affected by human constructions (dams, bridges)."
39. Line 358: *In this study, the version HBV light [: : :] with 15 calibrated parameters is used.* The considered model can be then considered as heavily parametrized, and thus fully affected by the equifinilaty problem of its parameter evoked in the remark made for line 120.

*AR: We agree that the model can be affected to some extent by the equifinality problem. However, the HBV is a conceptual model that covers different modes of the runoff generation accounting for snowmelt, soil moisture and groundwater. To correctly represent these processes and to provide hydrologically interpretable parameters, the parameters of individual modes should be included in the calibration process. Hence, we used here such a set-up with 15 calibrated parameter sets, which was previously successfully applied in Swiss catchments. We will add a following sentence to the text:*

"Such a set-up of the HBV light was previously successfully applied in Swiss catchments (e.g. Sikorska & Seibert2018a, Brunner et al. 2018c, Brunner & Sikorska-Senoner 2019, Müller-Thomy & Sikorska-Senoner 2019, Westerberg et al. 2020)."

40. Line 374: *for details on $R_{PEAK}$ and $R_{MARE}$, see the work of Vis et al. (2015).* A brief description of $R_{\mathrm{PEAK}}$ and $R_{\mathrm{MARE}}$ would be welcomed here, especially as within the calibration process, the parameters conditioning the modelling of floods are surely strongly conditioned by the $R_{\mathrm{PEAK}}$ score.

*AR: with response to this comment, we propose to add additional information in the major text and equations used for computing $R_{KGE}$, $R_{PEAK}$, $R_{MARE}$ in the newly created Appendix D. The new text reads as:*

"$R_{\mathrm{PEAK}}$ is defined in a similar way to the Nash-Sutcliffe efficiency but using peak flows instead of the entire time series. While both $R_{\mathrm{KGE}}$ and $R_{\mathrm{PEAK}}$ focus on high (peak) flows, $R_{\mathrm{MARE}}$ is sensitive to low flows. See Appendix D for equations."

41. Line 377: *The available observational datasets are split into a calibration (1990-*

*2005 years) and a validation (2006-2014 years) period.* What is the point of having a validation period here? This validation period is never used in that follows.

*AR: Splitting into calibration and validation period is needed to assess how well the calibrated model performs outside the calibration period. This assessment is important, as the paper focus lies here on simulations that are not included in the calibration period. We will add a following sentence into the text:*

"Evaluation of the model in the independent period is important as the model is applied to simulate time series outside the calibration period."

42. Line 381: *The calibration is repeated 100 times resulting in 100 independent optimal parameter sets.* I am surprised by the variability of the parameters obtained by these 100 calibration runs, performed on the same calibration data with the same objective function. I would like to read a comment from the authors on that. Mine is that the optimization is not complete, seeming to depend on the aleatory exploration performed by the genetic algorithm, somehow "trapped" in local optimums, and/or affected by a strong equifinality problem (yes, once again, sorry). An alternative strategy for the generation of model parameter sets, in my opinion providing more "independent" models, could be to bootstrap 12 years among the 24 years available in order to generate 100 truly different calibration & validation samples.

*AR: We agree that a global optimizer should always lead to the same optimized parameter set except for specific cases where the response surface (the objective function) clearly has several optima with exactly the same value (which can happen in theoretical settings, e.g. with the so-called Himmelblau function). In settings where the response surface has many local optima with very similar values (equifinality), the start point of the search can indeed lead to a trapping in local optima, especially if the optimizer is stopped after a fixed number of iterations as is the case here. A solution to overcome this problem is classical Markov Chain*

*Monte Carlo (MCMC) As we aimed here at developing the parameter selection method that could be applied also for selecting parameter sets from independent model calibrations, MCMC was not a desirable solution here. As a heuristic solution, we propose here multiple independent trials using a genetic algorithm to derive 100 sets of good model parameter sets. This is an approach which we have in numerous studies recently (e.g., Seibert and Vis, 2016, van Meerveld et al., 2018; Etter et al., 2020).*

*Note however that the method used for calibration of the model and the selection of 100 'best' parameter sets was not the objective of this paper. Obviously, there are other methods to obtain 100 parameter sets (or more) are possible such as based on Monte Carlo methods or by bootstrapping as suggested by the reviewer, but the approach of independent calibration trials has been found suitable in these previous studies. Additional text will be included in the revised manuscript:*

"These 100 parameter sets represent similarly likely parameterisations of the hydrological functioning of this catchment and their variation can be explained by the equifinality of hydrologic model parameters (Beven & Freer 2001)."
*And:*
"Note that the described above way to derive 100 parameter sets is one possible approach, and other calibration methods could be used (e.g. Monte Carlo or bootstrapping)."

43. Line 381: *The median model efficiency measured with Fobj over all 100 runs is 0.7.* To better assess the quality and the variability of the models generated at this step, it would be useful to show the distribution of NSE (Nash & Sutcliffe Efficiency) for both calibration  validation, and the ensemble plots of daily regime and classified discharge distribution for all the generated models. The ensemble simulation of the biggest observed would also be very pedagogic.
*AR: The Nash-Sutcliffe values for the calibration and validation periods will be*

*provided in the revised manuscript in newly included Appendix: Model calibration
results. In addition, flow duration curves will be provided for mean daily dis-
charges for simulated versus observed values and these will be also included in
the same appendix.*

New Figure A2. Flow duration curves and model performance metrics for calibra-
tion and validation periods over all 100 optimized parameter sets.

*Also, the following sentence will be added in the main text:*

"Also, diagnostics of the Nash-Sutcliffe efficiency and the Peak efficiency demon-
strate that the model performs well in the range of high flows which are most
important for simulation of rare floods studied in this paper (see Fig. A2 in Ap-
pendix C)."

44. Line 383: *which can be assumed to be a good model performance on an hourly
scale.* As mentioned above, this really need to be illustrated more richly.
*AR: please see our reply to the above comment.*

45. Line 397: *The daily values generated with $GWEX_{Disag}$ were then disaggregated
to hourly values using the meteorological analogues method.* More details would
be welcomed on that disaggregation: what fields/variable are used for analogy,
what analogy criterion, what about seasonality (i.e. are the analogues identified
within period of the year similar to the one of the simulation to be disaggregated,
etc.).
*AR: The disaggregation with the weather generator is not the scope of this paper,
and it was already published by Evin et al. (2018, 2019). Thus, we refer to the
source papers on this disaggregation scheme. Here we use the ready meteoro-
logical scenarios that were made available for this catchment. These scenarios
were developed for the entire Aare catchment, and here we only use a small
subset for a single catchment.*

46. Line 399: *Next, catchment means were computed using the Thiessen polygon method.* On how many simulated precipitation stations do the Thiessen average rely on for the considered catchment? How many simulated stations lie within the catchment?
*AR: 3 stations located close by were used for that purpose. This information will be added in the text.*

47. Line 403: *Thus, differences between scenarios are exclusively due to the natural variability of the meteorological time series. "[: : :] and modelled by the GWEX weather generator" could be added.* Similarly to the models, the variability of these scenarios deserve to be illustrated, and compared to the observations, e.g. thanks to their average and standard deviation of the annual maximum daily precipitations.
*AR: we will add the sentence as suggested.*
*In addition, the variability of meteorological and resulting hydrologic scenarios is presented in a newly introduced figure in the new Appendix: Scenarios variability and briefly described in the appendix.*
New Figure A3. Variability of 100 meteorological scenarios used in this study vs. observations.
New Figure A4. Variability of 100 hydrologic scenarios used in this study; left panel – hydrologic ensemble with all meteorological scenarios and all hydrologic model parameters; right panel – hydrologic ensemble with all hydrologic model parameters but for the median meteorological scenario only. PIs represent the 90% predictive intervals.

48. Line 406: *These 100 meteorological scenarios are used as input into the HBV model to generate streamflow time series with 100 different HBV parameter sets.* I am not sure that this sentence is useful. The simulation scheme is clear from the beginning.
*AR: we prefer to keep this sentence for clarity and the reading flow.*

49. Figure 4: The title of the second plot should be "Quantiling" instead of Quantailing.

    *AR: we will remove this plot from the manuscript.*

50. Lines 419-434: I find the results of this paragraph difficult to interpret. Some violin plots show odd parameter selection patterns (like in Clustering/Infimum), other show weak parameter discrimination (Quantiling/Median). The Table 2 is quite difficult to read/interpret with so many counts exposed. In this paragraph and in the following ones, some "illustrations" of the most selected parameter sets should be provided, e.g. by presenting the range of hydrological responses to observed meteorological data of the selected models compared to the full ensemble. In other words, some interpretation in term of modelling and hydrological processes would be welcomed.

    *AR: we will remove the violin plots, and instead, we will include an alternative figure that presents a grouping of representative parameters selected with three methods. This enables us to look at how the selection of parameters corresponds to different processes being modelled by the model.*

    New Figure 5. Box-plots showing the variability of the hydrologic parameter sets selected as the representative parameter sets over 100 meteorological scenarios chosen with three methods. The white box-plots illustrate the entire parameter ensemble (i.e. 100 sets), outliers are not presented. I - infimum, M - median and S - supremum set. Units as in Table A1. The blue box surrounds parameters from the response routine, the grey box from the snow routine and the yellow from the soil moisture routine. MAXBAS is the only parameter from the routing routine, and CET is a potential evaporation correction factor.

    *Following the new figure, a new paragraph will be included:*

    "The variability of selected parameter sets is presented in Fig.5. As can be seen from the figure, some parameters presented smaller and others larger variability of selected sets. It also appears that different values are selected for the infimum, median and supremum set but not always. Among three selection methods, the ranking method (marked in green) has the largest spread of parameter values for most of the parameters. The clustering (blue) and quantiling (yellow) selection methods seem to choose more extreme parameter values for both, i.e. infimum and supremum sets. Looking at different model routines and hydrological processes behind, no clear patterns could be seen regarding the choice of parameter sets. It appears however that the representative parameters from the response (blue) and soil moisture (yellow) routines have a smaller spread than those from the snow routine (grey), as they are more often outside and further away from the interquartile ranges (grey boxplots)."

*We will also reduce the table to show only the three most selected parameters (instead of 5).*

51. Line 434: *Interestingly, for the supremum set in the clustering method, only four parameter sets among all 100 available are chosen over all 100 scenarios.* Given that, I don't understand why in Table 2, column Clustering, 5 parameters sets (34, 22, 98, 86, 50) are identified.
    *AR: thank you for spotting this typo! It should be written for infimum set and not for supremum set.*

52. Table 2: a graphical alternative or a complement to that table deserves to be presented, to better assess the "density" of parameter sets selected by the different methods.
    *AR: see our response to the comment to lines 419-434.*

53. Line 437: *intervals for extreme flood predictions.* The term "extreme flood estimations" could be more appropriate.
    *AR: this will be corrected.*

54. Line 441: *According to a first visual assessment, these three methods lead to slightly different constructed frequency intervals particularly in the upper tail of*

*the distribution.* To ease this visual assessment, horizontal lines marking these intervals for the upper values could be added to the plots of Figure 5.
*AR: Horizontal and vertical lines will be added to all frequency plots.*

55. Line 446: *the three intervals are always correctly attributed.* I would temper this in writing that "the three intervals are always correctly ordered" as this exactly what it is measured in Rbias.
*AR: yes, it will be corrected.*

56. Line 456: From the visual assessment, it is difficult to judge the methods. See remark on Figures 7-8. Figure 5 to 8: Instead of having an x-axis graduated with the Gumbel variable U, some ticks at remarkable return times (2, 5, 10, 20, 50 & 100 years) could be added in order to ease the reading of these plots, and avoid the long caption The Gumbel variates etc. in Figure 4.
*AR: Information on return periods will be included in all figures.*

57. Line 463: *the highest values for both evaluation criteria, i.e., the median ratio of simulation points lying outside the predictive intervals (Rspo) and the median ratio of hydrological simulation ensemble [: : :].* Given the definitions of s.2.6.2, this is more a mean ratio than a median ratio.
*AR: It is a mean for each scenario but median over all scenarios. To clarify that we will include the additional text:*
"... ,both presented as median values over all scenarios."

58. Line 475: *Hence, again here all three method can be qualified as behaving well based on the multi-scenario evaluation, and only the order of their behavior can be established.* Honestly, at the end of this paragraph, I have no clear idea of the absolute performance of each method. One important point is that the methods provide rather different intervals (like illustrated in the Figure 6), thus a method providing wide intervals will have good "in/out" scores (like the ones in Table 3), better than for narrow intervals, but the question of the statistical relevance of

such intervals is not solved.

*AR: We will add an additional paragraph at the end of the result section that summarizes major findings:*

"As it appears from the above, the rejection or acceptance of one of three methods tested here is not straightforward. Apart from the ranking method, which was linked with a huge bias, both other methods, i.e. quantiling and clustering were performing similarly well. Yet, these methods provide quite different intervals (of a different spread). The validity and usefulness of these methods for selecting the representative parameter sets are thus further discussed below in sect. 5.1. The detailed analysis of the relative band spread and the overlapping pools indicated however that the clustering method was performing the best particularly in the range of rare floods. The quantiling method was scored as the second best, while the ranking method was performing poorest."

59. Figure 7-8: I found the plots of the top panels of both figures rather counterintuitive: *the prediction interval resulting from selecting representative parameter sets for 99 meteorological scenarios and compared to the full simulated range with all 100 parameter sets* seems narrower than "statistically expected" (more and less a 5-95% confidence interval given the quantiles or percentiles involved in the process). For the highest simulated quantiles, the prediction interval seems only to cover about 50 to 66% of their variability. In the bottom plot of the Figure 7 for clustering, a second blue interval is plotted without being identified in the legend nor in the caption.

*AR: This second blue interval is indeed the grey interval, which comes from the scenario assessed here. For clarity, we redesigned these plots by removing the upper panel, which might have been misleading, and by improving the readability of the bands.*

New Figure 9. Example of multi-scenario evaluation for the three selection methods and two meteorological scenarios. PIs represent the 90% predictive intervals.

60. Line 480: [: : :] *selecting representative parameter sets that yield reliable predictive intervals in the frequency domain.* Following the comment on line 475, I see no statistical demonstration of the reliability of the predictive interval (like the one that could be done by controlled random generation of a given variable to which a statistical test is applied, then a proper statistical scoring). I agree with the authors on that a ranking between the three methods (according to the presented scores) is however established.
*AR: Two additional metrics giving more focus to extreme floods will be included in the revised manuscript. See also our above reply to the reviewer comment to line 333.*

61. Table 3: Three quantiles of the Rspo score are given, although the caption mentions that *the values represent the median values over all 100 scenario runs. What for providing the 5th and 95th quantile of a score measuring the ratio of simulation points [: : :] lying outside the predictive intervals* (line 312), which should be, on average, close to 10% (once again given the quantiles involved in the selection process)? In the low part of the Table 3, the Metric method is written as Rhso (Rmso). Which scores are the ones provided?
*AR: The Rspo is now presented only for 50th percentile. Regarding $R_{hso}$ & $R_{mso}$, both metrics are provided in table 3 as $R_{hso}$ is used in the cross-validation test and $R_{mso}$ in the multi-scenario test. This will be clarified in the new table 3.*

62. Line 481: *all three methods are fit-for-purpose for extreme flood simulation.* Following the preceding comment, if the presented method cannot be statistically demonstrated, it can be considered as an ad-hoc heuristic, build for a given purpose, here extreme flood simulation/estimation. This last step is not evoked in the paper, then depriving the reader from assessing the relevancy/robustness of this heuristic.
*AR: two new metrics that will be introduced (relative band spread of PIs and their overlapping pools) should provide better evaluation and comparison of the*

[Figure]

*proposed selection methods. See also our reply to the comment to line 333.*

63. Line 487: *for additional ease of use criteria.* I don't understand this sentence.
*AR: We will correct the sentence as follows:*
"To further compare the methods, we provide below a detailed discussion of the major differences and present a synthesis of how the methods rank on average (averaged across all scenarios) for the quantitative evaluation criteria, which we support with further qualitative evaluation criteria (Table 4)."

64. Line 488: *From the visual assessment.* Based on which figure?
*AR: we will add additional explanation here:*
"From the visual assessment, i.e. based on the method bias $(R_{\text{bias}})$,..."

65. Table 4: The different ranking features should be linked to the scores presented in Table 3. Some of them deserve to be better explained in the text: *Independence from meteorological scenario, Independence from simulation years, Ease in application, Interpretation of prediction intervals.* They don't refer explicitly to scores, statistics or plots presented before.
*AR: More explanation will be provided on these metrics in the text:*
"We hence introduce here a criterion independence from meteorological scenario, which defines how strong the selected sets depend on the meteorological scenario used for selection of representative parameter sets.
In a similar way, independence from simulation years will define how strong the selected sets depend on the simulation years used for selection of the representative parameter sets. To make statements on that, one needs to recall how the selection methods are constructed:..."
*And:*
"Nevertheless, the ranking method can be considered as the (computationally and methodologically) easiest in application due to its selection criteria relying purely on ranking within individual simulation years. We call this criterion as ease

in application. The other two methods need to be performed in the frequency space on sorted annual maxima over the entire simulation period and, in the case of the clustering method, require some additional computational effort (which remains low, however, compared to the hydrologic simulation).

The use of the frequency space in selecting the representative parameter sets helps, however, to interpret the constructed prediction intervals and to directly assign return periods to them. This speaks for their higher interpretability of prediction intervals as compared to the ranking method, in which interpretation of intervals is very limited (as they are selected without any flow frequency analysis)."

66. Line 497-502: This lines could be put in section 2.5 to describe the assessment of the approach better.
*AR: as suggested, we will move these sentences to the method section.*

67. Line 514: *The other two methods need to be performed in the Gumbel space over the entire simulation period and, in the case of the clustering method, require some additional computational effort.* Once again this reference to "Gumbel space" is unappropriated given the scores computed, and the additional computational effort doesn't seem significant, completely justified by the added robustness.
*AR: we will correct it to the term 'frequency space'.*

68. Line 516: *The use of the Gumbel space in selecting the representative parameter sets helps, however, to interpret the constructed prediction intervals and to directly assign return periods to them.* Same remark as for line 295.
*AR: will be corrected to the term 'frequency space'.*

69. Lines 522-534: These lines are, in my opinion, a short summary of the study, and do not fit in this section (Limitations and perspectives).

*AR: Lines 522-529 will be removed, while lines 529-534 will be moved into the sect. 3.5., in which the set-up of the study is presented.*

70. Line 541: *This use of synthetic data makes the approach results independent from the catchment properties and limits the effect of the hydrological model error and errors in calibration data on the methods comparison results.* I may be more cautious on that, given that the scores and the ranking of the methods are somehow linked to 1) the variability of the ensemble models, which depends on the equifinality of the model's parameters, the calibration data and FOs, etc. and to 2) the meteorological variability of intense events generated by the weather generator which depends on the climatology, the scale etc. Only some tests on different catchments (in scale and climatology) could ground this assertion.
*AR: we thank for this comment. We suggest rewriting this paragraph as follows:* "We chose to use synthetic instead of real observed data to work with long enough continuous simulations that cover rare events and to minimize the focus of the model error arising from the calibration data and procedure. By using synthetic data as a reference (instead of observed data), the latter error can be here neglected. The proposed methods should be tested with more catchments and other models to verify the scoring of methods that was achieved in this study."

71. Line 544: *We can, however, not directly assess here how much variability in the full hydrological ensemble is due to the climate variability and how much is due to the uncertainty resulting from the hydrological model parameters, because these two components are not linearly additive.* A simple exercise could help by 1) choosing a "median" model (in term of median response on the meteorological ensemble) and plotting/scoring the variability of simulations for the whole meteorological ensemble, and 2) choosing similarly a median meteorological scenario and simulating with the whole set of models and then 3) comparing the spread/variance of definite quantiles in the simulations. In my opinion, this is an indispensable complement to the presented results.

*AR: we thank the reviewer for this comment which showed us that we were not clear here about the purpose of our study. Separation of uncertainty intervals is not the aim of this work and the developed methodology for providing the uncertainty intervals based on representative sets. The selection methods should one enable to construct the uncertainty intervals based on three pre-selected parameter sets of a hydrological model. We use here different meteorological scenarios to verify how the selection criteria depend on a meteorological scenario and whether it is valid for different scenarios (independent meteorological conditions). Having such representative parameter sets of a hydrologic model selected, opens several avenues for further research. One example would be to separate uncertainty into contributing sources, i.e. hydrologic model, natural climatological variability and others. As contributions of such a separation will be case-specific – i.e. they depend on the selected hydrologic model, available meteorological scenarios, etc. – thus, they must be performed for each case study independently. We thank for suggestions on how such an analysis could be performed and these suggestions will be included in the revised manuscript. We do not see however, a need to perform such decomposition in our case, as our paper presents only the methodology for deriving representative parameter sets of a hydrologic model.*

*To clarify this issue, we will modify our text, which the reviewer is referring to into:*

*"Selection methods proposed in this study enables one to choose representative parameter sets of a hydrologic model and based on those to construct predictive uncertainty intervals (PI) for extreme flood analysis in the frequency space. Here, we tested the methodology using 100 meteorological scenarios that should represent the natural climate variability, and in this way, should provide independent conditions for methods' evaluation. Such a method for constructing PI from a hydrological model ensemble is a powerful tool that opens several avenues for further detailed uncertainty analysis. For instance, one may be interested in contributions of different uncertainty sources into the total PI constructed, e.g.*

coming from the hydrologic model or the natural climate variability. As these two components are not linearly additive, their separation is not straightforward. Also, any ensemble simulation also encompasses other uncertainty sources of the modeling chain, such as resulting from the weather generator or from the structure of the selected hydrological model, from the prediction of very rare flood events, etc. (Lamb & Kay 2004, Schumann et al. 2010, Kundzewicz et al. 2017). To assess individual contributions of interest, a simple sensitivity analysis based on the variance variability could be here recommended, in which one uncertainty source is propagated through the method at once while other sources are kept at their mode or median values and by comparing resulting PI spread."

72. Line 560: *Thus, the proposed selection methods could potentially be extended to account for different flood types.* Another option could be to consider Peak-Over-Threshold selection instead of a block selection (annual maximum) in building the simulated distributions. If different flood processes are present above a certain intensity threshold, flood type and seasonality sampling will be relevant.
*AR: we agree and will add a following sentence in the text:*
"For that purpose, Peak-Over-Threshold (POT) selection criteria of flood peaks could be more appropriate, instead of a block selection (annual maximum) used here, in constructing the simulated distributions of hydrological responses, to cover a range of different flood processes."

73. Line 566: *the three sets emulate the common practice of communicating median values along with prediction limits.* But in that case, these predictive intervals have to be statistically calibrated (or checked) in order to be used in safety studies, especially if these studies lead to engineering or compliance check.
*AR: we agree and will add:*
"For safety-studies, these representative intervals should be additionally statistically proved."

74. Line 572-584: Here again, the term Gumbel space could be replaced by "frequency domain" or equivalent.
*AR: this will be changed into 'frequency domain'.*

75. Appendix A  Figure A1: Interesting but rather off-topic here, as only continuous hydrological simulation has been used in this study.
*AR: as suggested, we will remove the Appendix A and Fig. A1 from the revised manuscript.*

76. Figure A3: For a better assessment of the distribution of calibrated parameters, I suggest that the scale of the horizontal axis of the violin plots (parameter values) exactly matches the corresponding calibration range written in the Table A1.
*AR: The figure will be updated with initial calibration ranges as suggested.*
New Figure A1. Violin plots (blue) summarizing 100 optimized parameter sets of the HBV model for the Dünnern at Olten catchment vs. initial calibration ranges (gray). Units as in Table A1.

**References**

Beven, K. and Freer, J.: Equifinality, data assimilation, and uncertainty estimation in mechanistic modelling of complex environmental systems using the GLUE methodology, Journal of Hydrology, 249, 11 – 29, https://doi.org/https://doi.org/10.1016/S0022-1694(01)00421-8, 2001.

Brunner, M. I. and Sikorska-Senoner, A. E.: Dependence of flood peaks and volumes in modeled discharge time series: Effect of different uncertainty sources, Journal of Hydrology, 572, 620 – 629, https://doi.org/https://doi.org/10.1016/j.jhydrol.2019.03.024, 2019.

Brunner, M. I., Sikorska, A. E., and Seibert, J.: Bivariate analysis of floods in climate impact assessments, Science of The Total Environment, 616-617, 1392–1403, https://doi.org/https://doi.org/10.1016/j.scitotenv.2017.10.176, 2018c.

Evin, G., Favre, A.-C., and Hingray, B.: Stochastic generation of multi-site daily precipitation focusing on extreme events, Hydrology and Earth System Sciences, 22, 655–672, https://doi.org/10.5194/hess-22-655-2018, 2018.

Evin, G., Favre, A.-C., and Hingray, B.: Stochastic generators of multi-site daily temperature: comparison of performances in various applications, Theoretical and Applied Climatology, 135, 811–824, https://doi.org/doi:10.1007/s00704-018-2404-x, 2019.

Müller-Thomy, H. and Sikorska-Senoner, A. E.: Does the complexity in temporal precipitation disaggregation matter for a lumped hydrological model?, Hydrological Sciences Journal, 64, 1453–1471, https://doi.org/10.1080/02626667.2019.1638926, 2019.

Sikorska, A. and Seibert, J.: Appropriate temporal resolution of precipitation data for discharge modelling in pre-alpine catchments, Hydrol. Sci. J., 61, 1–16, https://doi.org/10.1080/02626667.2017.1410279, 2018a.

Westerberg, I.K., Sikorska-Senoner, A.E., Viviroli, D., Vis, M., Seibert, J.: Hydrological model calibration with uncertain discharge data, Hydrological Sciences Journal, doi: 10.1080/02626667.2020.1735638, 2020.

Leave−one−out cross−validation

Multi−scenario test

Relative band spread (−)

Relative pools (−)

Gumbel variate (−)

Return period (years)

Method:
Ranking  Quantiling  Clustering

**Fig. 1.** New Figure 10.

[Figure]

**Fig. 2.** New Figure A2

**Annual max. daily precipitation depth**

**Annual max. daily temperature**

**Annual min. daily temperature**

**Fig. 3.** New Figure A3

[Figure]

**Fig. 4.** New Figure A4

[Figure]

**Fig. 5.** New Figure 5

**Ranking** **Quantiling** **Clustering**

Scenario 14

Scenario 93

Annual maxima (mm h$^{-1}$)

Gumbel variate (−)

Return period (years)

Evaluation:
Ensemble of 99 scenarios
PIs based on 99 scenarios
PIs based on repr. sets

**Fig. 6.** New Figure 9

**Fig. 7.** New Figure A1

---

## Author Response (AR1)

**Authors' reply to editor's and referees' comments**

**MS No. nhess-2020-79**

We thank the editor and two anonymous reviewers for a positive feedback and your detailed suggestions for improving our manuscript.

Following suggestions of both referees, we changed our manuscript at several places as described in detail in our responses to specific reviewer's comments. Most importantly, two new metrics were included for comparing results of three selection methods; additional figures were included in the Appendices that demonstrate the variability of the meteorological and hydrological scenarios, and demonstrate the performance of the hydrologic model in the calibration and validation periods. All figures were redrawn to improve their readability and to include missing information. We also made additional corrections in the manuscript followed by our internal review, and these are also included in the revised manuscript.

Below we provide our replies (*in black, italic*) to all individual comments of this reviewer (grey) and list the changes that were made in the revised version of the manuscript (in blue). The line numbers refer to the revised manuscript unless it is stated differently.

We hope that our responses and the revised manuscript will meet your expectations. Thank you for your time and efforts, and for considering our manuscript for a possible publication in the NHESS.

Yours Sincerely,

Anna Sikorska-Senoner, on behalf of all co-authors.

**Editor**

Editor Decision: Reconsider after major revisions (further review by editor and referees) (05 Sep 2020) by Paolo Tarolli

Comments to the Author:

Dear authors,

your article has been revised by two reviewers who proposed major and minor changes. You provided a detailed reply in the open discussion forum. I think you should have a chance to propose a revised version of the work. Therefore I decided to reconsider this paper after major revisions.

Please note that this editorial decision does not guarantee that your paper will be accepted for final publication in NHESS. A decision will be made when the revised version will be available and evaluated again with the help of the same or new reviewers.

Best regards

Paolo Tarolli

***Authors' Reply (AR):*** We thank for your feedback and we hope and our revised manuscript meets now your and referees' expectations.

**Referee #1**

**General comments:**

The paper is well written and properly structured. The scope of the study is clear, and the purpose of the selection of models among a given ensemble is relevant in the perspective of hydrological and hydraulic simulation for extreme flood estimation.

*AR:* Thanks for the positive overall assessment.

However, the presented methods, their scoring, and the interpretation of these scores deserve a better statistical assessment. For example, the presented scores, used in the comparison of the selection methods, rely only on the counts of simulated annual maxima being in or out of the predictive interval, without consideration of scale or frequency within their distribution.

*AR: Following this suggestion, we included two additional metrics that explicitly consider the resulting prediction intervals. These are the relative band spread of predictive intervals ($R_{\Delta PIs}$) and the overlapping pools of the predictive intervals ($R_{OPPIs}$). The latter metric is an adaption of the evaluation metric suggested by the reviewer.*

*See also our detailed response to the reviewer's comment to line 333.*

No assessment of the width of the predictive intervals is given relative to a global metric (standard deviation of annual maxima of the calibration data for example). Even if a relative ranking of the three methods can be provided here, it is difficult to have a proper statistical characterization of them in terms of robustness and reliability, which can be an issue for using them for extreme flood evaluation.

*AR: The newly proposed metrics should provide a sound basis for a more objective comparison of the three selection methods developed here (for details see our reply to reviewer's comment to line 333). In addition, we computed the latter metric i.e. the overlapping pools of the predictive intervals ($R_{OPPIs}$) in two ways. First, looking at all annual maximas (AMs), and then by focusing only on most extreme floods, i.e. only for the upper tail of AMs in the frequency space. This should shed light on the reliability and the robustness of the methods for rare floods.*

*See also further our detailed reply to reviewer's comment to line 333.*

Furthermore, no interpretation of these selections in term of flood process or modelling is given. It is not supposed to be the core of this article, but it would help to connect the results to some hydro-climatological features and their impact in terms of variability.

*AR: As suggested by the reviewer, in the revised manuscript we provided more details on the model calibration and validation as well as on the variability of the hydro-meterological scenarios.*

*See also our detailed replies to reviewer's comments to lines 374, 381 & 403.*

The problem of parameter equifinality is not evoked here, although it is the main factor of the parameters set variability, given that here, the 100 models have been calibrated by the same algorithm using the same data.

*AR: We thank the reviewer for this comment. Indeed, the equifinality issue is often a problem for any environmental model being difficult to overcome. Here, we used a heuristic approach in which 100 parameter sets are derived from 100 independent calibration trails using a Genetic Algorithm, i.e., each parameter set comes from an independent model calibration but using the same dataset and the same calibration algorithm. By using independent model runs, the possibility of being trapped in the same local optimum should be reduced (at least to some extent). Albeit being not directly related, this*

*heuristic method of sampling the parameter space is in line with the philosophy underlying ensemble forecasts (where model initial conditions are changed). It represents an interesting solution to systematic sampling of the posterior parameter distributions (e.g. via Markov Chain Monte Carlo Sampling) or to any Monte Carlo method relying on a very high number of model runs. This is more explicitly stated in the revised paper.*

*Despite the choice of 3 representative parameter sets for each case study will depend on the 100 sets available to be chosen from, the developed methodology will not depend on these available sets. Obviously different calibration methods could be here applied to derive these 100 parameter sets. We agree that it would be interesting to explore how such 100 representative optimal parameter sets should be chosen to cover different flow conditions, which would be however a completely different study than the one presented here.*

*We discuss this issue further in our detailed reply to reviewer's comments to lines 120 & 381.*

I would recommend then a major revision of this paper, in order to tackle with those main issues.

***AR:*** *we thank the reviewer for this positive feedback and valuable suggestions. Our detailed responses to the issues raised above are given below in responses to detailed reviewer's comments.*

**Detailed comments/questions:**

Quoted sentences are written in italic.

Line 10: *10'000 years of synthetic streamflow data simulated with a weather generator.* Simulated "thanks to" a weather would be more appropriate, the weather generator doesn't generates streamflow directly, it feds the hydrological models.

***AR:*** *corrected in the revised manuscript.*

Line 12: *The methods are readily transferable to other situations where ensemble simulations are needed.* This is only evoked as a perspective at the end of the paper, without providing an example of such "other" application. I am not sure it deserves to be in the abstract.

***AR:*** *To better demonstrate the possible applications of our method, we included an additional paragraph in the discussion (line 657-661):*

*"Possible applications of these selection methods include all studies, where computational requirements are an issue, e.g., rare flood analysis in safety-studies concerning dams or bridge breaks, climate scenarios of these, evaluation of rare floods as due to changes in climatic variables using several emission scenarios and different uncertainty sources propagation. Finally, these methods could be used for quantifying different uncertainty source contributions in rare flood estimates but with little efforts from the hydrologic model as due to parametric uncertainty propagation."*

Line 39: *initial conditions for use in combination with design storms.* Regarding the SCHADEX method detailed in (Paquet el al., 2013), I would add "initial conditions for use in combination with design or randomly drawn storms".

***AR:*** *added.*

Line 43: *especially if long time series are to be simulated using ensembles of hydrological parameter sets.* And also if very high return times (above 1000 years) have to be robustly estimated, thus implying several thousand years of simulation.

*AR: this information was added in the text.*

Line 45: *extrapolating a synthetic design hydrograph.* How? By scaling up a synthetic hydrograph thanks to estimated extreme quantiles of peak and volume values?

*AR: the missing information was added in lines 45-46 as:*

*"by scaling up an estimated synthetic design hydrograph by quantiles of extreme peak and volume estimated using frequency analysis."*

Line 51: *These continuous hydrologic simulation frameworks are still rare for time series >100 years.* 100 years of simulation merely allow to compute a robust 20 year return period estimation, which is pretty useless for dam safety for example. I would rather say that "high computational power are needed in order to provide estimations for high to extreme return times (up to 1000 years) required for safety-related studies" (although this is almost written in the same terms in line 57).

*AR: We shortened this sentence to (line 54):*

*"These continuous hydrologic simulation frameworks are still rare for time series >100 years due to heavy computational requirements (Grimaldi et al. 2013)."*

*And we also included additional information in lines 58-59 as:*

*"High computational power is particularly needed to provide estimations for high to extreme return periods (up to 1000 years and higher) required for safety-related studies or hydrologic hazard management. For such rare events,…"*

Lines 77-86: The problem is particularly well stated here.

*AR: Thanks for this comment.*

Line 89: *to select a reduced-size parameter ensemble for the use with a hydrological model within a continuous simulation.* Here and later on I would always keep "parameter" linked to "hydrological model". Most simulation frameworks are heavily parametrized, and the uncertainty linked to the hydrological model is only one (important) of the numerous sources of uncertainty. I would then write "to select a reduced size ensemble of hydrological model parameters for the use within a continuous simulation".

*AR: We corrected the sentence as suggested.*

Line 92: *for simulation of extreme floods*. A recurrent formal remark about the word "extreme". Usually "extreme floods" refers to return times largely exceeding the observational range, currently more than 1000 years, thus being extrapolated (by FFA or simulation, or both). This is especially true in dam-safety related literature. In the presented case, the meteorological scenarios are 100 years long, meaning that only very few "extreme" floods are simulated in the whole experiment. At best a robust 1000 years estimation can be empirically inferred here given the fact than 100x100=10 000 meteorological years have been simulated. The whole set of AM being extracted can surely not be considered as a set of "extreme floods". The authors could consider using "intense floods", "rare floods" or more simply "floods" when they refer to the simulated floods.

*AR: We changed the term 'extreme floods' into 'rare floods' throughout the manuscript and also in the title.*

Line 97: *simulated rare flood events.* Following the remarks above, the term "rare flood" is also appropriate alternative.

*AR: see our reply to the comment above.*

Line 99: *the aim is thus i) to provide long enough simulation periods for extreme flood analysis, ii) to avoid the propagation of errors due to data/model calibration etc. and iii) to be able to focus entirely on the uncertainty of the hydrological response.* I my opinion this goes farer than the actual results of the study. The uncertainty linked to model parameters is assessed, and properly summarized thanks to a reduced number of meteorological simulations. But I don't understand why this "provides long enough simulation", and why it "avoids the propagation of errors". If you have 100 "bad" models due to date, calibration, etc., three of them are selected in order to keep a good representation of their variability, but you still work with "bad" models (sorry for the term "bad", it only means "affected by uncertainty" !).

*AR: We use here synthetic data, instead of real observations, for deriving model simulations for testing the methods of selecting representative parameter sets of a hydrological model. This approach enabled us to generate long pseudo-observations for analysis of rare floods (100 meteorological scenarios x 100 years). Using such setting (instead of real observations that are usually of ~30 years at best at an hourly temporal scale) enabled us, first, to test the methodology on rare events of return periods >100 years (instead of return periods of up to 20 years if real observations were used). Second, taking such a setting as a start for our analysis, we focus here entirely on the parametric uncertainty of a hydrological model and do not infiltrate into other uncertainty sources, i.e. input or output uncertainties of calibration data. How these parameter sets are derived, and the way of model calibration may contribute to the model ability to simulate real events. Calibration of the hydrological model is, however, not the focus of this paper. Hence, our ensemble of 100 parameters sets is assumed to be the best representation of model parameters one can get. We agree, however, with the reviewer that the uncertainty related to model calibration and calibration data are partly included in the calibrated model parameter sets and will be included in the simulations we get with these sets.*

*To clarify these issues, we modified our text in lines 104-109 into:*

*"Using synthetic instead of observed data is here important as only recently Brunner et al. (2018b) have shown that the record length is one of the most important sources of uncertainty in design floods. Hence, using a simulation setting with synthetic data as a start for our analysis enables us i) to provide long enough simulation periods for rare flood analysis with return periods ≥100 years, and ii) to be able to focus entirely on the uncertainty of the hydrological response, while other uncertainty sources of a hydrologic model (due to model calibration) are not explicitly considered. Note that way the hydrologic model is calibrated lies outside of this paper scope."*

Line 111: *and not the model uncertainty of a weather generator.* This is perhaps one of the main limit of the study. At line 69, it is written that Arnaud et al. (2017) *found that the uncertainty of the rainfall generator dominates the uncertainty in the simulated extreme flood quantiles.* This uncertainty will not be considered here, and I wonder how far the results exposed here would still be useful to deal with the weather generator uncertainty (which of course is not to be confused with the variability of the scenarios generated thanks a given set of parameters). A comparison of both uncertainties (model and weather generator), even basic, would have been welcomed here.

*AR: As mentioned in the response to the above comment and in the manuscript itself, the effect of weather generator uncertainty is not here considered. In this work, we use a tested and verified set-up of the weather generator (Ervin et al. 2018, 2019) and ready computed meteorological scenarios for our test study. As already written in the original manuscript (lines 98-111, lines 117-120 in the revised manuscript):*

*"A meteorological scenario represents a single realisation from a stochastic weather generator with constant model parameters. These meteorological scenarios are equally likely model realisations that differ in the precipitation and temperature patterns, and together they represent the natural variability of the climate (and not the model uncertainty of a weather generator)".*

*Therefore, the assessment of the uncertainty due to the weather generator is not possible here. To assess the effect of the uncertainty of the weather generator on the simulated time series, uncertainty in parameters of the weather generator should be considered that would lead to very different meteorological scenarios from these that are here applied. We agree that such a study would be very interesting, but it exceeds the scope of this paper that is the selection of representative sets of hydrologic model parameters.*

Line 120: *(ii) the distribution is known.* I am not sure that knowing the probability of parameters is a reasonable perspective, in my opinion the problem of equifinality of parameters in models like HBV prevents an a priori expression of parameter probability, as different sets of parameters can "produce" the same model, i.e. models having the same behaviour for a given meteorological scenario. And this is one of the interesting outcome of this study, which focuses on the hydrological response of the models, and not on the actual values of the parameters. I think that this equifinality problem deserves more writing in this paper.

**AR:** *Thank you for this comment. We refer here to the situation when the parameter distribution is estimated from the observation data, e.g. through Monte Carlo or Bayesian approaches. Hence, we clarified that in the text of the revised manuscript in lines 131-132 as:*

*"the distribution is known (i.e. estimated from data)…"*

*We agree however that the issue of the parameter equifinality deserves more attention in our paper and thus we included an additional paragraph on that in the revised manuscript in the introduction (lines 63-64):*

*"Using multiple parameter sets for a hydrological model is justified by the parameter equifinality..."*

*And in the discussion (lines 629-636):*

*"Downsizing the hydrological model parameter sample can only aim at understanding and characterizing the hydrological part of the full hydrological ensemble resulting from a combination of multiple parameter sets and multiple meteorological scenarios. The variability of hydrologic model parameters arises from the parameter equifinality (Beven and Freer 2001), and it can be overcome by using several hydrologic model parameter sets that should encompass the parametric and (implicitly) also other uncertainty sources. Our selection methods thus enable one to choose representative parameter sets from the hydrologic responses point of view and in this way to cover the variability of hydrologic responses with reduced hydrological model runs needed. These methods are however not applicable for characterizing the climate variability (nor for downsizing the number of meteorological scenarios needed)."*

Line 129: *The infimum (from the Latin – smallest) and supremum (from the Latin – largest) refer to the greatest lower bound and the least upper bound (Hazewinkel, 1994), i.e., the largest interval bounding the ensemble from below and the smallest interval bounding it from above.* This definition deserves to be connected to frequencies of the target variable, even if it's not straightforward. Does it (roughly) provides a 90, 95 or 99% confidence interval of the simulated variable? Given that follows in lines 184 to 234, with quantiles 5 and 95%, it "looks like" a 90% CI.

*AR: yes, the bounds bounded by the infimum and supremum set represents the 90% predictive intervals. This information was added in the text (lines 141-142).*

*"Thus, the representative band should correspond to 90% predictive bands of a target variable".*

Line 138: *we thus propose to use the representation of AMs in the Gumbel space as the reference model response space for parameter selection.* The plotting in Gumbel is useful here to illustrate the rare to extreme quantiles. Still, it doesn't explicitly play a role in the "parameter selection" (which doesn't imply any explicit reference to an implicit Gumbel distribution of AM in the statistical criteria/indicators used).

*AR: this is a good remark. Indeed, the selection is based on the absolute values of the annual maxima without referring to their distribution but with accounting for their location in the list of maxima sorted by their magnitudes. The Gumbel space is used for plotting purposes and to visualize the selection method and results. We corrected this issue throughout the manuscript and in the sentence referred above as (line 150):*

*"we thus propose to use the representation of AMs sorted by their magnitudes (i.e. frequency space) as the reference model response space for parameter selection."*

Line 142: *inverse modelling approach.* The term "inverse modelling" appears to me somehow excessive. An inverse hydrologic modelling would be for example to infer rainfall from discharges. Here it's more a "post-modelling" approach.

*AR: we agree and corrected that in the revised manuscript.*

Line 145: *the parameter set selection is made based on the full hydrological simulation ensemble but using only a limited simulation period.* To be more specific I suggest to write "based on the simulation with all the hydrological models but using [: : :]".

*AR: we corrected this sentence to (lines 157-158):*

*"The main idea behind all three methods is that the hydrologic parameter set selection is made based on the full ensemble with all hydrological model simulations…"*

Line 189: The parameter sets selected in step (d). Should be step (c).

*AR: this was corrected.*

Line 189: *the sets which are chosen most often as the 5th, 50th and 95th ranks are retained as the parameter sets [: : :] representative for the entire simulation period.* The ranking methods yet shows its weakness: the 5th and 95th of a given year have very low chance to match to the overall corresponding quantiles, given the "climate variability" illustrated in Figure 1, thus preventing the parameters selected on a given year to have a global representativeness. I am not sure it's worth keeping this method "in the game" for the rest of the paper: : :

*AR: Thanks for this comment. We agree that the ranking method has its weaknesses, as it also appears from our analysis. However, its great advantage is that it is straightforward to be performed. Thus, we prefer to keep this method for detailed analysis in the paper and reject it based on the results and not prior to that.*

Lines 194-206: I don't understand why the "Gumbel space" is evoked here (three times!), and constantly throughout the paper. Apart from the plots of Figure 2 and others, what is "Gumbel specific" in the metrics and statistics presented? For example, the RMSE scores are computed using each simulated annual maximum, regardless its empirical frequency.

*AR: yes, we agree with the reviewer and removed referring to the Gumbel space in selecting the representative parameter sets. We kept referring to the Gumbel space when it comes to visually presenting and comparing results.*

Line 201-205, equations 1-3: This is the same equation for the three considered quantiles. Only one is necessary.

*AR: This is another very good point! These three equations were combined into one equation.*

Line 211-234: same remark as above about the "Gumbel space".

*AR: We removed the reference to Gumbel space where it is not valid.*

Lines 214: *These members are next clustered in the Gumbel space into three representative groups (clusters) based on all J simulation years using the k-means clustering.* If I understood properly it means that the clustering has been performed in the J-dimensional space of the full set of members values?

*AR: yes, this is correct. Clustering was computed on J-annual maxima.*

Line 217: *Next, these clusters are sorted by their magnitude.* What variable/quantile is used for this sorting?

*AR: Sorting of clusters is done based on their means and their magnitudes for the quantile 90%. We have tested several quantiles in the upper range (i.e. 80%-99%), and the choice of quantile did not impact the ordering of clusters. To clarify that, we added the following additional information in the revised manuscript in lines 225-230:*

*"Next, these clusters are sorted based on cluster means by their magnitude by comparing percentiles in the upper tail of the distribution (here we used the 90th percentile). Use of a percentile from the upper tail is important as methods are focusing on rare floods. However, we found that the method was insensitive to the percentile choice as long as it lies in the upper tail (i.e. ≥80th percentile). Based on the percentiles computed for each cluster mean, the lower, middle and upper clusters are defined. Next, for the lower cluster a 5th percentile, for the upper -- 95th percentile, and for the middle -- 50th percentile are computed, i.e., $P_5$, $P_{50}$ and $P_{95}$..."*

Line 218: *Note that we use here percentiles instead of cluster means to make this method comparable with the other two methods*. I am not sure of that : say that each cluster regroups one third of the ensembles, and for a given quantile in the AM distribution (say the 50%), it is evenly distributed through all the members, the percentile 5% of the lower cluster would more or less correspond to a 0.05x0.33 = 0.17 global percentile. The 5% and 95% are more "rare" than their corresponding quantile in the quantiling method: : :

*AR: we understand the reviewer's point. Yet, the clusters are rarely symmetric, i.e., there is no assumption that the clusters have to be symmetric. Hence, it is hard to define a more fair selection of intervals than the one we describe here, i.e. based on percentiles. Choosing 5% and 95% makes the clustering theoretically comparable to the other two methods. Having symmetric clusters would imply an idealistic situation when the responses of a hydrological model provide simulations of the annual peaks that are symmetrically spread around the median values, which is hardly realistic for real cases. This, however, does not have to be the case, as the choice of different parameter sets may lead to emphasize different hydrological processes in the catchment, i.e. floods resulting from snowmelts or intensive rainfalls.*

*We added an additional sentence on that in the text of the revised manuscript in lines 232-234:*

*"Use of the 5th and 95th percentiles appears as a fair choice for asymmetrically spread clusters, which is most often the case as different parameter sets of a hydrologic model may emphasize different hydrologic processes in the catchment."*

Line 223-227, equations 4-6: Same remark as for equations 1 to 3.

***AR:*** *We combined equations 4-6 to a single equation in the revised manuscript.*

Line 233, equation 7: This mention of the plotting position mention could be moved at line 195.

***AR:*** *as we removed the reference to the Gumbel space in line 195, we keep this equation at its original place. However, we added additional text in line 242 to link the method to plotting results in the Gumbel space:*

*"For visualizing the selection methods, we use the Gumbel space (Generalized Extreme Value distribution Type-I) with the Gringorten's method (Gringorten 1963) to compute the plotting positions of AMs in the Gumbel plots:…"*

Table 1: *Sorting space = Gumbel space.* Once again, I don't undertand how "Gumbel specific" the sorting process is for Quantiling and Clustering.

***AR:*** *we corrected 'Gumbel space' into annual maxima frequency when referring to the sorting space in the table.*

Table 1: *Interpretation of pred. intervals / Parameter grouping.* I don't see to what these lines refer in the text before.

***AR:*** *We included additional information in the text (lines 260-261) that refers to the interpretation of predictive intervals contained in the table:*

*"The derived predictive intervals thus are sensitive to individual years of simulations, and their interpretation may be difficult (as they do not result from any flow frequency analysis)."*

*And regarding the other two methods (lines 264-265):*

*"This enables statistical statements to be made about the selected parameter sets and about the predictive intervals constructed with the help of these parameter sets (as they are constructed on the entire simulation ensemble)."*

*And regarding the parameter grouping (lines 266-268):*

*"Finally, the clustering method splits all ensemble members (hydrologic simulations) into three clusters, and so each parameter set can be attributed into corresponding clusters. This could be useful if one would like to extract more information on each cluster behaviour."*

Line 254: *assessing how well the reduced ensembles cover the reference simulation ensemble.* More specifically I would rather say "how well the reduced ensembles substitute the whole simulation ensemble for the selection of representative parameter sets".

***AR:*** *This was corrected accordingly.*

Line 273: *assess how well the defined identified intervals represent the ensemble members of this Sr meteorological scenario.* What metric is used to do this assessment?

***AR:*** *All assessment criteria are described in sect. 2.6. For clarity, we included in this sentence link to this section.*

Line 288: *Compute the 5th percentile [: : :] and the 95th for {H(Θsup,p/Sm)}* . The mention "for m=1,2: : :,M and m≠p" could be added for more clarity.

***AR:*** *was added.*

Line 290: same question as for line 273.

***AR:*** *see the reply to the question for line 273.*

Line 295 and below: as evoked above, the recurrent mention to the Gumbel space is, in my opinion, useless, and over time tedious to read. In the paper, it is quickly implicit that the plots and the metrics used are defined in the frequency space (or frequency domain), being Gumbel or not, without need to repeat it.

***AR:*** *we agree with the reviewer and modified this text which should read now (lines 317-318):*

*"The simplest way of assessing the behavior of these three methods is a visual inspection of curves plotted in the* frequency space (e.g. using Gumbel distribution for plotting)…"

Line 330: *Here we propose to use different percentiles, i.e., the 5th, 50th, and 95[th] percentiles, to characterize the ratio of the simulation points lying outside the computed predictive intervals for each of the methods.* I don't understand this? Why not using only the 50th percentiles of this ratio? Refer to comments on Table 3 for a more detailed version of this question.

***AR:*** *yes, this is a good suggestion. We removed the 5[th] and 95[th] percentiles and kept only the 50[th] percentile for comparing methods. This should correspond to the situation with the majority of simulation points or scenarios lying inside/outside the bands.*

Line 333: *how many out of J hydrological simulation points [: : :] must lie outside the defined predictive intervals.* I think that the problem of such a simple "count" of points (simulated annual maximum) outside the predictive interval doesn't take into account their position in the simulated distribution. As written in the title, the methods exposed here are supposed to be used in the estimation of extreme floods, which in any post-treatment of the hydrological simulation will strongly rely on the high simulated quantiles. The scores should somehow reflect this focus on high quantiles, which is not the case here. Instead of this count of "outside points", the area outside the predictive interval could be computed, using the Gumbel variable (as x) and the discharge value (as y), thus giving a contrasted score in which lying outside the predictive interval for high quantiles is more important than for low values.

***AR:*** *We thank the reviewer for this very useful comment! In the revised manuscript, we included two additional metrics that should put more focus on extreme floods. The newly proposed metrics are the relative band spread of predictive intervals ($R_{\Delta PIs}$) and the overlapping pools of the predictive intervals ($R_{OPPIs}$). The latter metric is an adaption of the suggested by the reviewer computation of pools lying outside PIs. These new metrics are described in text in sect. 2.6.2, i.e. Quantitative assessment (lines 348-360 & 371-373):*

*"[(IV)] Relative band spread of PIs ($R_{\Delta PIs}$) that is computed for both tests and which compares the spread of PIs constructed with the representative parameter sets versus 90%-PIs of the full hydrologic ensemble. In details, $R_{\Delta PIs}$ is computed for each m-th scenario as:*

$$R_{\Delta PIs,m} = \sum_j^J \frac{S_{PIs,repr,m}}{S_{PIs,full,m}} \quad m = 1,2,\dots,M \ \& \ j = 1,2,\dots,J \qquad\qquad Eq.\ 13$$

*where $S_{PIs,repr.,m}$ and $S_{PIs,full,m}$ are band spreads of the 90%-PIs constructed with the representative parameter sets and with the full hydrologic ensemble. The band spread is computed as a difference*

*between the upper (or supremum) and the lower (or infimum) interval at each j-th simulation point in the frequency space.*

*[(V)] Overlapping pools of PIs $R_{OPPIs}$ that are computed for both tests in the frequency space by taking the Gumbel variate and discharge values of sorted AMs as coordinates of the PI pools. In details, $R_{OPPIs}$ of PIs constructed with the representative parameter sets is computed for each m-th scenario as:*

$$R_{OPPIs,m} = \sum_{j}^{J} \frac{(k_j - k_{j-1})}{2} \left( H(\theta_{sup,m,j}) + H(\theta_{sup,m,j-1}) - H(\theta_{inf,m,j}) - H(\theta_{inf,m,j-1}) \right)$$

$m = 1,2,\dots,M \ \& \ j = 2,3,\dots,J$ *Eq. 14*

*Similarly, $R_{OPPIs}$ is computed for the full hydrologic ensemble using the pool restricted by the 90%-PIs, i.e. taking the 5% and 95% intervals as pool borders.*

*(…)*

*With regards to $R_{\Delta PIs}$, we propose to compute the relative band spread as a mean overall sorted AMs at first. Also, to focus more on rare floods, we propose to compute the means of rare floods limited by different Gumbel variates. Here we computed $R_{\Delta PIs}$ for the upper half of AMs ($R_{\Delta PIs,j\geq51}$), for the most upper ten AMs ($R_{\Delta PIs,j\geq91}$) and the most upper five AMs ($R_{\Delta PIs,j\geq96}$)".*

*These metrics are computed for both the cross-validation and multi-scenario evaluation tests and the medians over all scenarios will be summarized in the table 3 and in lines 524-532:*

*"Analysis of overlapping PIs pools ($R_{OPPIs}$) and relative band spreads ($R_{\Delta PIs}$) show that in the cross-validation test, all methods provide bands that are wider than the 90%-PIs computed using the full simulation ensemble. This should not surprise as the selection of relative parameter sets is based on a large sample of hydrologic model realisations (i.e. 99 scenarios) than the full ensemble for model assessment (i.e. single scenario). However, these metrics show large differences in the multi-scenario test, in which the clustering method outperforms other two selection methods, particularly when the focus lies on rare floods (compare $R_{\Delta PIs,j51}$, $R_{\Delta PIs,j91}$ and $R_{\Delta PIs,j96}$ in Table 3). The quantiling was the second good method, while the ranking was performing the worst. These observations are also confirmed when looking at the variability of these two metrics for different return periods (Fig.10). A better performance of the clustering method can be again noticed in the range of rare floods. While quantiling was performing worse than clustering, it was still better than the ranking method."*

*In addition, we introduced an additional figure which presents cumulative results of these two metrics for all scenarios (new Figure 10).*

[Figure]

*Figure 10. Evaluation of the leave-one-out cross-validation and the multi-scenario test for the three selection methods using the relative band spread ($R_{\Delta PIs}$) and the relative overlapping pools ($R_{OPPIs}$), both computed with reference to the 90%-PIs of the full hydrologic simulation ensemble.*

Line 337: *In this work, we consider the following values for rthr = {0.50,0.25,0.10,0.05}.* Following the preceding comment, a metric accounting for the scale or the frequency of the points being outside the predictive interval would avoid to distinguish such thresholds, which apart from the rtht=0.50 or 0.10 have little statistical meaning in this context.

**AR:** *We appreciate this comment and reduced the thresholds to only two values, i.e. 0.50 and 0.10.*

Line 343: *For testing the methods developed here, a small natural catchment is preferable.* Why small?

**AR:** *The choice of a small catchment is driven by the fact that hydrological responses in a small catchment may be better understood and are rather unaffected by anthropogenic factors (such as dams, bridges, etc.) which may perturb the flood peaks in terms of very rare floods. We included additional information in the text in the revised manuscript (lines 378-380):*

*"For testing the methods developed here, a small close to natural catchment is preferable, i.e. with only little anthropogenic influences, in which hydrological responses are transparent, and the generation of rare flood (peaks) is not affected by human constructions (dams, bridges)."*

Line 358: *In this study, the version HBV light [: : :] with 15 calibrated parameters is used.* The considered model can be then considered as heavily parametrized, and thus fully affected by the equifinilaty problem of its parameter evoked in the remark made for line 120.

**AR:** *We agree that the model can be affected to some extent by the equifinality problem. However, the HBV is a conceptual model that covers different modes of the runoff generation accounting for snowmelt, soil moisture and groundwater. To correctly represent these processes and to provide hydrologically interpretable parameters, the parameters of individual modes should be included in the calibration process. Hence, we used here such a set-up with 15 calibrated parameter sets, which was previously successfully applied in Swiss catchments. We added a following sentence to the text in lines 394-396:*

*"Such a set-up of the HBV light was previously successfully applied in Swiss catchments (e.g. Sikorska & Seibert2018a, Brunner et al. 2018c, Brunner & Sikorska-Senoner 2019, Müller-Thomy & Sikorska-Senoner 2019, Westerberg et al. 2020)."*

Line 374: *for details on RPEAK and RMARE, see the work of Vis et al. (2015).* A brief description of RPEAK and RMARE would be welcomed here, especially as within the calibration process, the parameters conditioning the modelling of floods are surely strongly conditioned by the RPEAK score.

**AR:** *with response to this comment, we added additional information in the major text and equations used for computing $R_{KGE}$, $R_{PEAK}$, $R_{MARE}$ in the newly created Appendix D.*

*The new text (lines 410-412) reads as:*

*"$R_{PEAK}$ is defined in a similar way to the Nash-Sutcliffe efficiency but using peak flows instead of the entire time series. While both $R_{KGE}$ and $R_{PEAK}$ focus on high (peak) flows, $R_{MARE}$ is sensitive to low flows. See Appendix D for equations."*

Line 377: *The available observational datasets are split into a calibration (1990-2005 years) and a validation (2006-2014 years) period.* What is the point of having a validation period here? This validation period is never used in that follows.

**AR:** *Splitting into calibration and validation period is needed to assess how well the calibrated model performs outside the calibration period. This assessment is important, as the paper focus lies here on simulations that are not included in the calibration period. We added a following sentence into the text (line 417-418):*

*"Evaluation of the model in the independent period is important as the model is applied to simulate time series outside the calibration period."*

Line 381: *The calibration is repeated 100 times resulting in 100 independent optimal parameter sets.* I am surprised by the variability of the parameters obtained by these 100 calibration runs, performed on the same calibration data with the same objective function. I would like to read a comment from the authors on that. Mine is that the optimization is not complete, seeming to depend on the aleatory exploration performed by the genetic algorithm, somehow "trapped" in local optimums, and/or affected by a strong equifinality problem (yes, once again, sorry). An alternative strategy for the generation of model parameter sets, in my opinion providing more "independent" models, could be to bootstrap 12 years among the 24 years available in order to generate 100 truly different calibration & validation samples.

**AR:** *We agree that a global optimizer should always lead to the same optimized parameter set except for specific cases where the response surface (the objective function) clearly has several optima with exactly the same value (which can happen in theoretical settings, e.g. with the so-called Himmelblau function). In settings where the response surface has many local optima with very similar values (equifinality), the start point of the search can indeed lead to a trapping in local optima, especially if the optimizer is stopped after a fixed number of iterations as is the case here. A solution to overcome this problem is classical Markov Chain Monte Carlo (MCMC). As we aimed here at developing the parameter selection method that could be applied also for selecting parameter sets from independent model calibrations, MCMC was not a desirable solution here. As a heuristic solution, we proposed here multiple independent trials using a genetic algorithm to derive 100 sets of good model parameter sets. This is an approach which we have in numerous studies recently (e.g., Seibert and Vis, 2016, van Meerveld et al., 2018; Etter et al., 2020).*

*Note however that the method used for calibration of the model and the selection of 100 'best' parameter sets was not the objective of this paper. Obviously, there are other methods to obtain 100 parameter sets (or more) are possible such as based on Monte Carlo methods or by bootstrapping as suggested by the reviewer, but the approach of independent calibration trials has been found suitable in these previous studies. Additional text was included in the revised manuscript and the calibration description was rewritten as (lines 406-410):*

*"To derive multiple parameter sets of the HBV model, we propose a heuristic approach that relies on multiple independent model calibration trials using a Genetic Algorithm approach (Appendix A). By using independent model runs, the possibility of being trapped in the same local optimum should be reduced. The Genetic Algorithm is used together with a multi-objective function $F_{obj}$ with three scores: the Kling-Gupta efficiency ($R_{KGE}$) and the efficiency for peak flows ($R_{PEAK}$), and a measure based on the Mean Absolute Relative Error ($R_{MARE}$)."*

*And in lines 421-423:*

*"Here, the Genetic Algorithm is run 100 times resulting in 100 independent optimal parameter sets (see Fig.A1 in Appendix C). These 100 parameter sets represent similarly likely parameterisations of the hydrological functioning of this catchment and their variation can be explained by the equifinality of hydrologic model parameters (Beven & Freer 2001)."*

*And in lines 428-429:*

*"Note that the described above way to derive 100 parameter sets is one possible approach, and other calibration methods could be used (e.g. Monte Carlo or bootstrapping)."*

*And in the discussion in lines 596-600:*

*"In this work, to derive 100 parameter sets, we proposed a heuristic approach that relies on multiple independent model calibration trials using a Genetic Algorithm approach and a multi-objective function. This method represents an interesting solution to systematic sampling of the posterior parameter distributions (e.g. via Markov Chain Monte Carlo Sampling) or to any Monte Carlo method relying on a very high number of model runs. Its strength is that it can be applied for selecting parameter sets from independent model calibration settings (with different scores, calibration periods, etc.)."*

Line 381: *The median model efficiency measured with Fobj over all 100 runs is 0.7.* To better assess the quality and the variability of the models generated at this step, it would be useful to show the distribution of NSE (Nash & Sutcliffe Efficiency) for both calibration & validation, and the ensemble plots of daily regime and classified discharge distribution for all the generated models. The ensemble simulation of the biggest observed would also be very pedagogic.

**AR:** *The Nash-Sutcliffe values for the calibration and validation periods are now provided in the revised manuscript in newly included Appendix: Model calibration results. In addition, flow duration curves are provided for mean daily discharges for simulated versus observed values and these are also included in the same appendix.*

[Figure]

*Figure A2. Flow duration curves and model performance metrics for calibration and validation periods over all 100 optimized parameter sets.*

*Also, the following sentence was added in the main text in lines 425-427:*

*"Also, diagnostics of the Nash-Sutcliffe efficiency and the Peak efficiency demonstrate that the model performs well in the range of high flows which are most important for simulation of rare floods studied in this paper (see Fig. A2 in Appendix C)."*

Line 383: *which can be assumed to be a good model performance on an hourly scale.* As mentioned above, this really need to be illustrated more richly.

**AR:** *please see our reply to the above comment.*

Line 397: *The daily values generated with GWEX_Disag were then disaggregated to hourly values using the meteorological analogues method.* More details would be welcomed on that disaggregation: what fields/variable are used for analogy, what analogy criterion, what about seasonality (i.e. are the analogues identified within period of the year similar to the one of the simulation to be disaggregated, etc.).

**AR:** *The disaggregation with the weather generator is not the scope of this paper, and it was already published by Evin et al. (2018, 2019). Thus, we refer to the source papers on this disaggregation scheme. Here we use the ready meteorological scenarios that were made available for this catchment. These scenarios were developed for the entire Aare catchment, and here we only use a small subset for a single catchment.*

Line 399: *Next, catchment means were computed using the Thiessen polygon method.* On how many simulated precipitation stations do the Thiessen average rely on for the considered catchment? How many simulated stations lie within the catchment?

**AR:** *3 stations located close by were used for that purpose. This information was added in the text.*

Line 403: *Thus, differences between scenarios are exclusively due to the natural variability of the meteorological time series. "[: : :] and modelled by the GWEX weather generator" could be added.* Similarly to the models, the variability of these scenarios deserve to be illustrated, and compared to the observations, e.g. thanks to their average and standard deviation of the annual maximum daily precipitations.

**AR:** *we added the sentence as suggested.*

*In addition, the variability of meteorological and resulting hydrologic scenarios is presented in a newly introduced figure in the new Appendix: Scenarios variability and briefly described in the appendix.*

[Figure]

*Figure A3. Variability of 100 meteorological scenarios used in this study vs. observations.*

[Figure]

*Figure A4. Variability of 100 hydrologic scenarios used in this study; left panel – hydrologic ensemble with all meteorological scenarios and all hydrologic model parameters; right panel – hydrologic ensemble with all hydrologic model parameters but for the median meteorological scenario only. PIs represent the 90% predictive intervals.*

Line 406: *These 100 meteorological scenarios are used as input into the HBV model to generate streamflow time series with 100 different HBV parameter sets.* I am not sure that this sentence is useful. The simulation scheme is clear from the beginning.

**AR:** *we prefer to keep this sentence for clarity and the reading flow.*

Figure 4: The title of the second plot should be "Quantiling" instead of Quantailing.

**AR:** *we removed this plot from the manuscript.*

Lines 419-434: I find the results of this paragraph difficult to interpret. Some violin plots show odd parameter selection patterns (like in Clustering/Infimum), other show weak parameter discrimination (Quantiling/Median). The Table 2 is quite difficult to read/interpret with so many counts exposed. In this paragraph and in the following ones, some "illustrations" of the most selected parameter sets should be provided, e.g. by presenting the range of hydrological responses to observed meteorological data of the selected models compared to the full ensemble. In other words, some interpretation in term of modelling and hydrological processes would be welcomed.

**AR:** *we remove the violin plots, and instead, we included an alternative figure that presents a grouping of representative parameters selected with three methods. This enables us to look at how the selection of parameters corresponds to different processes being modelled by the model.*

[Figure]

*Figure 5. Box-plots showing the variability of the hydrologic parameter sets selected as the representative parameter sets over 100 meteorological scenarios chosen with three methods. The white box-plots illustrate the entire parameter ensemble (i.e. 100 sets), outliers are not presented. I - infimum, M - median and S - supremum set. Units as in Table A1. The blue box surrounds parameters from the response routine, the grey box from the snow routine and the yellow from the soil moisture*

*routine. MAXBAS is the only parameter from the routing routine, and CET is a potential evaporation correction factor.*

*Following the new figure, a new paragraph was included in lines 475-483:*

*"The variability of selected parameter sets is presented in Fig.5. As can be seen from the figure, some parameters presented smaller and others larger variability of selected sets. It also appears that different values are selected for the infimum, median and supremum set but not always. Among three selection methods, the ranking method (marked in green) has the largest spread of parameter values for most of the parameters. The clustering (blue) and quantiling (yellow) selection methods seem to choose more extreme parameter values for both, i.e. infimum and supremum sets. Looking at different model routines and hydrological processes behind, no clear patterns could be seen regarding the choice of parameter sets. It appears however that the representative parameters from the response (blue) and soil moisture (yellow) routines have a smaller spread than those from the snow routine (grey), as they are more often outside and further away from the interquartile ranges (grey boxplots)."*

*We also reduced the table to show only the three most selected parameters (instead of 5).*

Line 434: *Interestingly, for the supremum set in the clustering method, only four parameter sets among all 100 available are chosen over all 100 scenarios.* Given that, I don't understand why in Table 2, column Clustering, 5 parameters sets (# 34, 22, 98, 86, 50) are identified.

**AR:** *thank you for spotting this typo! It should be written for infimum set and not for supremum set.*

Table 2: a graphical alternative or a complement to that table deserves to be presented, to better assess the "density" of parameter sets selected by the different methods.

**AR:** *see our response to the comment to lines 419-434.*

Line 437: *intervals for extreme flood predictions.* The term "extreme flood estimations" could be more appropriate.

**AR:** *this was corrected.*

Line 441: *According to a first visual assessment, these three methods lead to slightly different constructed frequency intervals particularly in the upper tail of the distribution.* To ease this visual assessment, horizontal lines marking these intervals for the upper values could be added to the plots of Figure 5.

**AR:** *Horizontal and vertical lines were added to all frequency plots.*

Line 446: *the three intervals are always correctly attributed.* I would temper this in writing that "the three intervals are always correctly ordered" as this exactly what it is measured in Rbias.

**AR:** *yes, it was corrected.*

Line 456: From the visual assessment, it is difficult to judge the methods. See remark on Figures 7-8. Figure 5 to 8: Instead of having an x-axis graduated with the Gumbel variable U, some ticks at remarkable return times (2, 5, 10, 20, 50 & 100 years) could be added in order to ease the reading of these plots, and avoid the long caption *The Gumbel variates etc.* in Figure 4.

**AR:** *Information on return periods was included in all figures.*

Line 463: *the highest values for both evaluation criteria, i.e., the median ratio of simulation points lying outside the predictive intervals (Rspo) and the median ratio of hydrological simulation ensemble [: : :].* Given the definitions of §2.6.2, this is more a mean ratio than a median ratio.

**AR:** *It is a mean for each scenario but median over all scenarios. To clarify that, we included the additional text in line 512:*

*"… ,both presented as median values over all scenarios."*

Line 475: *Hence, again here all three method can be qualified as behaving well based on the multi-scenario evaluation, and only the order of their behavior can be established.* Honestly, at the end of this paragraph, I have no clear idea of the absolute performance of each method. One important point is that the methods provide rather different intervals (like illustrated in the Figure 6), thus a method providing wide intervals will have good "in/out" scores (like the ones in Table 3), better than for narrow intervals, but the question of the statistical relevance of such intervals is not solved.

**AR:** *We added an additional paragraph at the end of the result section that summarizes major findings (lines 533-539):*

*"As it appears from the above, the rejection or acceptance of one of three methods tested here is not straightforward. Apart from the ranking method, which was linked with a huge bias, both other methods, i.e. quantiling and clustering were performing similarly well. Yet, these methods provide quite different intervals (of a different spread). The validity and usefulness of these methods for selecting the representative parameter sets are thus further discussed below in sect. 5.1. The detailed analysis of the relative band spread and the overlapping pools indicated however that the clustering method was performing the best particularly in the range of rare floods. The quantiling method was scored as the second best, while the ranking method was performing poorest."*

Figure 7-8: I found the plots of the top panels of both figures rather counterintuitive: *the prediction interval resulting from selecting representative parameter sets for 99 meteorological scenarios and compared to the full simulated range with all 100 parameter sets* seems narrower than "statistically expected" (more and less a 5-95% confidence interval given the quantiles or percentiles involved in the process). For the highest simulated quantiles, the prediction interval seems only to cover about 50 to 66% of their variability. In the bottom plot of the Figure 7 for clustering, a second blue interval is plotted without being identified in the legend nor in the caption.

**AR:** *This second blue interval is indeed the grey interval, which comes from the scenario assessed here. For clarity, we redesigned these plots by removing the upper panel, which might have been misleading, and by improving the readability of the bands.*

[Figure]

*Figure 9. Example of multi-scenario evaluation for the three selection methods and two meteorological scenarios. PIs represent the 90% predictive intervals*

Line 480: [: : :] *selecting representative parameter sets that yield reliable predictive intervals in the frequency domain.* Following the comment on line 475, I see no statistical demonstration of the reliability of the predictive interval (like the one that could be done by controlled random generation of a given variable to which a statistical test is applied, then a proper statistical scoring). I agree with the authors on that a ranking between the three methods (according to the presented scores) is however established.

***AR:*** *Two additional metrics giving more focus to extreme floods were included in the revised manuscript. See also our above reply to the reviewer comment to line 333.*

Table 3: Three quantiles of the Rspo score are given, although the caption mentions that *the values represent the median values over all 100 scenario runs.* What for providing the 5th and 95th quantile of a score measuring *the ratio of simulation points [: : :] lying outside the predictive intervals* (line 312), which should be, on average, close to 10% (once again given the quantiles involved in the selection process)? In the low part of the Table 3, the Metric method is written as Rhso (Rmso). Which scores are the ones provided?

***AR:*** *The $R_{spo}$ is now presented only for $50^{th}$ percentile. Regarding $R_{hso}$ & $R_{mso}$, both metrics are provided in table 3 as $R_{hso}$ is used in the cross-validation test and $R_{mso}$ in the multi-scenario test. This was now clarified in the new table 3.*

Line 481: *all three methods are fit-for-purpose for extreme flood simulation.* Following the preceding comment, if the presented method cannot be statistically demonstrated, it can be considered as an ad-hoc heuristic, build for a given purpose, here extreme flood simulation/estimation. This last step is not evoked in the paper, then depriving the reader from assessing the relevancy/robustness of this heuristic.

***AR:*** *two new metrics that were introduced (relative band spread of PIs and their overlapping pools) should provide better evaluation and comparison of the proposed selection methods. See also our reply to the comment to line 333.*

Line 487: *for additional ease of use criteria*. I don't understand this sentence.

**AR:** *We corrected the sentence as follows (line 550):*

*"To further compare the methods, we provide below a detailed discussion of the major differences and present a synthesis of how the methods rank on average (averaged across all scenarios) for the quantitative evaluation criteria, which we support with further qualitative evaluation criteria (Table 4)."*

Line 488: *From the visual assessment*. Based on which figure?

**AR:** *we added additional explanation here (line 551):*

*"From the visual assessment, i.e. based on the method bias ($R_{bias}$),…"*

Table 4: The different ranking features should be linked to the scores presented in Table 3. Some of them deserve to be better explained in the text: *Independence from meteorological scenario, Independence from simulation years, Ease in application, Interpretation of prediction intervals*. They don't refer explicitly to scores, statistics or plots presented before.

**AR:** *More explanation was provided on these metrics in the text (lines 568-572):*

*"We hence introduce here a criterion independence from meteorological scenario, which defines how strong the selected sets depend on the meteorological scenario used for selection of representative parameter sets.*

*In a similar way, independence from simulation years will define how strong the selected sets depend on the simulation years used for selection of the representative parameter sets. To make statements on that, one needs to recall how the selection methods are constructed:…"*

*And in lines 577-585:*

*"Nevertheless, the ranking method can be considered as the (computationally and methodologically) easiest in application due to its selection criteria relying purely on ranking within individual simulation years. We call this criterion as ease in application. The other two methods need to be performed in the frequency space on sorted annual maxima over the entire simulation period and, in the case of the clustering method, require some additional computational effort (which remains low, however, compared to the hydrologic simulation).*

*The use of the frequency space in selecting the representative parameter sets helps, however, to interpret the constructed prediction intervals and to directly assign return periods to them. This speaks for their higher interpretability of prediction intervals as compared to the ranking method, in which interpretation of intervals is very limited (as they are selected without any flow frequency analysis)."*

Line 497-502: This lines could be put in section 2.5 to describe the assessment of the approach better.

**AR:** *as suggested, we moved these sentences to the method section.*

Line 514: *The other two methods need to be performed in the Gumbel space over the entire simulation period and, in the case of the clustering method, require some additional computational effort*. Once again this reference to "Gumbel space" is unappropriated given the scores computed, and the additional computational effort doesn't seem significant, completely justified by the added robustness.

**AR:** *we corrected it to the term 'frequency space'.*

Line 516: *The use of the Gumbel space in selecting the representative parameter sets helps, however, to interpret the constructed prediction intervals and to directly assign return periods to them.* Same remark as for line 295.

**AR:** *it was corrected to the term 'frequency space'.*

Lines 522-534: These lines are, in my opinion, a short summary of the study, and do not fit in this section (Limitations and perspectives).

**AR:** *Lines 522-529 were removed, while lines 529-534 were moved into the sect. 3.5., in which the set-up of the study is presented.*

Line 541: *This use of synthetic data makes the approach results independent from the catchment properties and limits the effect of the hydrological model error and errors in calibration data on the methods comparison results.* I may be more cautious on that, given that the scores and the ranking of the methods are somehow linked to 1) the variability of the ensemble models, which depends on the equifinality of the model's parameters, the calibration data and FOs, etc. and to 2) the meteorological variability of intense events generated by the weather generator which depends on the climatology, the scale etc. Only some tests on different catchments (in scale and climatology) could ground this assertion.

**AR:** *we thank for this comment. We rewritten this paragraph as follows (lines 611-615):*

*"We chose to use synthetic instead of real observed data to work with long enough continuous simulations that cover rare events and to minimize the focus of the model error arising from the calibration data and procedure. By using synthetic data as a reference (instead of observed data), the latter error can be here neglected. The proposed methods should be tested with more catchments and other models to verify the scoring of methods that was achieved in this study."*

Line 544: *We can, however, not directly assess here how much variability in the full hydrological ensemble is due to the climate variability and how much is due to the uncertainty resulting from the hydrological model parameters, because these two components are not linearly additive.* A simple exercise could help by 1) choosing a "median" model (in term of median response on the meteorological ensemble) and plotting/scoring the variability of simulations for the whole meteorological ensemble, and 2) choosing similarly a median meteorological scenario and simulating with the whole set of models and then 3) comparing the spread/variance of definite quantiles in the simulations. In my opinion, this is an indispensable complement to the presented results.

**AR:** *we thank the reviewer for this comment that showed us that we were not clear here about the purpose of our study. Separation of uncertainty intervals is not the aim of this work and the developed methodology for providing the uncertainty intervals based on representative sets. The selection methods should one enable to construct the uncertainty intervals based on three pre-selected parameter sets of a hydrological model. We use here different meteorological scenarios to verify how the selection criteria depend on a meteorological scenario and whether it is valid for different scenarios (independent meteorological conditions). Having such representative parameter sets of a hydrologic model selected, opens several avenues for further research. One example would be to separate uncertainty into contributing sources, i.e. hydrologic model, natural climatological variability and others. As contributions of such a separation will be case-specific – i.e. they depend on the selected hydrologic model, available meteorological scenarios, etc. – thus, they must be performed for each case study independently. We thank for suggestions on how such an analysis could be performed and these suggestions were included in the revised manuscript. We do not see however, a need to perform such*

*decomposition in our case, as our paper presents only the methodology for deriving representative parameter sets of a hydrologic model.*

*To clarify this issue, we modified our text, which the reviewer is referring to, into (lines 616-628):*

*"Selection methods proposed in this study enable one to choose representative parameter sets of a hydrologic model and based on those to construct predictive uncertainty intervals (PIs) for extreme flood analysis in the frequency space. Here, we tested the methodology using 100 meteorological scenarios that should represent the natural climate variability, and in this way, should provide independent conditions for methods' evaluation. Such a method for constructing PIs from a hydrological model ensemble is a powerful tool that opens several avenues for further detailed uncertainty analysis. For instance, one may be interested in contributions of different uncertainty sources into the total PIs constructed, e.g. coming from the hydrologic model or the natural climate variability. As these two components are not linearly additive, their separation is not straightforward. Also, any ensemble simulation also encompasses other uncertainty sources of the modeling chain, such as resulting from the weather generator or from the structure of the selected hydrological model, from the prediction of very rare flood events, etc. (Lamb & Kay 2004, Schumann et al. 2010, Kundzewicz et al. 2017). To assess individual contributions of interest, a simple sensitivity analysis based on the variance variability could be here recommended, in which one uncertainty source is propagated through the method at once while other sources are kept at their mode or median values and by comparing resulting PIs spread."*

Line 560: *Thus, the proposed selection methods could potentially be extended to account for different flood types.* Another option could be to consider Peak-Over-Threshold selection instead of a block selection (annual maximum) in building the simulated distributions. If different flood processes are present above a certain intensity threshold, flood type and seasonality sampling will be relevant.

*AR: we agree and added a following sentence in the text in lines 644-646:*

*"For that purpose, Peak-Over-Threshold (POT) selection criteria of flood peaks could be more appropriate, instead of a block selection (annual maximum) used here, in constructing the simulated distributions of hydrological responses, to cover a range of different flood processes."*

Line 566: *the three sets emulate the common practice of communicating median values along with prediction limits.* But in that case, these predictive intervals have to be statistically calibrated (or checked) in order to be used in safety studies, especially if these studies lead to engineering or compliance check.

*AR: we agree and added (lines 652):*

*"For safety-studies, these representative intervals should be additionally statistically proved."*

Line 572-584: Here again, the term *Gumbel space* could be replaced by "frequency domain" or equivalent.

*AR: this was changed into 'frequency domain'.*

Appendix A & Figure A1: Interesting but rather off-topic here, as only continuous hydrological simulation has been used in this study.

*AR: as suggested, we removed the Appendix A and Fig. A1 from the revised manuscript.*

Figure A3: For a better assessment of the distribution of calibrated parameters, I suggest that the scale of the horizontal axis of the violin plots (parameter values) exactly matches the corresponding calibration range written in the Table A1.

*AR: The figure was updated with initial calibration ranges as suggested.*

[Figure]

*Figure A1. Violin plots (blue) summarizing 100 optimized parameter sets of the HBV model for the Dünnern at Olten catchment vs. initial calibration ranges (gray). Units as in Table A1.*

Please also note the supplement to this comment: https://www.nat-hazards-earth-syst-sci-discuss.net/nhess-2020-79/nhess-2020-79-RC1-supplement.pdf

**Anonymous Referee #2**

*General comments:*

The contribution provides an interesting approach to the selection of representative parameter sets for continuous hydrological modelling in the framework of derived flood frequency analyses considering uncertainty. The methodology is quite clear and plausible. The manuscript is well written and concise. I have only some minor comments for improvement (see detailed comments).

*AR: We thank reviewer #2 for the positive feedback on our manuscript.*

*Detailed comments:*

1. Line 129: : : : "selected in step (d)" should read "selected in step (b)"

*AR: corrected in the text.*

2. Line 196: It is not clear to me how Q5, Q50 and Q95 are obtained? For each parameter set there is one of such quantiles. Are they averaged over all parameter sets or are they estimated as double quantiles (quantiles from the set of quantiles)?

*AR: Q5, Q50 and Q95 are estimated for each simulation point of sorted annual maxima in the frequency space, i.e. over all parameter sets. Hence, quantiling is done on each point of the sorted annual maxima and not on the entire simulation resulting from a single parameter set. To clarify this issue, we included additional text (lines 208-209):*

*"The 5%, 50% and 95% quantiles of these ensembles are computed* at each j-th point in the frequency space, resulting in quantiles *$Q_5$, $Q_{50}$ and $Q_{95}$ over the entire simulation period…"*

3. Line 344: I would suggest to put the figure A2 with the study region also in the main text.

*AR: thank you for this suggestion. The figure was moved into the main text.*

4. Line 446: I think the bias is "highest" for the ranking method and not "lowest".

*AR: yes, this is correct. Thank you for spotting this typo!*

5. Figures 7-10: I assume the "blue" range is bounded by the infimum and supremum, here coming from the 0.05 and 0.95 quantiles, meaning only 90% of the possible range are covered. What are the boundaries for the "grey" range? Is it covering 100%. May be this need to be indicated in the figure caption.

*AR: It is correct that the blue band is the coverage of the range bounded by the infimum and supremum parameter sets, i.e. 90% predictive intervals. The grey range corresponds to the band estimated using all 100 parameter sets. To clarify this issue, we included additional explanation in the figure captions.*

6. Limitations: This study uses sufficient long hourly discharge time series of 25 years for calibration on extremes. Often the hourly records are much shorter (e.g. 5 to 10 years) and a calibration on extremes is not feasible this way. Then, the calibration is done alternatively on observed flood statistics, for which often longer records are available, using synthetic rainfall as input. In this case the proposed procedure is hardly possible. Please discuss.

*AR: we agree that we are in a lucky situation when the hydrologic model can be calibrated with a continuous time series of more than 20 years. Yet, if such long time series are not available, other calibration procedure could be used (e.g. based on signatures) or model parameters could be required*

*from regionalization approaches. The way, the model is calibrated is not relevant for the selection methods as long as at least 100 parameter sets can be derived (and that can well represent rare floods). We added the following text in the limitation section (lines 592-607):*

*"Despite the calibration of a hydrological model lies beyond the scope of this paper, it is assumed that (at least) 100 parameter sets of a hydrologic model can be made available for selecting the representative parameter sets. For that purpose, a hydrological model should be calibrated with observed data of a long enough record that covers rare floods so that rare floods could be realistically simulated. In this work, to derive 100 parameter sets, we proposed a heuristic approach that relies on multiple independent model calibration trials using a Genetic Algorithm approach and a multi-objective function. This method represents an interesting solution to systematic sampling of the posterior parameter distributions (e.g. via Markov Chain Monte Carlo Sampling) or to any Monte Carlo method relying on a very high number of model runs. Its strength is that it can be applied for selecting parameter sets from independent model calibration settings (with different scores, calibration periods, etc.).*

*Note however that for the purpose of deriving 100 parameter sets, a continuous hydrologic model does not necessarily require continuous calibration data and it could be also calibrated to discrete data (e.g. using hydrologic signatures (Kavetski et al. 2018)). If no observed data or only too short records are available, model parameters can be acquired through regionalization approaches (see the work of Brunner et al. (2018a) for an overview of regionalization methods). The developed methods are of use for applications when a hydrologic model should be employed for simulations of rare floods. If the use of a hydrologic model is not possible, i.e. neither information for calibration nor sufficient information for parameter regionalization is available, these methods cannot be applied."*

7. Appendix A. This appendix is not really necessary from my point of view.

**AR:** *thank you for this suggestion. As both reviewers suggested to remove it, this appendix was removed from the revised manuscript.*

**References**

*Beven, K. and Freer, J.: Equifinality, data assimilation, and uncertainty estimation in mechanistic modelling of complex environmental systems using the GLUE methodology, Journal of Hydrology, 249, 11 − 29, https://doi.org/https://doi.org/10.1016/S0022-1694(01)00421-8, 2001.*

*Brunner, M. I. and Sikorska-Senoner, A. E.: Dependence of flood peaks and volumes in modeled discharge time series: Effect of different uncertainty sources, Journal of Hydrology, 572, 620 − 629, https://doi.org/https://doi.org/10.1016/j.jhydrol.2019.03.024, 2019.*

*Brunner, M. I., Furrer, R., Sikorska, A. E., Viviroli, D., Seibert, J., and Favre, A. C.: Synthetic design hydrographs for ungauged catchments: a comparison of regionalization methods, Stochastic Environmental Research and Risk Assessment, 32, 1993–2023, https://doi.org/10.1007/s00477-018-1523-3, 2018a.*

*Brunner, M. I., Sikorska, A. E., and Seibert, J.: Bivariate analysis of floods in climate impact assessments, Science of The Total Environment, 616-617, 1392–1403, https://doi.org/https://doi.org/10.1016/j.scitotenv.2017.10.176, 2018c.*

*Evin, G., Favre, A.-C., and Hingray, B.: Stochastic generation of multi-site daily precipitation focusing on extreme events, Hydrology and Earth System Sciences, 22, 655–672, https://doi.org/10.5194/hess-22-655-2018, 2018.*

*Evin, G., Favre, A.-C., and Hingray, B.: Stochastic generators of multi-site daily temperature: comparison of performances in various applications, Theoretical and Applied Climatology, 135, 811–824, https://doi.org/doi:10.1007/s00704-018-2404-x, 2019.*

*Kavetski, D., Fenicia, F., Reichert, P., & Albert, C.: Signature-domain calibration of hydrological models using approximate Bayesian computation: Theory and comparison to existing applications. Water Resources Research, 54, 4059– 4083. https://doi.org/10.1002/2017WR020528, 2018.*

*Müller-Thomy, H. and Sikorska-Senoner, A. E.: Does the complexity in temporal precipitation disaggregation matter for a lumped hydrological model?, Hydrological Sciences Journal, 64, 1453–1471, https://doi.org/10.1080/02626667.2019.1638926, 2019.*

*Sikorska, A. and Seibert, J.: Appropriate temporal resolution of precipitation data for discharge modelling in pre-alpine catchments, Hydrol. Sci. J., 61, 1–16, https://doi.org/10.1080/02626667.2017.1410279, 2018a.*

*Westerberg, I.K., Sikorska-Senoner, A.E., Viviroli, D., Vis, M., Seibert, J.: Hydrological model calibration with uncertain discharge data, Hydrological Sciences Journal, doi: 10.1080/02626667.2020.1735638, 2020.*

---

## Referee Report (RR1)

**Referee Report of**
**« Downsizing parameter ensembles for simulations of rare floods»**
**(NHESS-2020-79-version2)**

**General comments:**

I would like to congratulate the authors for great work of revision that has been done to issue this second version.

All the questions and comments of the reviewers have been addressed, clearly and with many details, both in the "author responses" and in the revised manuscript. New metrics have been introduced to complete the ones from the first version. New figures have been added, and some of the existing ones have been modified, giving more readability to the methods and to the results.

Overall it's a rich paper, sometimes demanding to read, but presenting extensively an interesting method.

I would recommend the publication of this second version "as is", with only one minor modification of a figure as recommended below.

**Detailed comments/questions:**

Figure 5 : two modifications could improve the readability of this interesting figure :
- For each parameter, the scale of the boxplot could be the same scale as in Table A1 and Figure A1.
- The color mix of the Ranking/Quantiling/Clustering method is sometimes muddled. For each subset (S/M/I), 3 barplots (Ranking/Quantiling/Clustering) could be stacked instead of being mixed, in my opinion giving a clearer vision of the variability of each subset.